# Structure *from* Duplicates:
# Neural Inverse Graphics from a Pile of Objects

**Tianhang Cheng[1]   Wei-Chiu Ma[2]   Kaiyu Guan[1]   Antonio Torralba[2]   Shenlong Wang[1]**

[1]University of Illinois Urbana-Champaign   [2]Massachusetts Institute of Technology

`{tcheng12, kaiyug, shenlong}@illinois.edu`
`{weichium, torralba}@mit.edu`

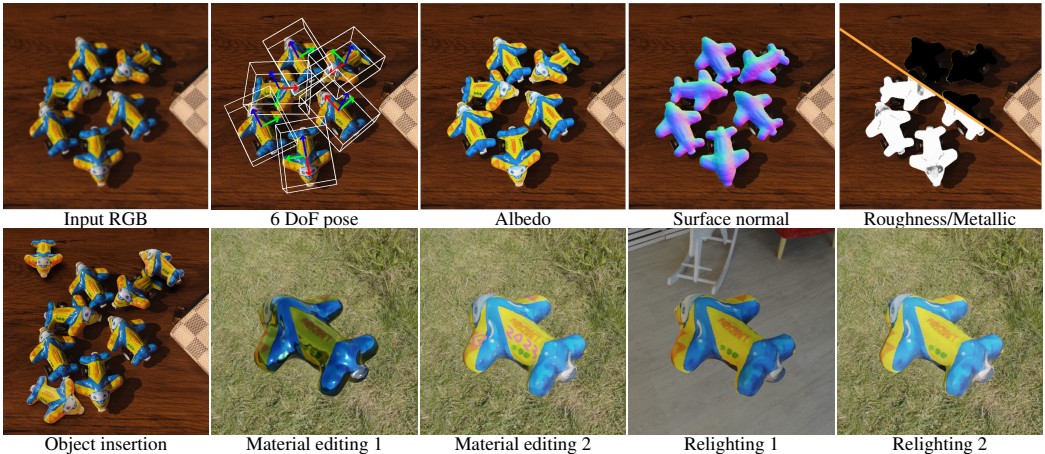

Figure 1: **Structure from duplicates (SfD)** is a novel inverse graphics framework that reconstructs geometry, material, and illumination from a single image containing multiple identical objects.

## Abstract

Our world is full of identical objects (*e.g.*, cans of coke, cars of same model). These duplicates, when seen together, provide additional and strong cues for us to effectively reason about 3D. Inspired by this observation, we introduce Structure from Duplicates (SfD), a novel inverse graphics framework that reconstructs geometry, material, and illumination from a single image containing multiple identical objects. SfD begins by identifying multiple instances of an object within an image, and then jointly estimates the 6DoF pose for all instances. An inverse graphics pipeline is subsequently employed to jointly reason about the shape, material of the object, and the environment light, while adhering to the shared geometry and material constraint across instances. Our primary contributions involve utilizing object duplicates as a robust prior for single-image inverse graphics and proposing an in-plane rotation-robust Structure from Motion (SfM) formulation for joint 6-DoF object pose estimation. By leveraging multi-view cues from a single image, SfD generates more realistic and detailed 3D reconstructions, significantly outperforming existing single image reconstruction models and multi-view reconstruction approaches with a similar or greater number of observations. Code available at https://github.com/Tianhang-Cheng/SfD

## 1   Introduction

Given a single/set of image(s), the goal of inverse rendering is to recover the underlying geometry, material, and lighting of the scene. The task is of paramount interest to many applications in computer vision, graphics, and robotics and has drawn extensive attention across the communities over the past few years[74; 47; 21; 45].

37th Conference on Neural Information Processing Systems (NeurIPS 2023).

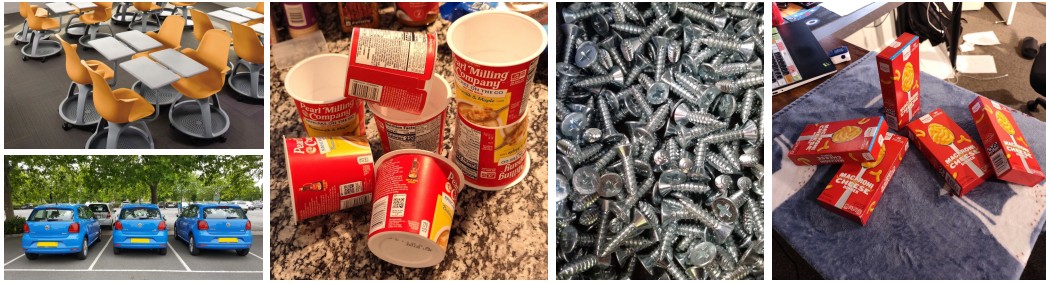

Figure 2: **Repetitions in the visual world.** Our physical world is full of identical objects (*e.g.*, cans of coke, cars of the same model, chairs in a classroom). These duplicates, when seen together, provide additional and strong cues for us to effectively reason about 3D.

Since the problem is ill-posed, prevailing inverse rendering approaches often leverage multi-view observations to constrain the solution space. While these methods have achieved state-of-the-art performance, in practice, it is sometimes difficult, or even impossible, to obtain those densely captured images. To overcome the reliance on multi-view information, researchers have sought to incorporate various structural priors, either data-driven or handcrafted, into the models[5; 36]. By utilizing the regularizations, these approaches are able to approximate the intrinsic properties (*e.g.*, material) and extrinsic factors (*e.g.*, illumination) even from one single image. Unfortunately, the estimations may be biased due to the priors imposed. This makes one ponder: is it possible that we take the best of both worlds? Can we extract multi-view information from a single image under certain circumstances?

Fortunately the answer is yes. Our world is full of repetitive objects and structures. Repetitive patterns in single images can help us extract and utilize multi-view information. For instance, when we enter an auditorium, we often see many identical chairs facing slightly different directions. Similarly, when we go to a supermarket, we may observe multiple nearly-identical apples piled on the fruit stand. Although we may not see *the exact same object* from multiple viewpoints in just one glance, we do see many of the "identical twins" from various angles, which is equivalent to multi-view observations and even more (see Sec. 3 for more details). Therefore, the goal of this paper is to develop a computational model that can effectively infer the underlying 3D representations from a single image by harnessing the repetitive structures of the world.

With these motivations in mind, we present Structure from Duplicates (SfD), a novel inverse rendering model that is capable of recovering high-quality geometry, material, and lighting of the objects from a single image. SfD builds upon insights from structure from motion (SfM) as well as recent advances on neural fields. At its core lies two key modules: (i) a *in-plane rotation robust pose estimation module*, and (ii) a *geometric reconstruction module*. Given an image of a scene with duplicate objects, we exploit the pose estimation module to estimate the relative 6 DoF poses of the objects. Then, based on the estimated poses, we align the objects and create multiple "virtual cameras." This allows us to effectively map the problem from a single-view multi-object setup to a multi-view single-object setting (see Fig. 3). Finally, once we obtain multi-view observations, we can leverage the geometric module to recover the underlying intrinsic and extrinsic properties of the scene. Importantly, SfD can be easily extended to multi-image setup. It can also be seen as a superset of existing NeRF models, where the model will reduce to NeRF when there is only one single object in the scene.

We validate the efficacy of our model on a new dataset called **Dup**, which contains synthetic and real-world samples of duplicated objects since current multi-view datasets lack duplication samples. This allows us to benchmark inverse rendering performance under single-view or multi-view settings. Following previous work [60; 65; 47; 69; 74], we evaluate rendering, relighting and texture quality with MSE, PSNR, SSIM, LPIPS [70], geometry with Chamfer Distance (CD), and environment light with MSE.

Our contributions are as follows: 1) We proposed a novel setting called "single-view duplicate objects" (S-M), which expands the scope of the inverse graphics family with a multi-view single-instance (M-S) framework. 2) Our method produces more realistic material texture than the existing multi-view inverse rendering model when using the same number of training views. 3) Even only relying on a single-view input, our approach can still recover comparable or superior materials and geometry compared to baselines that utilize multi-view images for supervision.

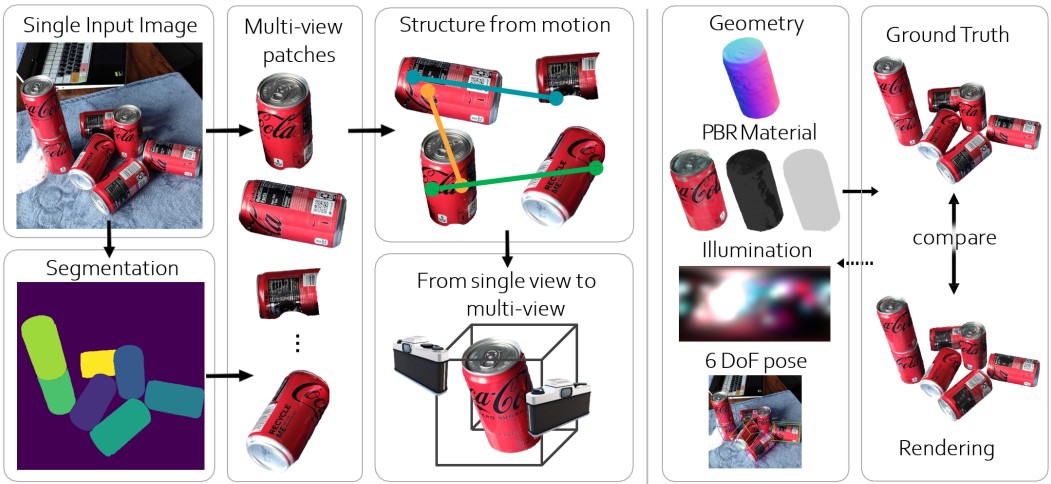

Figure 3: **Method overview: (Left)** SfD begins by identifying multiple instances of an object within an image, and then jointly estimates the 6DoF pose for all instances. **(Right)** An inverse graphics pipeline is subsequently employed to reason about the shape, material of the object, and the environment light, while adhering to the shared geometry and material constraint across instances.

## 2 Related Work

**Inverse rendering:** The task of inverse rendering can be dated back to more than half a century ago [31; 23; 24; 2]. The goal is to factorize the appearance of an object or a scene in the observed image(s) into underlying geometry, material properties, and lighting conditions [53; 43; 66]. Since the problem is severely under-constrained, previous work mainly focused on controlled settings or simplifications of the problem [19; 22; 1]. For instance, they either assume the reflectance of an object is spatially invariant [69], presume the lighting conditions are known [57], assume the materials are lambertian [76; 41], or presume the proxy geometry is available [52; 32; 10; 30]. More recently, with the help of machine learning, in particular deep learning, researchers have gradually moved towards more challenging in-the-wild settings [73; 47]. By pre-training on a large amount of synthetic yet realistic data [36] or baking inductive biases into the modeling pipeline [7; 72], these approaches can better tackle unconstrained real-world scenarios (*e.g.*, unknown lighting conditions) and recover the underly physical properties more effectively [47; 68]. For example, through properly modeling the indirect illumination and the visibility of direct illumination, Zhang *et al.* [74] is able to recover interreflection- and shadow-free SVBRDF materials (*e.g.*, albedo, roughness). Through disentangling complex geometry and materials from lighting effects, Wang *et al.* [62] can faithfully relight and manipulate a large outdoor urban scene. Our work builds upon recent advances in neural inverse rendering. Yet instead of grounding the underlying physical properties through multi-view observations as in prior work, we focus on the single image setup and capitalize on the duplicate objects in the scene for regularization. The repetitive structure not only allows us to ground the geometry, but also provide additional cues on higher-order lighting effects (*e.g.*, cast shadows). As we will show in the experimental section, we can recover the geometry, materials, and lighting much more effectively even when comparing to multi-view observations.

**3D Reconstruction:** Recovering the spatial layout of the cameras and the geometry of the scene from a single or a collection of images is a longstanding challenge in computer vision. It is also the cornerstone for various downstream applications in computer graphics and robotics such as inverse rendering [43; 60; 62], 3D editing [39; 63], navigation [42; 67], and robot manipulation [27; 38]. Prevailing 3D reconstruction systems, such as structure from motion (SfM), primarily rely on multi-view geometry to estimate the 3D structure of a scene [40; 20; 54]. While achieving significant successes, they rely on densely captured images, limiting their flexibility and practical use cases. Single image 3D reconstruction, on the other hand, aims to recover metric 3D information from a monocular image [14; 6; 50; 49]. Since the problem is inherently ill-posed and lacks the ability to leverage multi-view geometry for regularization, these methods have to resort to (learned) structural priors to resolve the ambiguities. While they offer greater flexibility, their estimations may inherit biases from the training data. In this paper, we demonstrate that, under certain conditions, it is possible to incorporate multi-view geometry into a single image reconstruction system. Specifically, we leverage repetitive objects within the scene to anchor the underlying 3D structure. By treating

each of these duplicates as an observation from different viewpoints, we can achieve highly accurate metric 3D reconstruction from a single image.

**Repetitions:** Repetitive structures and patterns are ubiquitous in natural images. They play important roles in addressing numerous computer vision problems. For instance, a single natural image often contains substantial redundant patches [77]. The recurrence of small image patches allows one to learn a powerful prior which can later be utilized for various tasks such as super-resolution [18; 26], image deblurring [44], image denoising [15], and texture synthesis [12]. Moving beyond patches, repetitive primitives or objects within the scene also provide informative cues about their intrinsic properties [33; 64]. By sharing or regularizing their underlying representation, one can more effectively constrain and reconstruct their 3D geometry [25; 16; 9], as well as enable various powerful image/shape manipulation operations [34; 59]. In this work, we further push the boundary and attempt to recover not just the geometry, but also the materials (*e.g.*, albedo, roughness), visibilities, and lighting conditions of the objects. Perhaps closest to our work is [75]. Developed independently and concurrently, Zhang *et al.* build a generative model that aims to capture object intrinsics from a single image with multiple similar/same instances. However, there exist several key differences: 1) we explicitly recover metric-accurate camera poses using multi-geometry, whereas Zhang et al. learn this indirectly through a GAN-loss; 2) we parameterize and reason realistic PBR material and environmental light; 3) we handle arbitrary poses, instead of needing to incorporate a prior pose distribution. Some previous work also tried to recover 6D object pose from crowded scene or densely packed objects[46; 11] from RGB-D input, but our model only require RGB input.

## 3 Structure from Duplicates

In this paper, we seek to devise a method that can precisely reconstruct the geometry, material properties, and lighting conditions of an object from *a single image containing duplicates of it*. We build our model based on the observation that repetitive objects in the scene often have different poses and interact with the environment (*e.g.*, illumination) differently. This allows one to extract rich *multi-view* information even from one single view and enables one to recover the underlying physical properties of the objects effectively.

We start by introducing a method for extracting the "multi-view" information from duplicate objects. Then we discuss how to exploit recent advances in neural inverse rendering to disentangle both the object intrinsics and environment extrinsics from the appearance. Finally, we describe our learning procedure and design choices.

### 3.1 Collaborative 6-DoF pose estimation

As we have alluded to above, a single image with multiple duplicate objects contains rich multi-view information. It can help us ground the underlying geometry and materials of the objects, and understand the lighting condition of the scene.

Our key insight is that the image can be seen as *a collection of multi-view images stitching together*. By cropping out each object, we can essentially transform the single image into a set of multi-view images of the object from various viewpoints. One can then leverage structure from motion (SfM) [55] to estimate the relative poses among the multi-view images, thereby aggregating the information needed for inverse rendering. Notably, the estimated camera poses can be inverted to recover the 6 DoF poses of the duplicate objects. As we will elaborate in Sec. 3.2, this empowers us to more effectively model the extrinsic lighting effect (which mainly depends on world coordinate) as well as to properly position objects in perspective and reconstruct the exact same scene.

To be more formal, let $\mathcal{I} \in \mathbb{R}^{H \times W \times 3}$ be an image with $N$ duplicate objects. Let $\{\mathcal{I}_i^{\text{obj}}\}_{i=1}^N \in \mathbb{R}^{w \times h \times 3}$ be the corresponding object image patches. We first leverage SfM [55; 51] to estimate the camera poses of the multi-view cropped images* $\{\boldsymbol{\xi}_i \in \mathbb{SE}(3)\}_{i=1}^N$:

$$\boldsymbol{\xi}_1^{\text{cam}}, \boldsymbol{\xi}_2^{\text{cam}}, ..., \boldsymbol{\xi}_N^{\text{cam}} = f^{\text{SfM}}(\mathcal{I}_1^{\text{obj}}, \mathcal{I}_2^{\text{obj}}, ..., \mathcal{I}_K^{\text{obj}}). \tag{1}$$

Next, since there is only one real camera in practice, we can simply align the $N$ virtual cameras $\{\boldsymbol{\xi}_i\}_{i=1}^N$ to obtain the 6 DoF poses of the duplicate objects. Without loss of generality and for

---

*In practice, the cropping operation will change the intrinsic matrix of the original camera during implementation. For simplicity, we assume the intrinsics are properly handled here.

simplicity, we align all the cameras to a reference coordinate $\boldsymbol{\xi}^{\text{ref}}$. The 6 DoF poses of the duplicate objects thus become $\boldsymbol{\xi}_i^{\text{obj}} = \boldsymbol{\xi}^{\text{ref}} \circ (\boldsymbol{\xi}_i^{\text{cam}})^{-1}$, where $\circ$ is matrix multiplication for pose composition.

In practice, we first employ a state-of-the-art panoptic segmentation model [8] to segment all objects in the scene. Then we fit a predefined bounding box to each object and crop it. Lastly, we run COLMAP [55] to estimate the 6 DoF virtual camera poses, which in turn provides us with the 6 DoF object poses. Fig. 3(left) depicts our collaborative 6-DoF pose estimation process.

**Caveats of random object poses:** Unfortunately, naively feeding these object patches into SfM would often leads to failure, as little correspondence can be found. This is due to the fact that state-of-the-art correspondence estimators [51] are trained on Internet vision data, where objects are primarily upright. The poses of the duplicate objects in our case, however, vary significantly. Moreover, the objects are often viewed from accidental viewpoints [17]. Existing estimators thus struggle to match effectively across such extreme views.

**Rotation-aware data augmentation:** Fortunately, the scene contains numerous duplicate objects. While estimating correspondences reliably across arbitrary instances may not always be possible, there are certain objects whose viewpoints become significantly similar after in-plane rotation. Hence, we have developed an in-plane rotation-aware data augmentation for correspondence estimation.

Specifically, when estimating correspondences between a pair of images, we don't match the images directly. Instead, we gradually rotate one image and then perform the match. The number of correspondences at each rotation is recorded and then smoothed using a running average. We take the `argmax` to determine the optimal rotation angle. Finally, we rotate the correspondences from the best match back to the original pixel coordinates. As we will demonstrate in Sec. 4, this straightforward data augmentation strategy significantly improves the accuracy of 6 DoF pose estimation. In practice, we rotate the image by $4°$ per step. All the rotated images are batched together, enabling us to match the image pairs in a single forward pass.

## 3.2 Joint shape, material, and illumination estimation

Suppose now we have the 6 DoF poses of the objects $\{\boldsymbol{\xi}_i^{\text{obj}}\}_{i=1}^N$. The next step is to aggregate the information across duplicate objects to recover the *intrinsics properties of the objects* (*e.g.*, geometry, materials) and *the extrinsic factors of the world* (*e.g.*, illumination). We aim to reproduce these attributes as faithfully as possible, so that the resulting estimations can be utilized for downstream tasks such as relighting and material editing. Since the task is under-constrained and joint estimation often leads to suboptimal results, we follow prior art [74; 62] and adopt a stage-wise procedure.

**Geometry reconstruction:** We start by reconstructing the geometry of the objects. In line with NeuS [60], we represent the object surfaces as the zero level set of a signed distance function (SDF).

We parameterize the SDF with a multi-layer perceptron (MLP) $S : \mathbf{x}^{\text{obj}} \mapsto s$ that maps a 3D point under object coordinate $\mathbf{x}^{\text{obj}} \in \mathbb{R}^3$ to a signed distance value $s \in \mathbb{R}$. Different from NeuS [60], we model the geometry of objects in local object space. This allows us to guarantee shape consistency across instances by design.

We can also obtain the surface normal by taking the gradient of the SDF: $\mathbf{n}(\mathbf{x}^{\text{obj}}) = \nabla_{\mathbf{x}^{\text{obj}}} S$. To learn the geometry from multi-view images, we additionally adopt an *auxiliary* appearance MLP $C : \{\mathbf{x}, \mathbf{x}^{\text{obj}}, \mathbf{n}, \mathbf{n}^{\text{obj}}, \mathbf{d}, \mathbf{d}^{\text{obj}}\} \mapsto \mathbf{c}$ that takes as input a 3D point $\mathbf{x}$, $\mathbf{x}^{\text{obj}}$, surface normal $\mathbf{n}$, $\mathbf{n}^{\text{obj}}$, and view direction $\mathbf{d}$, $\mathbf{d}^{\text{obj}}$ under both coordinate systems and outputs the color $\mathbf{c} \in \mathbb{R}^3$. The input from world coordinate system helps the MLP to handle the appearance inconsistencies across instances caused by lighting or occlusion. We tied the weights of the early layers of $C$ to those of $S$ so that the gradient from color can be propagated to geometry. We determine which object coordinate to use based on the object the ray hits. This information can be derived either from the instance segmentation or another allocation MLP $A : \{\mathbf{x}, \mathbf{d}\} \mapsto q$ (distilled from SDF MLP), where $q \in \mathbb{N}$ is the instance index. After we obtain the geometry MLP $S$, we discard the auxiliary appearance MLP $C$. As we will discuss in the next paragraph, we model the object appearance using physics-based rendering (PBR) materials so that it can handle complex real-world lighting scenarios.

**Material and illumination model:** Now we have recovered the geometry of the objects, the next step is to estimate the environment light of the scene as well as the material of the object. We assume

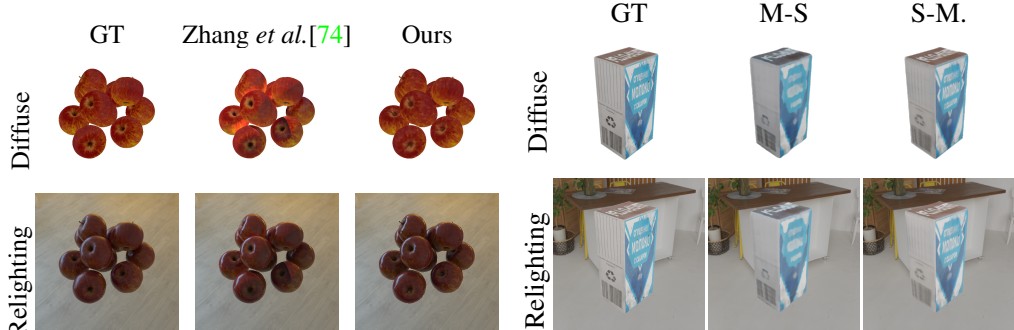

Figure 4: **Multi-view inverse rendering.**

Figure 5: **Multi-view single object (M-S) vs single-view multi-objects (S-M).**

all lights come from an infinitely faraway sphere and only consider direct illumination. Therefore, the illumination emitted to a 3D point in a certain direction is determined solely by the incident light direction $\boldsymbol{w}_i$ and is independent of the point's position. Similar to [69; 74], we approximate the environment light with $M = 128$ Spherical Gaussians (SGs):

$$L_i\left(\omega_i\right) = \sum\nolimits_{k=1}^{M} \boldsymbol{\mu}_k e^{\lambda_k(\boldsymbol{w}_i \cdot \boldsymbol{\phi}_k - 1)}, \tag{2}$$

where $\lambda \in \mathbb{R}^+$ is the lobe sharpness, $\boldsymbol{\mu}$ is the lobe amplitude, and $\boldsymbol{\phi}$ is the lobe axis. This allows us to effectively represent the illumination and compute the rendering equation (Eq. 3) in closed-form.

As for object material, we adopt the simplified Disney BRDF formulation [3; 29] and parameterized it as a MLP $M : \mathbf{x}^{\text{obj}} \mapsto \{\mathbf{a}, r, m\}$. Here, $\mathbf{a} \in \mathbb{R}^3$ denotes albedo, $r \in [0, 1]$ corresponds to roughness, and $m \in [0, 1]$ signifies metallic. Additionally, inspired by InvRender [74], we incorporate a visibility MLP $V : (\mathbf{x}, \boldsymbol{w}_i) \mapsto v \in [0, 1]$ to approximate visibility for each world coordinate faster reference. The difference is that we use the sine activation function[56] rather than the ReLU activation for finer detail. Since the visibility field is in world space, we can model both inter-object self-casted shadows and inter-object occlusions for multiple instances in our setup. We query only surface points, which can be derived from the geometry MLP $S$ using volume rendering. The material MLP $M$ also operates in object coordinates like $S$, which ensure material consistency across all instances. Moreover, the variations in lighting conditions between instances help us better disentangle the effects of lighting from the materials. We set the dielectrics Fresnel term to $F_0 = 0.02$ and the general Fresnel term to $\mathbf{F} = (1 - m)F_0 + m\mathbf{a}$ to be compatible with both metals and dielectrics.

Combining all of these components, we can generate high-quality images by integrating the visible incident lights from hemisphere and modeling the effects of BRDF [28]:

$$L_o(\boldsymbol{w}_o; \mathbf{x}) = \int_\Omega L_i(\boldsymbol{w}_i) f_r(\boldsymbol{w}_i, \boldsymbol{w}_o; \mathbf{x})(\boldsymbol{w}_i \cdot \mathbf{n}) d\boldsymbol{w}_i. \tag{3}$$

Here, $\boldsymbol{w}_i$ is the incident light direction, while $\boldsymbol{w}_o$ is the viewing direction. The BRDF function $f_r$ can be derived from our PBR materials. We determine the visibility of an incident light either through sphere tracing or following Zhang *et al.* [74] to approximate it with the visibility MLP $V$. We refer the readers to the supplementary materials for more details.

### 3.3 Optimization

Optimizing shape, illumination, and material jointly from scratch is challenging. Taking inspiration from previous successful approaches [74; 69], we implement a multi-stage optimization pipeline. We progressively optimize the geometry first, then visibility, and finally, material and illumination.

**Geometry optimization:** We optimize the geometry model by minimizing the difference between rendered cues and observed cues.

$$\min_{S,C} E_{\text{color}} + \lambda_1 E_{\text{reg}} + \lambda_2 E_{\text{mask}} + \lambda_3 E_{\text{normal}}, \tag{4}$$

where $\lambda_1 = 0.1$, $\lambda_2 = \lambda_3 = 0.5$. And each term is defined as follows:

| Multi-view | Albedo | Roughness | Relighting | Env Light | Geometry | Single-view | Albedo | Roughness | Relighting | Env Light | Geometry |
|---|---|---|---|---|---|---|---|---|---|---|---|
| | PSNR ↑ | MSE ↓ | PSNR ↑ | MSE ↓ | CD ↓ | | PSNR ↑ | MSE ↓ | PSNR ↑ | MSE ↓ | CD ↓ |
| PhySG | 16.233 | 0.087 | 21.323 | 0.054 | 0.024 | PhySG* | 14.977 | 0.255 | 18.504 | 0.082 | **0.033** |
| Nv-DiffRec | 16.123 | 0.116 | 17.418 | 0.168 | 0.268 | Nv-DiffRec* | 14.021 | 0.165 | 17.214 | 0.067 | 0.050 |
| InvRender | 16.984 | 0.084 | 22.224 | 0.067 | 0.024 | InvRender* | 14.724 | 0.247 | 17.998 | 0.082 | 0.033 |
| Ours | **21.961** | **0.026** | **25.486** | **0.029** | **0.011** | Ours | **17.629** | **0.062** | **21.374** | **0.052** | 0.034 |

Table 1: **(Left) Multi-view inverse rendering on synthetic data**. Both our model and the baseline are trained on multi-view images. Our model is significantly better than baseline in terms of geometry and PBR texture. **(Right) Single-view inverse rendering on synthetic data.** While our model is trained on a *single-view image*, the baselines ∗ are trained on 10 *multi-view images* of the same scene.

- The *color consistency term* $E_{\text{color}}$ is a L2 color consistency loss between the rendered color $\mathbf{c}$ and the observed color $\hat{\mathbf{c}}$ for all pixel rays: $E_{\text{color}} = \sum_{\mathbf{r}} \|\mathbf{c_r} - \mathbf{c_r}\|_2$.

- The *normal consistency term* $E_{\text{normal}}$ measures the rendered normal $\hat{\mathbf{n}}$ and a predicted normal $\hat{\mathbf{n}}$: $E_{\text{normal}} = \sum_{\mathbf{r}} \|1 - \hat{\mathbf{n}}_{\mathbf{r}}^{\text{T}} \mathbf{n_r}\|_1 + \|\hat{\mathbf{n}}_{\mathbf{r}} - \mathbf{n_r}\|_1$. Our monocular predicted normal $\hat{\mathbf{n}}$ is obtained using a pretrained Omnidata model [13].

- The *mask consistency term* $E_{\text{mask}}$ measures the discrepancy between the rendered mask $\mathbf{m_r}$ and the observed mask $\hat{\mathbf{m}}_{\mathbf{r}}$, in terms of binary cross-entropy (BCE): $L_{\text{mask}} = \sum_{\mathbf{r}} \text{BCE}\,(\mathbf{m_r}, \hat{\mathbf{m}}_{\mathbf{r}})$.

- Finally, inspired by NeuS [60], we incorporate an *Eikonal regularization* to ensure the neural field is a valid signed distance field: $L_{\text{eikonal}} = \sum_{\mathbf{x}} (\|\nabla_{\mathbf{x}} S\|_2 - 1)^2$,

**Visibility optimization:** Ambient occlusion and self-cast shadows pose challenges to the accuracy of inverse rendering, as it's difficult to separate them from albedo when optimizing photometric loss. However, with an estimated geometry, we can already obtain a strong visibility cue. Consequently, we utilize ambient occlusion mapping to prebake the visibility map onto the object surfaces obtained from the previous stage. We then minimize the visibility consistency term to ensure the rendered visibility $v_{\mathbf{r}}$ from MLP $V$ aligns with the derived visibility $\hat{v}_r$: $\min_V \sum_{\mathbf{r}} \text{BCE}(v_{\mathbf{r}}, \hat{v}_{\mathbf{r}})$.

**Material and illumination optimization:** In the final stage, given the obtained surface geometry and the visibility network, we jointly optimize the environmental light and the PBR material network. The overall objective is as follows:

$$\min_{M, \boldsymbol{\omega}, \boldsymbol{\phi}} E_{\text{color}} + \lambda_4 E_{\text{sparse}} + \lambda_5 E_{\text{smooth}} + \lambda_6 E_{\text{metal}}, \tag{5}$$

where $\lambda_4 = 0.01$, $\lambda_5 = 0.1$, $\lambda_6 = 0.01$. The four terms are defined as follows:

- The *color consistency term* $E_{\text{color}}$ minimizes the discrepancy between the rendered color and observed color, akin to the geometry optimization stage. However, we use PBR-shaded color in place of the color queried from the auxiliary radiance field.

- The *sparse regularization* $E_{\text{sparse}}$ constrains the latent code $\boldsymbol{\rho}$ of the material network to closely align with a constant target vector $\boldsymbol{\rho}'$: $E_{\text{latent}} = \text{KL}\,(\boldsymbol{\rho} \| \boldsymbol{\rho}')$. We set $\boldsymbol{\rho}' = 0.05$.

- The *smooth regularization* $E_{\text{smooth}}$ force the BRDF decoder to yield similar value for close latent code $\boldsymbol{z}$: $E_{smooth} = \|D(\boldsymbol{z}) - D(\boldsymbol{z} + \boldsymbol{dz})\|_1$, where $\boldsymbol{dz}$ is randomly sampled from normal distribution $N(0; 0.01)$.

- Lastly, inspired by the fact that most common objects are either metallic or not, we introduce a *metallic regularization* $L_{\text{metal}}$ to encourage the predicted metallic value to be close to either 0 or 1: $L_{\text{metal}} = \sum_{\mathbf{r}} m_{\mathbf{r}} (1 - m_{\mathbf{r}})$.

## 4 Experiment

In this section, we evaluate the effectiveness of our model on synthetic and real-world datasets, analyze its characteristics, and showcase its applications.

### 4.1 Experiment setups

**Data:** Since existing multi-view datasets do not contain duplicate objects, we collect ***Dup***, a novel inverse rendering dataset featuring various duplicate objects. *Dup* consists of 13 synthetic and 6

| | Albedo | Roughness | Env Light | Mesh | Rotation | Translation |
|---|---|---|---|---|---|---|
| **Num** | **PSNR ↑** | **MSE ↓** | **MSE ↓** | **CD ↓** | **degree ↓** | **length ↓** |
| 6 | 18.48 | 0.13 | 0.116 | 0.025 | 2.515 | 0.070 |
| 8 | 18.26 | 0.09 | 0.172 | 0.017 | 0.600 | 0.003 |
| 10 | 18.99 | 0.09 | 0.204 | 0.029 | 0.186 | 0.003 |
| 20 | 19.22 | 0.09 | 0.063 | 0.021 | 0.299 | 0.005 |
| 30 | 19.69 | 0.07 | 0.119 | 0.020 | 0.448 | 0.006 |
| 50 | 18.56 | 0.08 | 0.153 | 0.038 | 0.464 | 0.008 |
| 60 | 16.45 | 0.14 | 0.150 | 0.091 | 51.512 | 0.644 |

| | Albedo | Roughness | Metallic | Relighting |
|---|---|---|---|---|
| **Model** | **PSNR ↑** | **MSE ↓** | **MSE ↓** | **PSNR ↑** |
| full model | **17.27** | 0.30 | 0.02 | **19.95** |
| w/o clean seg | 16.68 | 0.23 | 0.13 | 19.12 |
| w/o metal loss | 17.09 | **0.19** | 0.08 | 19.02 |
| w/o latent smooth | 17.00 | 0.23 | **0.02** | 19.71 |
| w/o normal | 16.51 | 0.42 | 0.97 | 18.81 |
| w/o eik loss | 16.79 | 0.51 | 0.97 | 19.31 |
| w/o mask | 15.14 | 0.29 | 0.02 | 18.11 |

Table 2: **(Left) Performance vs. number of duplicates for "color box" dataset.** We highlight the best , second and third values. **(Right) Ablation study for the contribution of each loss term.**

| | Rendering | | |
|---|---|---|---|
| | **PSNR ↑** | **SSIM ↑** | **LPIPS ↓** |
| PhySG* | **20.624** | 0.641 | 0.263 |
| Nv-DiffRec* | 18.818 | 0.569 | 0.282 |
| InvRender* | 20.665 | 0.639 | 0.262 |
| Ours | 20.326 | **0.660** | **0.192** |

| | Albedo | Roughness | Relighting | Env Light | Geometry |
|---|---|---|---|---|---|
| | **PSNR ↑** | **MSE ↓** | **PSNR ↑** | **MSE ↓** | **CD ↓** |
| M-S | 20.229 | 0.096 | 21.328 | **0.045** | 0.010 |
| S-M | **23.448** | **0.050** | **24.254** | 0.052 | **0.007** |

Table 3: **(Left) Single-view inverse rendering on real-world data**. ∗ indicates that the baselines are trained on multi-view observations. **(Right) Multi-view single-object (M-S) vs. single-view multi-object (S-M).**

real-world scenes, each comprising 5-10 duplicate objects such as cans, bottles, fire hydrants, etc. For synthetic data, we acquire 3D assets from PolyHaven[†] and utilize Blender Cycles for physics-based rendering. We generate 10-300 images per scene. As for the real-world data, we place the objects in different environments and capture 10-15 images using a mobile phone. The data allows for a comprehensive evaluation of the benefits of including duplicate objects in the scene for inverse rendering. We refer the readers to the supp. materials for more details.

**Metrics:** Following prior art [74; 69], we employ Peak Signal-to-Noise Ratio (PSNR), Structural Similarity (SSIM), and LPIPS [71] to assess the quality of rendered and relit images. For materials, we utilize PSNR to evaluate albedo, and mean-squared error (MSE) to quantify roughness and environmental lighting. And following[60; 65; 47], we leverage the Chamfer Distance to measure the distance between our estimated geometry and the ground truth.

**Baselines:** We compare against three state-of-the-art multi-view inverse rendering approaches: Physg [69], InvRender [74], and NVdiffrec [47]. Physg and InvRender employ implicit representations to describe geometric and material properties, while NVdiffrec utilizes a differentiable mesh representation along with UV textures. Additionally, we enhance Physg by equipping it with a spatially-varying roughness.

**Implementation details:** We use the Adam optimizer with an initial learning rate 1e-4. All experiments are conducted on a single Nvidia A40 GPU. The first stage takes 20 hours, and the 2nd and 3rd stage takes about 2 hours. Please refer to the supp. material for more details.

### 4.2 Experimental results

**Single-view inverse rendering:** We first evaluate our approach on the single-image multi-object setup. Since the baselines are not designed for this particular setup, we randomly select another 9 views, resulting in a total of 10 multi-view images, to train them. As shown in Tab. 1(right), we are able to leverage duplicate objects to constrain the underlying geometry, achieving comparable performance to the multi-view baselines. The variations in lighting conditions across instances also aid us in better disentangling the effects of lighting from the materials.

**Multi-view inverse rendering:** Our method can be naturally extended to the multi-view setup, allowing us to validate its effectiveness in traditional inverse rendering scenarios. We utilize synthetic data to verify its performance. For each scene, we train our model and the baselines using 100 different views and evaluate the quality of the reconstruction results. Similar to previous work, we

---

[†]https://polyhaven.com/models

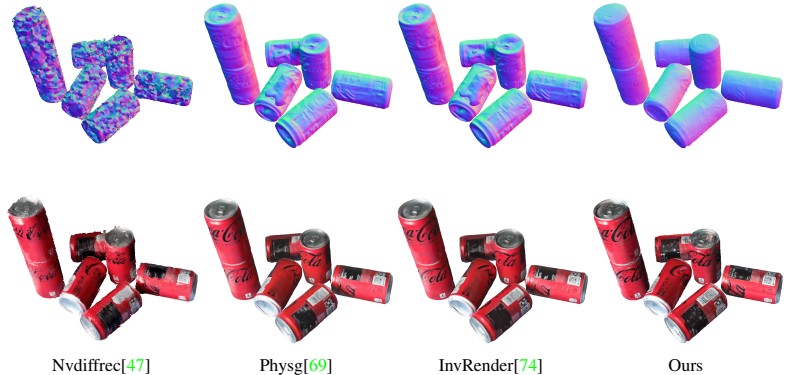

| Nvdiffrec[47] | Physg[69] | InvRender[74] | Ours |

Figure 6: **The surface normal and rendering result on real-world cola image.** Our model has the smoothest surface normal compared with other baselines.

assume the ground truth poses are provided. As shown in Tab. 1(left), we outperform the baselines on all aspects. We conjecture that this improvement stems from our approach explicitly incorporating a material- and geometry-sharing mechanism during the modeling process. As a result, we have access to a significantly larger number of "effective views" during training compared to the baselines. We show some qualitative results in Fig. 4(left).

**Real-world single-view inverse rendering:**    Up to this point, we have showcased the effectiveness of our approach in various setups using synthetic data. Next, we evaluate our model on real-world data. Due to the challenge of obtaining highly accurate ground truth for materials and geometry, we focus our comparison solely on the rendering results. As indicated in Tab. 3(Left), our method achieves comparable performance to the multi-view baselines, even when trained using only a single view. We visualize some results in Fig. 1, Fig. 6 and Fig. 7.

**Ablation study for each loss term:**    As shown in Tab. 2(right), since the metallicness of natural materials are usually binary, incorporating the metallic loss can properly regularize the underlying metallic component and prevent overfitting; Eikinol loss and mask loss help us better constrain the surface and the boundary of the objects, making them more accurate and smooth. Removing either term will significantly affect the reconstructed geometry and hence affect the relighting performance; The pre-trained surface normal provides a strong geomeotry prior, allowing us to reduce the ambiguity of sparse view inverse rendering. Removing it degrades the performance on all aspects.

### 4.3   Analysis

**Importance of the number of duplicate objects:**    Since our model utilize duplicate objects as a prior for 3D reasoning, one natural question to ask is: how many duplicate objects do we need? To investigate this, we render 9 synthetic scenes with 4~60 duplicates with the same object. We train our model under each setup and report the performance in Tab. 2. As expected, increasing the number of duplicates from 6 to 30 improves the accuracy of both material and geometry, since it provides more constraints on the shared object intrinsics. However, the performance decrease after 30 instances due to the accumulated 6 DoF pose error brought by heavier occlusion, the limited capacity of the visibility field, and limited image resolution.

**The influence of different geometry representation:**    To demonstrate that our method does not rely on a specific neural network architecture, we replace the vanilla MLP using Fourier position encoding to triplane representation from PET-NeuS[61] and hash position encoding from neuralangelo[35] respectively. The result (please refer to supplementary material) shows that the triplane (28.0MB) can recover better geometry and texture than our original model (15.8MB) because the explicit planes provide higher resolution features and can better capture local detail[4]. However, the model of hash positional encoding (1.37GB) and produces high frequency noisy in geometry, indicating that it overfits the training view.

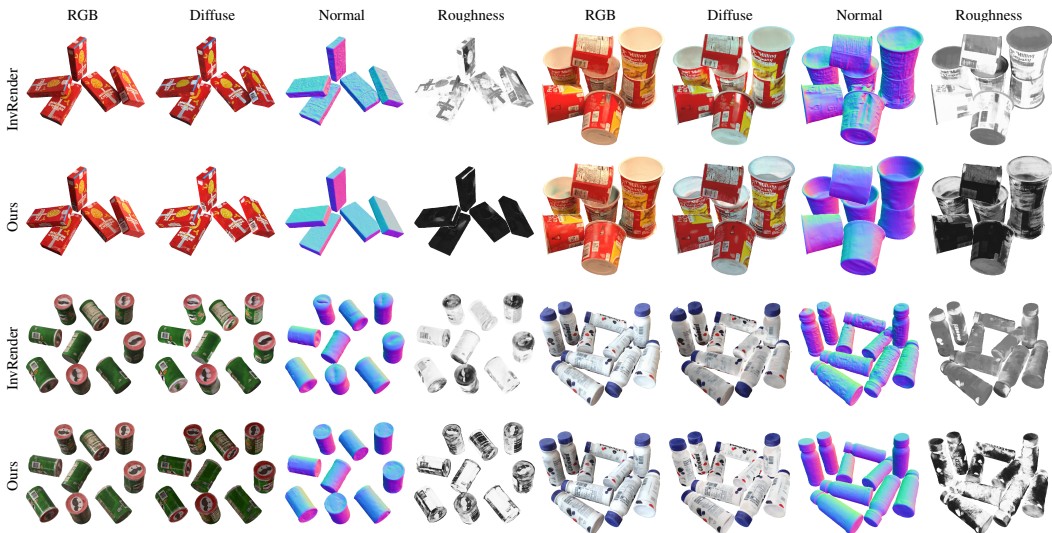

Figure 7: **Qualitative comparison on real-world data.** InvRender [74] takes as input 10 images, while we only consider one single view. Yet our approach is able to recover the underlying geometry and materials more effectively.

**Multi-view single object (M-S) vs. single-view multi-objects (S-M):** Is observing an object from multiple views equivalent to observing multiple objects from a single view? Which scenario provides more informative data? To address this question, we first construct a scene containing 10 duplicate objects. Then, we place the same object into the same scene and capture 10 multi-view images. We train our model under both setups. Remarkably, the single-view setting outperforms the multi-view setting in all aspects (see Tab. 3(right)). We conjecture this discrepancy arises from the fact that different instances of the object experience environmental lighting from various angles. Consequently, we are better able to disentangle the lighting effects from the material properties in the single-view setup. Fig. 5(right) shows some qualitative results.

**Applications:** Our approach supports various scene edits. Once we recover the material and geometry of the objects, as well as the illumination of the scene, we can faithfully relight existing objects, edit their materials, and seamlessly insert new objects into the environment as if they were originally present during the image capturing process (see Fig. 1).

**Limitations:** One major limitation of our approach is that we require the instances in each image to be nearly identical. Our method struggles when there are substantial deformations between different objects, as we adopt a geometry/material-sharing strategy. One potential way to address this is to loosen the sharing constraints and model instance-wise variations. We explore this possibility in the supplementary material. Furthermore, our model requires accurate instance segmentation masks. Additionally, our approach currently requires decent 6 DoF poses as input and keeps the poses fixed. We found that jointly optimizing the 6 DoF pose and SDF field with BARF [37] will cause the pose error to increase. This is because recovering camera pose from sparse views and changing light is a difficult non-convex problem and prone to local minima. We believe the more recent works, like CamP[48] and SPARF[58] could potentially further refine our estimations. Moreover, our model cannot effectively model/constrain the geometry of unseen regions similar to existing neural fields methods.

## 5   Conclusion

We introduce a novel inverse rendering approach for single images with duplicate objects. We exploit the repetitive structure to estimate the 6 DoF poses of the objects, and incorporate a geometry and material-sharing mechanism to enhance the performance of inverse rendering. Experiments show that our method outperforms baselines, achieving highly detailed and precise reconstructions.

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
