# Structure *from* Duplicates:
# Neural Inverse Graphics from a Pile of Objects
# —Supplementary Material—

**Tianhang Cheng[1]**   **Wei-Chiu Ma[2]**   **Kaiyu Guan[1]**   **Antonio Torralba[2]**   **Shenlong Wang[1]**
[1]University of Illinois Urbana-Champaign    [2]Massachusetts Institute of Technology
{tcheng12, kaiyug, shenlong}@illinois.edu
{weichium, torralba}@mit.edu

## 1   Novelty and contributions

How to make inverse graphics/3D reconstruction more robust and work under more extreme scenarios is a challenging and longstanding problem in computer vision. In this work, we take a step forward by exploring the potential of performing structure from motion and recovering object intrinsics and environmental extrinsics from a single image without pre-trained priors. Specifically, we focus on the scenarios where there are multiple (near-)identical objects within the scene. By carefully formulating a duality between multiple copies of an object in a single image and multiple views of a single object, we are able to resolve the ambiguities in 3D and effectively recover the properties of interest.

Over the years, the community has been actively investigating how to harness multi-view information from videos or sparse, extreme-view images, and push forward the frontier of 3D reconstruction and inverse graphics. Our work can be seen as an attempt in such a stride. To our knowledge, this is the first effort to conduct structure from motion from a single image. Furthermore, based on our preliminary experiments, our approach also has the potential to deal with slight variations, as shown in Sec. 6.1. Specifically, we test our approach on the crane image that [14] provided, where each instance is slightly different. By augmenting our geometry backbone with a instance-specific deformation field, we are able to reconstruct reasonable poses and recover sensible shape and material. We hope it can shed light on future research along similar directions, such as handling articulate objects or objects with large deformation.

## 2   Additional details of proposed pipeline

### 2.1   The In-plane augmentation of pose estimation for the single image

Vanilla COLMAP often fails to reconstruct the duplicated instances of a single image. The reason for the failure can probably be explained by Fig. 7: when the lighting effect and occlusion are disregarded, a single-view image containing duplicated objects can be treated as observing a single object from multiple viewpoints using a multi-view camera setup. This conversion results in numerous accidental and non-uniform multi-view images with varying orientations that may not align uniformly with the upward axis. Therefore, vanilla COLMAP cannot find stable matching points from these extreme camera distribution. By contrast, with the help of in-plane rotation augmentation, we can greatly improve the performance (e.g., from failure to success). This demonstrate that the incorporation of in-plane rotation augmentation becomes essential to facilitate robust point matching, as demonstrated in Figure 8. However, there are still limitations of this pose estimation module. First, for low-texture objects or low-resolution scenes where pixels are not distinct, our method may still suffer, like the 60 boxes in Fig. 3. Second, the time complexity of the algorithm scales with the square of the number of objects, so it will be slow when there are many instances. We believe that combining some transformation-invariant feature extractor (such as GIFT[4]) can solve this problem faster.

37th Conference on Neural Information Processing Systems (NeurIPS 2023).

For synthetic dataset we use 3200×3200 resolution for pose estimation and resize to 800×800 resolution for training. For real-world dataset we use the 3072×3072 resolution for pose estimation and resize to 800×800 resolution for training.

For $n$ instances in the scene, we first find relative rotation angles for all $\frac{n(n-1)}{2}$ instance pairs. For each pair, we fix one of them and gradually rotate the other, increasing the angle by 4 degrees each time. We record the number of matching points $n$ at different rotation angles $\theta$, i.e. $n(\theta)$, and mark those rotation angles with a greater number of matching points than the average as "good angles", i.e.

$$\boldsymbol{\theta_{good}} \in \left\{ \theta | \theta > \sum_{i=0}^{N} \frac{n(\theta_i)}{N} \right\}, \text{ where } N = \frac{360}{d\theta} = 90.$$

However, the relative rotation angles $\boldsymbol{\theta}_{good}$ of one pair is usually conflict with other pairs. To resolve this issue, We use a customized procedure to transform multiple relative pairwise poses into an initialized global rotation angle for each instance. Specifically, we use Scipy's BFGS optimizer to find a global rotation angle that minimizes the loss. Transferring relative rotation angles into global ones ensures the integration into the standard BA pipeline. After we correct each instance with the global rotation, we use Superglue[7] and Superpoint[1] to extract and match key points. Then we apply a standard bundle adjustment algorithm to solve the 6Dof pose of each instances. The algorithm is as follows:

**Require:** $\boldsymbol{\theta}_{good}^i$, where $i \in [0, \frac{n(n-1)}{2} - 1]$ (good relative rotation angles of each pair)
    $L \leftarrow \inf$ (Initialize loss)
    $\boldsymbol{\theta}_{global}^j \leftarrow U[0, 2\pi]$, where $j \in [0, n-1]$ (uniform initialized global rotation angle for each pair)
    **while** $L$ not converge **do**
        $\theta_{rel}^{p,q} \leftarrow |\theta_{global}^p - \theta_{global}^q| \bmod 2\pi$, where $p, q \in [0, n-1]$
        $L_{p,q} \leftarrow \min\left(\left|\theta_{rel}^{p,q} - \boldsymbol{\theta}_{good}^p\right|\right) + \min\left(\left|\theta_{rel}^{p,q} - \boldsymbol{\theta}_{good}^q\right|\right)$
        $L \leftarrow \sum_{p,q} L_{p,q}$
    **end while**
    **Return** $\theta_{global}^j$ (resulting global rotations)

## 2.2  Training

We train 100000 iterations for geometry stage with 1e-4 learning rate. The visibility stage takes 3000 iterations with 2e-5 learning rate. The material stage takes 10000 iterations with 2e-4 learning rate. Please check our code for more detail. This supplementary material provides further details on our method and presents an extended set of experimental results.

# 3   Comparison with recent works

## 3.1  "Seeing a Rose in Five Thousand Ways"[14]

The setup of the two papers are similar, but they are different in the following aspects:

- Assumptions: While [14] is able to model the variations among the instances, they impose other strong assumptions such as knowing the camera distribution in advance. The strong camera assumption allows them to sidestep the pose estimation step (i.e., SfM) and focus on modeling the variation. In contrast, we assume no knowledge about the poses and attempt to solve for the full inverse rendering pipeline from the beginning. We thus resort to the (near-)identical instances to recover the exact 6 DoF poses.

- Approaches: [14] tackle the task through generative modeling. Since they need to train a generative model per scene, their approach is very data-hungry. In contrast, our approach mainly exploits multi-view geometry to recover the underlying intrinsic and extrinsic properties. By explicitly baking the constraints into the modeling procedure, our approach becomes much more data-efficient. To validate our conjecture, we train [14] on three randomly selected scenes from our dataset, each of which has 10 identical instances. As shown in the pdf, the generative model failed to recover either of them. For comparison, we also test our approach on the crane image that [14] provided (the only publicly available data), where each instance is slightly different.

By augmenting our geomtry backbone with a instance-specific deformation field, we are able to reconstruct resonable poses and recover sensible shape and material.

- Extrinsics: [14] assume a simple phong shading model and assume a dominant directional light, whereas we parameterize our materials with PBR materials and the lighting with enironmental map, allowing us to model complex real world scenarios more effectively. Finally, it is unclear how to extend [14] to multi-view setup. In contrast, our method is naturally compatible with multi-view observations.

## 3.2 "Modeling Indirect Illumination for Inverse Rendering"[13]

It is important to note that our model does not rely on a specific geometric model and it can be replaced with more advanced neural representations. In this paper, we build our inverse rendering pipeline from Invrender[13]. But there are some differences in execution:

- Backbone: In our approach, we utilize NeuS[9] as our neural surface model instead of IDR[11]; and for the visibility field, we opt for Siren[8] instead of ReLU.
- Metallic: Our model can reconstruct metallic object besides pure diffuse object.
- MLP distillation: We distill the geometry MLP into a smaller one for fast classification.
- Self-occlusion and Inter-occlusion: Since we have multiple instances in our setup, it is essential to model both inter-object self-casted shadows and inter-object occlusions. Our model goes beyond simple object-centric representation.

## 4 Additional details of the dataset

### 4.1 Synthetic dataset

Our new dataset **Dup** consists of 13 synthetic scenes. "Apple", "Medicine box", "Can" and "Driller" consists of 100 training views and 200 testing views for multi-view experiments. "Color box", "Cash machine", "Cleaner", "Clock", "Coffee machine", "Fire extinguisher", "Wood guitar", "Warning sign" and "Food tin" consists of 7-10 multi-view images to test our model on single-view and baselines for multi-view. The resolution of the raw images are $3200 \times 3200$.

### 4.2 Real-world dataset

We scatter object on the table and use mobile phone to gather several scenes, named "Toy airplane", "Cake box", "Cheese box", "Cola", "Potato chips" and "Yogurt". The number of objects in the scene ranges from 5 to 10. The resolution of the raw images are $3072 \times 3072$.

## 5 Additional details or analysis of experiments

### 5.1 The influence of different number of instances

We conducted experiment on the image of "box". The training image are visualized in Fig. 3. Please refer to the paper for quantitative result. The results show that there is a "sweet spot" for the box dataset that achieves the best trade-off between image resolution and number of views. We believe that this "sweet spot" exists for other data sets as well. For objects with simple textures and complex geometric shapes, a smaller number of instances should be processed, otherwise there will be large errors in pose estimation. On the contrary, for objects with simple shapes and complex textures, the number of instances can be increased to reduce the ambiguity of material recovery.

### 5.2 The influence of different neural representation

We conducted experiment on the image of "Cash machine". The triplane representation is adapted from PET-NeuS[10] and hash representation is adapted from Neuralangelo[3].

- Triplane: The triplane representation is consist of three planes, each plane is of $512 \times 512 \times 32$ resolution. The triplane will passed to a self-attention convolution module to produce features with

different frequency bands. Then a 3D point will sample space features from these triplane and decode into SDF value and color with a small MLP.

- Hash position encoding: The point is encoded by hash function with the default setting in Neuralangelo[3]. Then the hash feature is passed to MLP layers and decode to SDF and color values. Here We use analytical gradient instead of numerical gradient in the default setting because we found the former will produce less high frequency noise in our tasks.

We show the visual image in Fig.11 and it demonstrate that the triplane has the best performance. Please refer to the paper for the quantitative result.

### 5.3 Multi-view single-object vs single-view multi-object

To ensure a fair comparison, we maintain the multi-view single-object (M-S) setting, while adjusting the single-view multi-object (S-M) setting to have a similar number of non-empty pixels. The training images in the M-S setting (see Fig. 12, first two rows) contain 244,335 non-empty pixels, whereas the training image in the S-M setting (see Fig. 12, bottom row) consists of 263,910 non-empty pixels. We provide a qualitative comparison in Fig. 13 and present the corresponding quantitative analysis in Table 4.

### 5.4 The contribution of each loss term

We experiment the contribution of each loss term on the image of "fire-extinguisher". We train 50000 iterations for geometry reconstruction, 2000 for visibility fields and 13000 iterations for material recover (less iterations than the main paper) for faster verification. The results show that our full model has the best albedo and relighting results. Its rendering performance is only surpassed by the ablation model without the metallic binary loss term.

The result table indicates that our loss function is sensible. First, The metallic of natural materials is mostly binary. However the "w/o metal loss" does not limit the metallic proprieties so it may has a stronger fitting ability, but there is also a risk of over-fitting. Second, the same as [13], the latent smooth loss term in texture-MLP can reduce the possibility of over fitting because the materials of real world objects are limited. Third, the eik loss and mask loss proposed by NeuS[9] can constrain the surface and boundary of the geometry, making the surface of the object more accurate and smooth. Fourth, the pre-trained surface normal can provide a strong geomeotry prior, reduce the ambiguity of sparse view inverse rendering.

### 5.5 The influence of noisy instance segmentation image

In the main paper, we use ground-truth segmentation mask for synthetic dataset and use an use interactive pre-trained segmentation models for real-world dataset. To assess the influence of segmentation map noise, we conducted a comparative analysis of the model's performance on the "fire extinguisher" image. We compared results obtained using ground truth segmentation maps(left image of Fig. 9) with those generated by pre-trained models through segmentation maps generated with 2 to 4 clicks per instance, without subsequent post-processing (right image of Fig. 9).

The result in second row of Fig. 10 shows that the our model has certain robustness to the noise of segmentation.

## 6 Additional experiments

### 6.1 Single-view reconstruction on "paper crane" dataset from Zhang *et al*.[14]

To demonstrate the difference between our method and [14], we test their on our single-image dataset and versa versa. We run their method on the image of "airplane", "cake box" and "cola", which contains 6, 7, 7 instances respectively. The result in Fig. 4 shows that their methods fail to reconstruct a good geometry and texture and generate over-smooth output. This is because they suppose a pose prior rather than accurate camera pose and don't support zero-variant scenes, thus cannot accurately capture complex materials and geometry. In addition, it will produce large errors when objects have mutual occlusion.

We also run our method on crane dataset proposed by [14] in their Github repository. We apply SfM with in-plane rotation, which gives us better registration (Fig. 8). To model the geometry variances of instances, we also adopt an deformable-aware pipeline by inserting a geometry latent into the middle of geometry MLP. Our method can reconstruct the 6DoF pose, geometry and PBR texture of these cranes. We visualize the recoverd texture in Fig. 1 and the surface normal of different instances in Fig 2. The difference from [14] is that our latent vector can correspond to the instances in the original picture one by one, but what they learn is a geometry distribution and need "latent inversion"(similar to GAN inversion) to calculate the latent for a specific instance.

However, as shown in 1, our model does not recover a fully consistent texture at corresponding points across different instances. This is because we model the variance of instances by implicit instance vector rather than the explicit displacement field. The displacement field based neural representation, such as Nvdiffrec-MC[2] and D-NeRF[6], can achieve strict consistency between different time or instances. Nevertheless, the displacement field usually has a greater number of parameters than implicit instance vector, which may lead to overfitting. We leave this for future study.

## 6.2 The influence of inaccurate pose

Since our method adopts a stage-wise inference procedure, errors in pose estimation can propagate and impact the quality of the inverse rendering reconstructions. To verify the extent of this impact, we conduct an oracle experiment where we replace the estimated 6 DoF object poses with ground truth. The results, presented in Table 5, demonstrate that our model achieves performance similar to that of oracles, primarily due to its precise estimation of small object poses. However, when faced with samples involving significant pose estimation errors for objects, the performance of the oracle outperforms our model, as shown in Table 6.

We jointly optimize the geometry and camera pose, achieving similar results(**??**) to the original pipeline. Optimizing camera poses under sparse-views and varying lighting conditions presents a notably ambiguous challenge. Despite the alterations in the pose of each instance, the average rotation and translation errors of the final 6Dof pose for the model have shown no reduction.

# 7 Additional visualizations

## 7.1 Synthetic multi-view experiment

The qualitative results are shown in Fig. 15, 16, 17, 18. The quantitative results are shown in table. 1.

## 7.2 Synthetic single-view experiment

The qualitative results are shown in Fig. 19, 20, 21, 22, 23, 24, 25, 26, 27. The quantitative results are shown in table. 2. The recovered bounding boxes are shown in Fig 6.

## 7.3 Real-world single-view experiment

The qualitative results are shown in Fig. 28, 29, 30, 31, 32, 33. The quantitative results are shown in table. 3. The recovered bounding boxes are shown in Fig 5.

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

| | Rendering | | | Albedo | | | Roughness | Relighting | | | Env Light | Geometry |
|---|---|---|---|---|---|---|---|---|---|---|---|---|
| | PSNR ↑ | SSIM ↑ | LPIPS ↓ | PSNR ↑ | SSIM ↑ | LPIPS ↓ | MSE ↓ | PSNR ↑ | SSIM ↑ | LIPIPS ↓ | MSE ↓ | CD ↓ |
| PhySG | 25.985 | 0.809 | 0.199 | 16.233 | 0.620 | 0.363 | 0.087 | 21.323 | 0.748 | 0.270 | 0.054 | 0.024 |
| Nv-DiffRec | **27.840** | **0.886** | **0.089** | 16.123 | 0.533 | 0.412 | 0.116 | 17.418 | 0.459 | 0.388 | 0.168 | 0.268 |
| InvRender | 26.452 | 0.809 | 0.206 | 16.984 | 0.637 | 0.370 | 0.084 | 22.224 | 0.757 | 0.267 | 0.067 | 0.024 |
| Ours | 23.213 | 0.781 | 0.222 | **21.961** | **0.655** | **0.260** | **0.026** | **25.486** | **0.830** | **0.183** | **0.029** | **0.011** |

Table 1: **Synthetic multi-view result**. All models are trained with 100 multi-view images. Our model has the best texture recovery performance. Nv-DiffRec reaches the best rendering result in training image, but has the worst texture recovery due to overfitting.

| | Rendering | | | Albedo | | | Roughness | Relighting | | | Env Light | Geometry |
|---|---|---|---|---|---|---|---|---|---|---|---|---|
| | PSNR ↑ | SSIM ↑ | LPIPS ↓ | PSNR ↑ | SSIM ↑ | LPIPS ↓ | MSE ↓ | PSNR ↑ | SSIM ↑ | LIPIPS ↓ | MSE ↓ | CD ↓ |
| PhySG | 20.047 | 0.584 | 0.323 | 14.977 | 0.460 | 0.405 | 0.255 | 18.504 | 0.554 | 0.356 | 0.082 | 0.033 |
| Nv-DiffRec | 20.513 | 0.638 | 0.248 | 14.021 | 0.416 | 0.431 | 0.165 | 17.214 | 0.427 | 0.391 | 0.067 | 0.050 |
| InvRender | 19.489 | 0.557 | 0.351 | 14.724 | 0.438 | 0.431 | 0.247 | 17.998 | 0.527 | 0.381 | 0.082 | **0.033** |
| Ours | **24.307** | **0.752** | **0.152** | **17.629** | **0.594** | **0.229** | **0.062** | **21.374** | **0.695** | **0.189** | 0.052 | 0.034 |

Table 2: **Synthetic single view result**. The baseline models are trained with 10 multiview images and our model is trained in single image. Our model has the best texture recovery performance.

| | Rendering | | |
|---|---|---|---|
| | PSNR ↑ | SSIM ↑ | LPIPS ↓ |
| PhySG | **20.624** | 0.641 | 0.263 |
| Nv-DiffRec | 18.818 | 0.569 | 0.282 |
| InvRender | 20.665 | 0.639 | 0.262 |
| Ours | 20.326 | **0.660** | **0.192** |

Table 3: **Experiment result on real-world single-view dataset**. Our model has a comparable quality even with a single-view.

| | Rendering | | | Albedo | | | Roughness | Relighting | | | Env Light | Geometry |
|---|---|---|---|---|---|---|---|---|---|---|---|---|
| | PSNR ↑ | SSIM ↑ | LPIPS ↓ | PSNR ↑ | SSIM ↑ | LPIPS ↓ | MSE ↓ | PSNR ↑ | SSIM ↑ | LIPIPS ↓ | MSE ↓ | CD ↓ |
| M-S | 21.347 | 0.591 | 0.511 | 20.229 | 0.594 | 0.514 | 0.096 | 21.328 | 0.600 | 0.494 | **0.045** | 0.010 |
| S-M (ours) | **23.994** | **0.657** | **0.375** | **23.448** | **0.666** | **0.365** | **0.050** | **24.254** | **0.667** | **0.359** | 0.0519954 | **0.007** |

Table 4: **Quantitative results of M-S setting and S-M setting**. When the #instance × #views is a constant and with good pose estimation, our model has better performance than the traditional multi-view single object setting.

| | Rendering | | | Albedo | | | Roughness | Relighting | | | Env Light | Geometry |
|---|---|---|---|---|---|---|---|---|---|---|---|---|
| | PSNR ↑ | SSIM ↑ | LPIPS ↓ | PSNR ↑ | SSIM ↑ | LPIPS ↓ | MSE ↓ | PSNR ↑ | SSIM ↑ | LIPIPS ↓ | MSE ↓ | CD ↓ |
| Oracle | **24.570** | **0.782** | **0.128** | **17.858** | **0.597** | **0.223** | 0.105 | 21.132 | **0.709** | **0.174** | 0.063 | **0.031** |
| Ours | 24.307 | 0.752 | 0.152 | 17.629 | 0.594 | 0.229 | **0.062** | **21.374** | 0.695 | 0.189 | **0.051** | 0.034 |

Table 5: **Ablation result for ground-truth 6Dof pose (oracle model)**.

| Sample | Model | dR (°) ↓ | dT (°) ↓ | Rendering | | | Albedo | Roughness | Env Light | | Geometry | | |
|---|---|---|---|---|---|---|---|---|---|---|---|---|---|
| | | | | PSNR ↑ | SSIM ↑ | LPIPS ↓ | PSNR ↑ | MSE ↓ | MSE ↓ | CD ↓ | Precision ↑ | Recall ↑ | F1 ↑ |
| Cleaner | Oracle | - | - | **26.600** | **0.903** | **0.055** | **22.210** | **0.033** | **0.028** | **0.008** | **1.000** | **0.984** | **0.992** |
| | Ours | 1.344 | 3.067 | 25.059 | 0.844 | 0.105 | 21.647 | 0.042 | 0.033 | 0.012 | 0.986 | 0.967 | 0.976 |
| Gitar | Oracle | - | - | **25.966** | **0.809** | **0.132** | **20.608** | **0.057** | 0.049 | **0.018** | 0.954 | **0.923** | **0.938** |
| | Ours | 1.076 | 1.653 | 24.599 | 0.736 | 0.172 | 19.516 | 0.063 | **0.034** | 0.046 | **0.984** | 0.513 | 0.675 |
| Coffee | Oracle | - | - | 22.266 | **0.747** | **0.141** | **13.419** | 0.286 | 0.128 | 0.040 | 0.649 | 0.587 | 0.617 |
| | Ours | 0.589 | 1.015 | **22.477** | 0.711 | 0.166 | 13.347 | **0.064** | **0.057** | **0.039** | **0.732** | **0.610** | **0.665** |

Table 6: **The influence of inaccurate pose**. Oracle model perform much better than our model when we have a large pose estimation error.

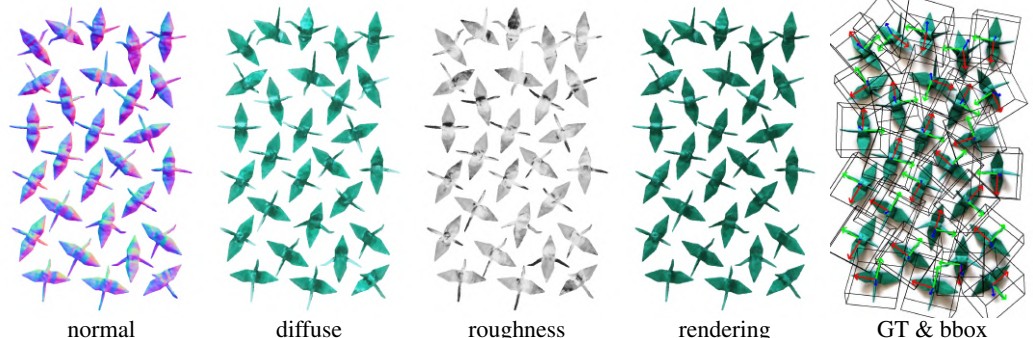

| normal | diffuse | roughness | rendering | GT & bbox |

Figure 1: **Qualitative result of our method on the *Crane* image.** We manually flipped the incorrect global rotation for some cranes before extract the final matching points to reduce the impact of symmetry. The result shows that our model can recover the geometry, texture, and bounding box from a single image, even with objects with variations in shape and appearance. However, the PBR texture is not fully consistent across different instances since we use instant vector rather than learning a displacement field. We believe that mesh based representation, like Nvdiffrec-MC [2] can achieve better consistency since it modeling the scene by displacement field.

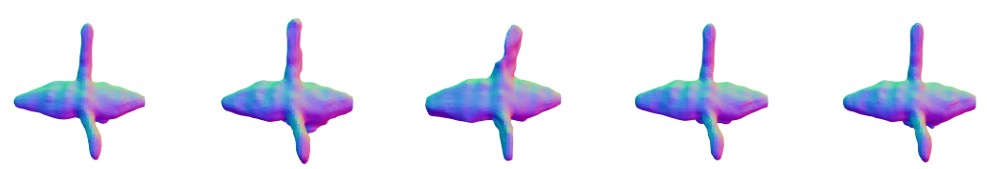

Figure 2: **Different instances of our method on paper-crane dataset.** We randomly visualize the surface normal of 5 instances.

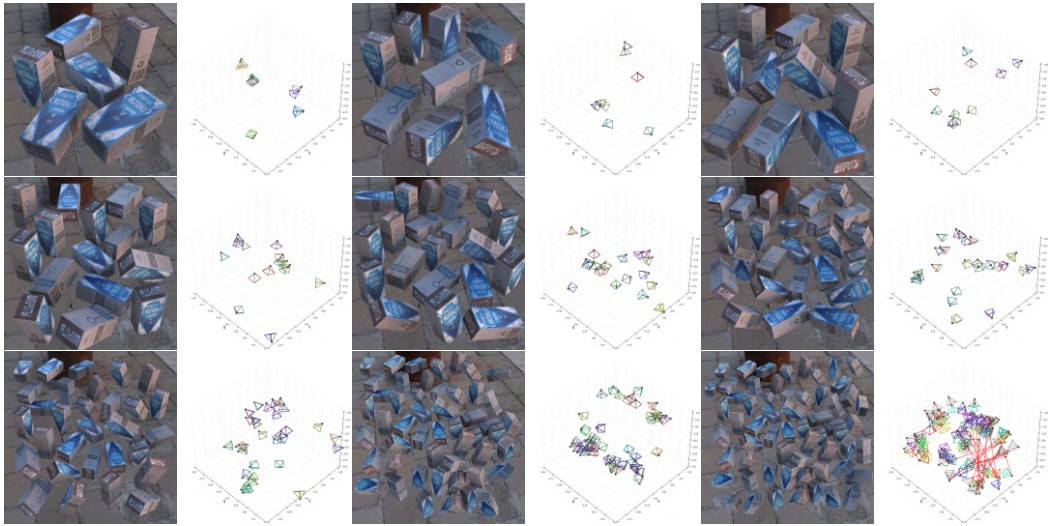

Figure 3: **Training images for different number of duplicated objects and the corresponding 6 DoF pose error.** The black camera represent the ground-truth and the colorful cameras are estimation. There are 6,8,10,15,20,25,30,50,60 instances in the scene. Our method has a large pose estimation errors for 60 boxes in 3200×3200 resolution due to the lower resolution of each instance.

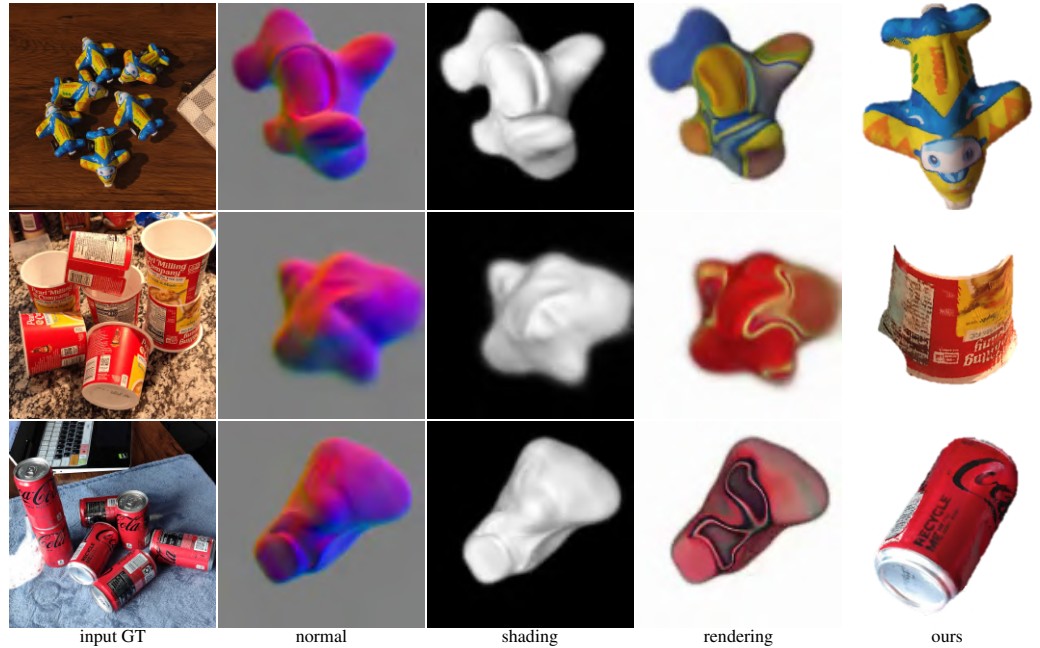

input GT        normal        shading        rendering        ours

Figure 4: **Qualitative results of [13] on our single-image dataset.** When there are fewer instances, their generative approach produces only a blurred texture and imprecise geometry on our single-image datasets. In contrast, our method (as shown in the 5th columns) accurately reconstructs the objects.

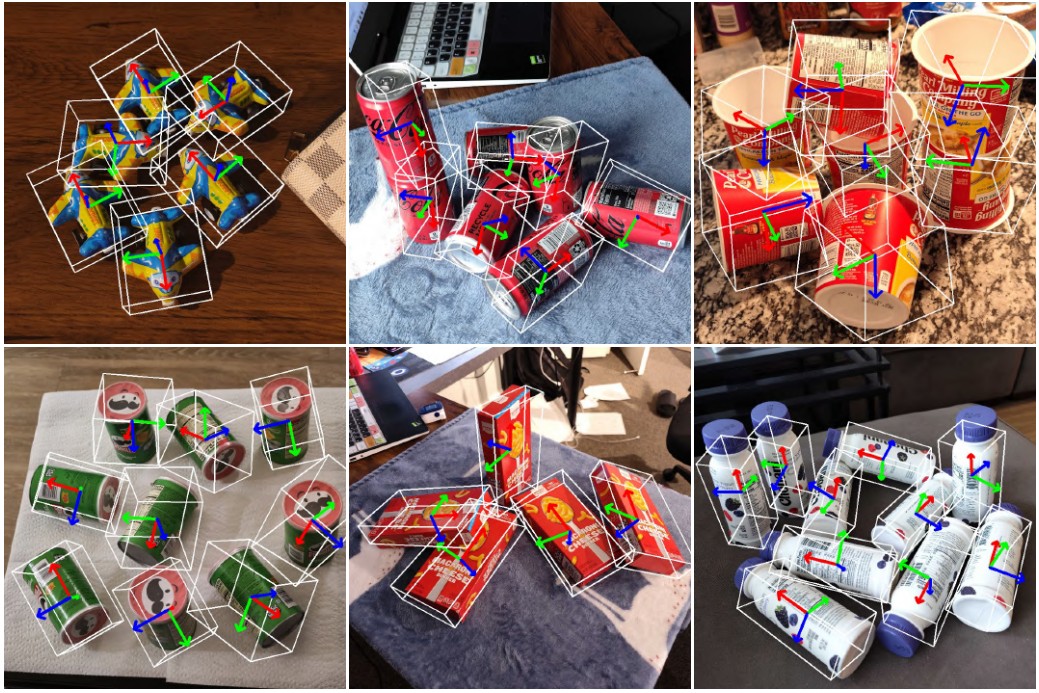

Figure 5: **The bounding box of real-world dataset.** The bounding boxes does not fully overlapped with each object because the bounding boxes are plotted according to the SfM points clouds, which does not fully cover object's surface.

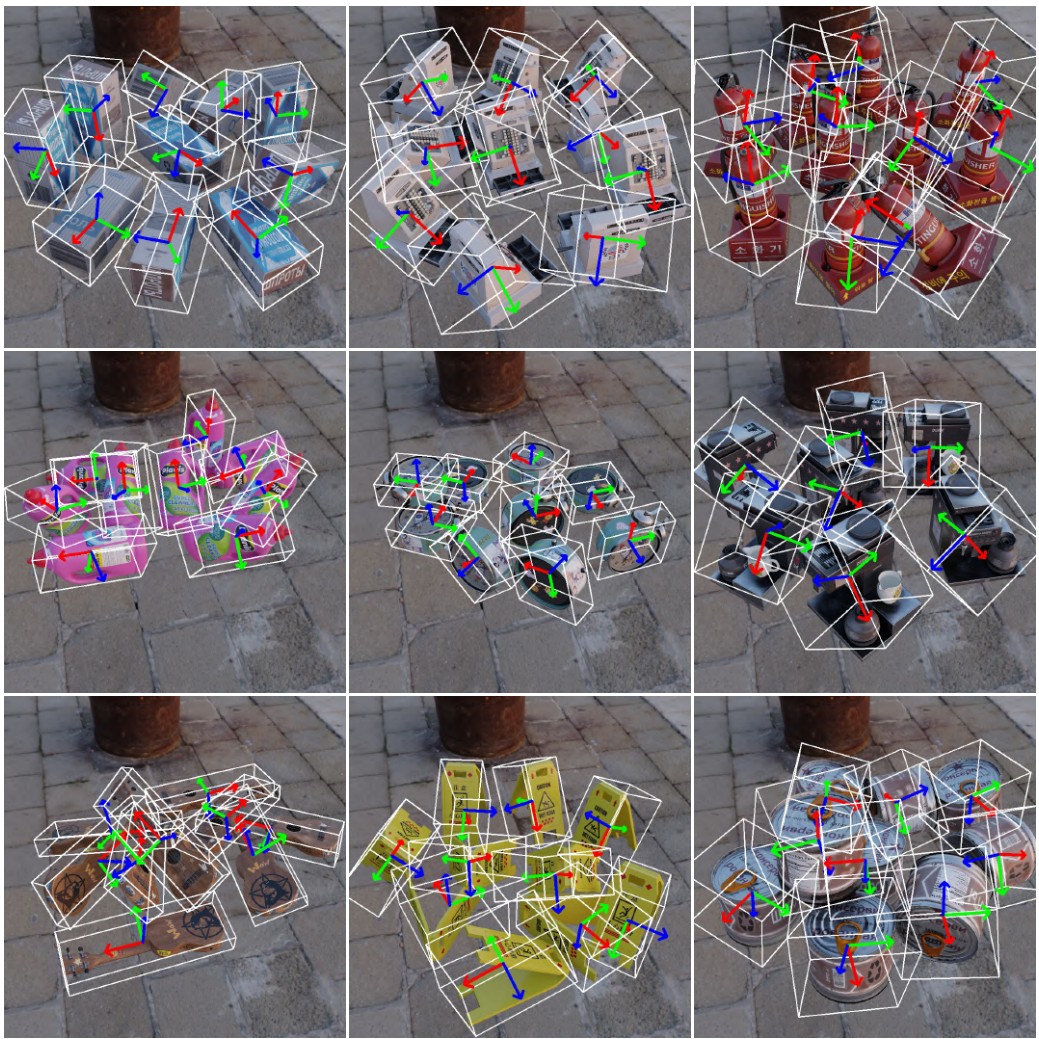

Figure 6: **The bounding box of real-world dataset.** The bounding boxes does not fully overlapped with each object because the bounding boxes are plotted according to the SfM points clouds, which does not fully cover object's surface.

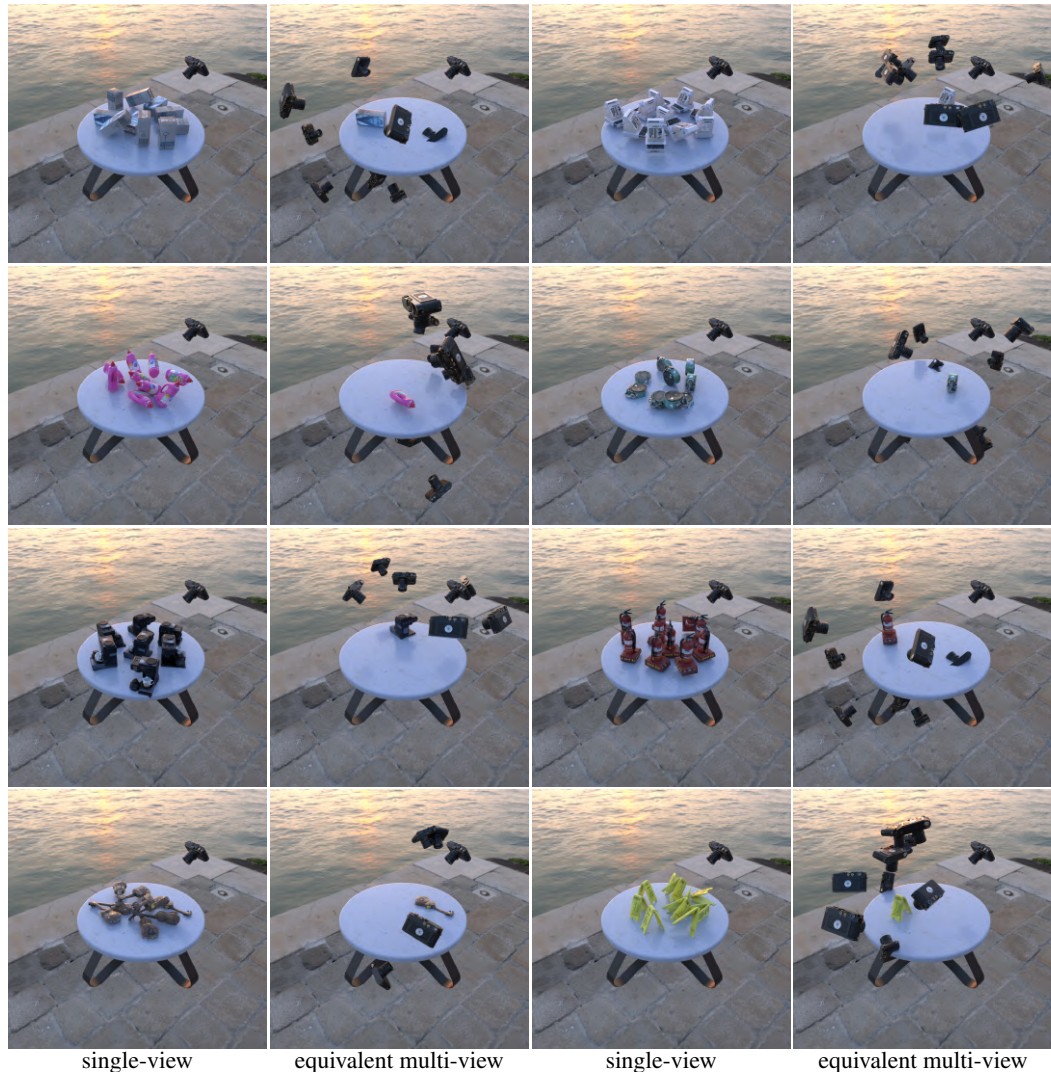

| single-view | equivalent multi-view | single-view | equivalent multi-view |

Figure 7: Without considering the lighting effect and occlusion, a single-view image with duplicated objects (left) is equivalent to use multi-view camera to observe a single object (right).

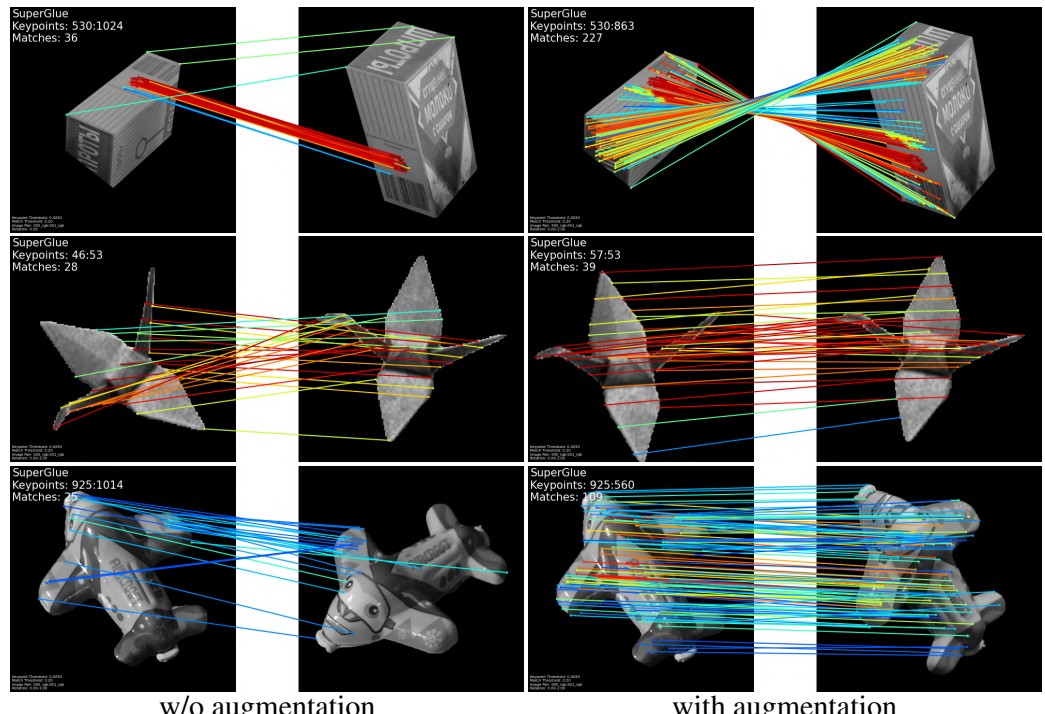

w/o augmentation                    with augmentation

Figure 8: After in-plane augmentation, the pre-trained Super-point[1] and Super-glue[7] model can generate more matching points between two instances.

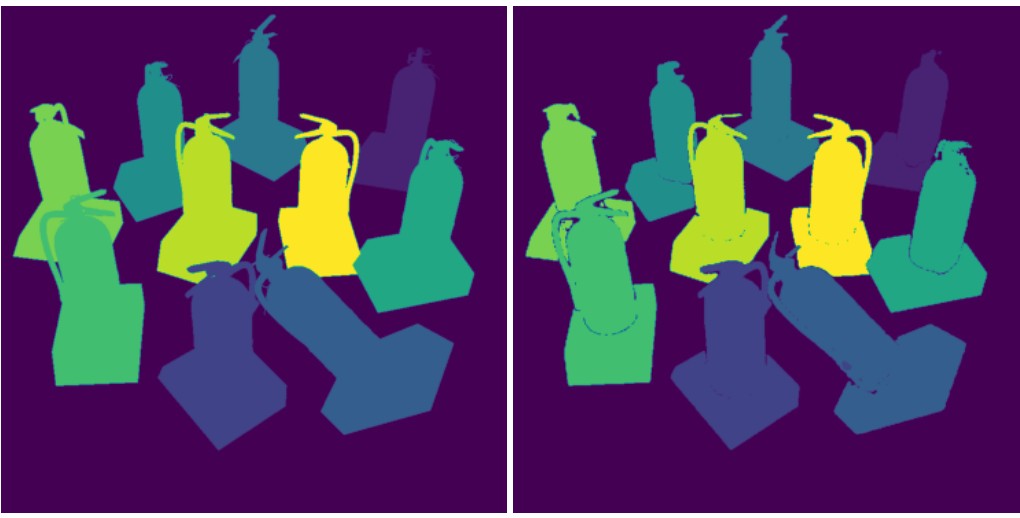

Figure 9: **Clean segmentation map(Left) and noisy segmentation map(right).** The right segmentation map is generated by pre-trained model and without post-processing.

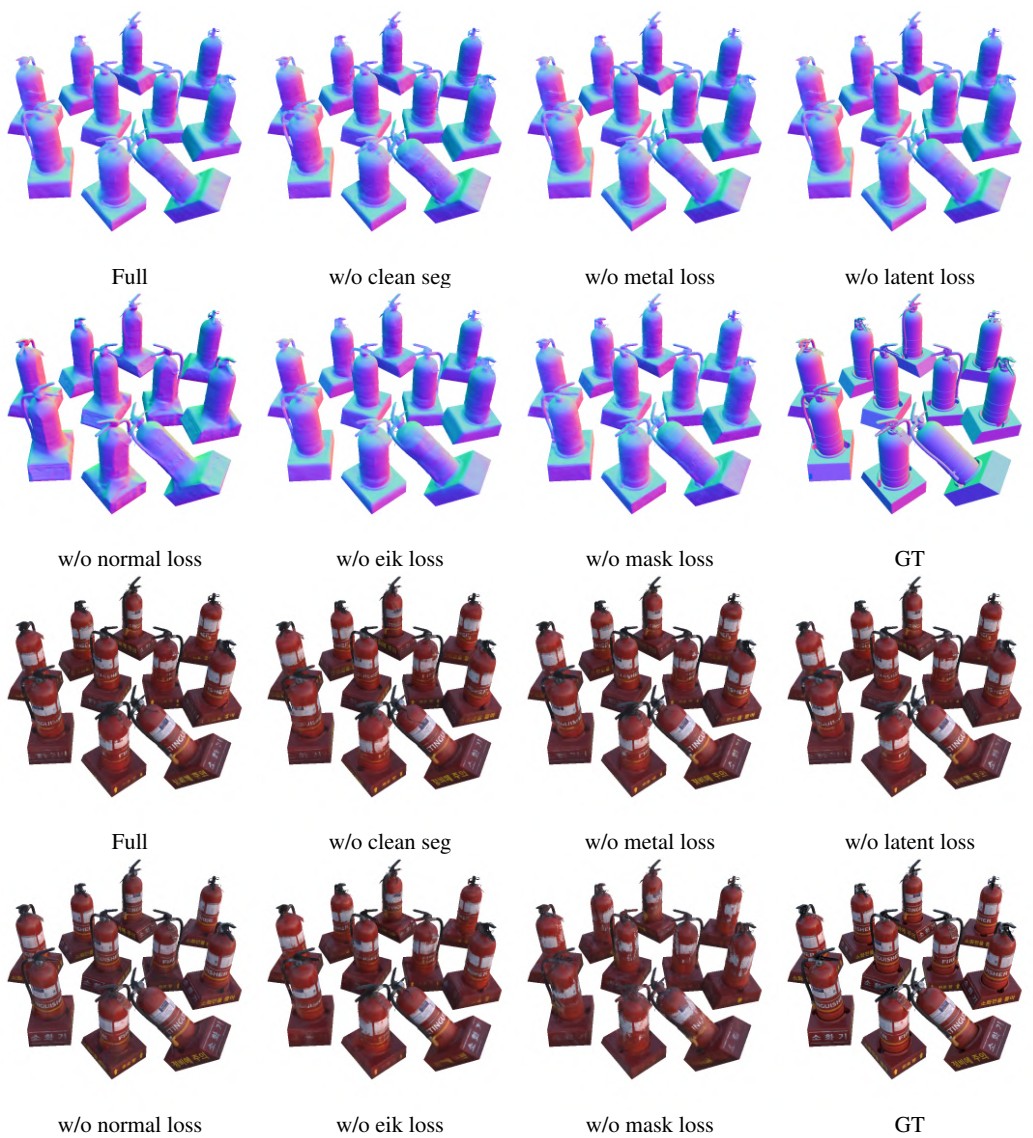

Figure 10: **Ablation for different loss term.** We evaluate the contribution of each loss term or input noise to our model. Our full most reaches the best result on most metrics

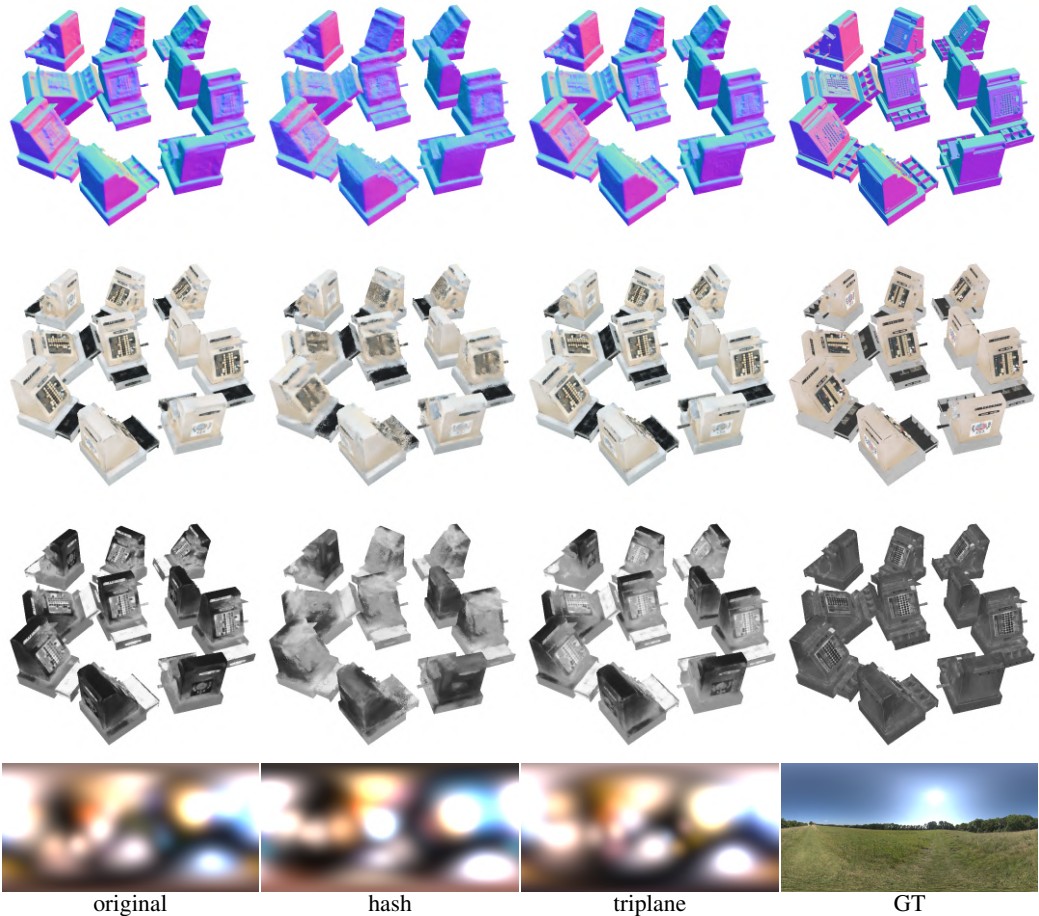

| original | hash | triplane | GT |

Figure 11: **Different neural representation.** The triplane representation (adapted from [10]) has better performance than our naive MLP representation, while hash position encoding (adapted from [3]) has worse performance due to overfitting.

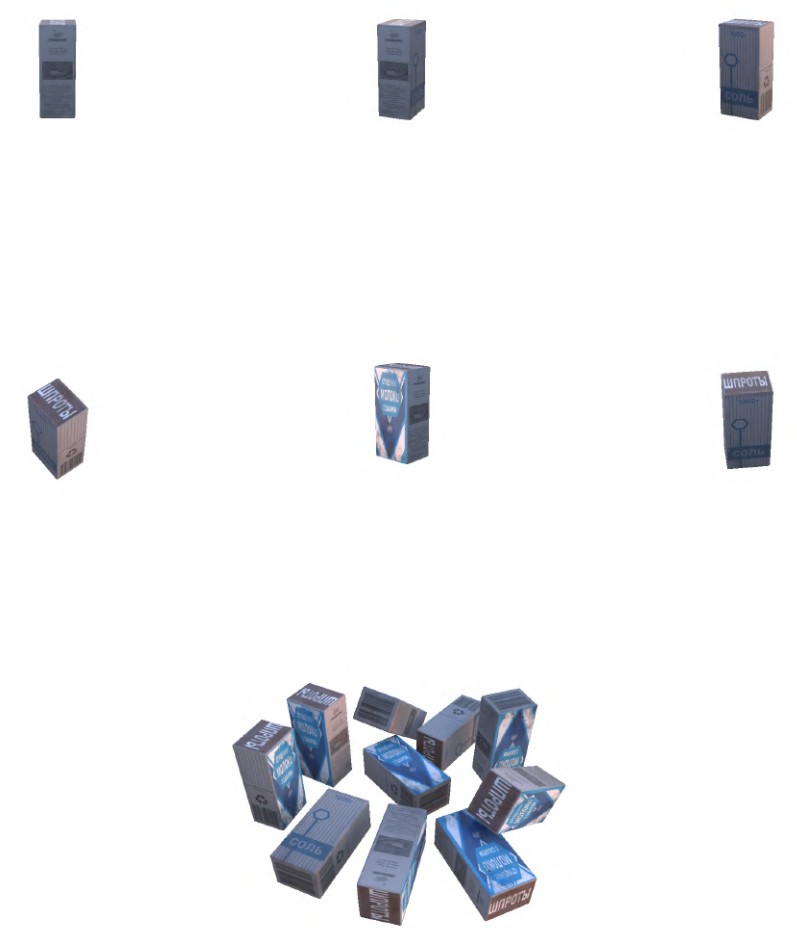

Figure 12: **First two rows**: Training images for multi-view single object (M-S), there are 10 in total, only show 6 here. **Last row**: Training image for single-view multi-object (S-M).

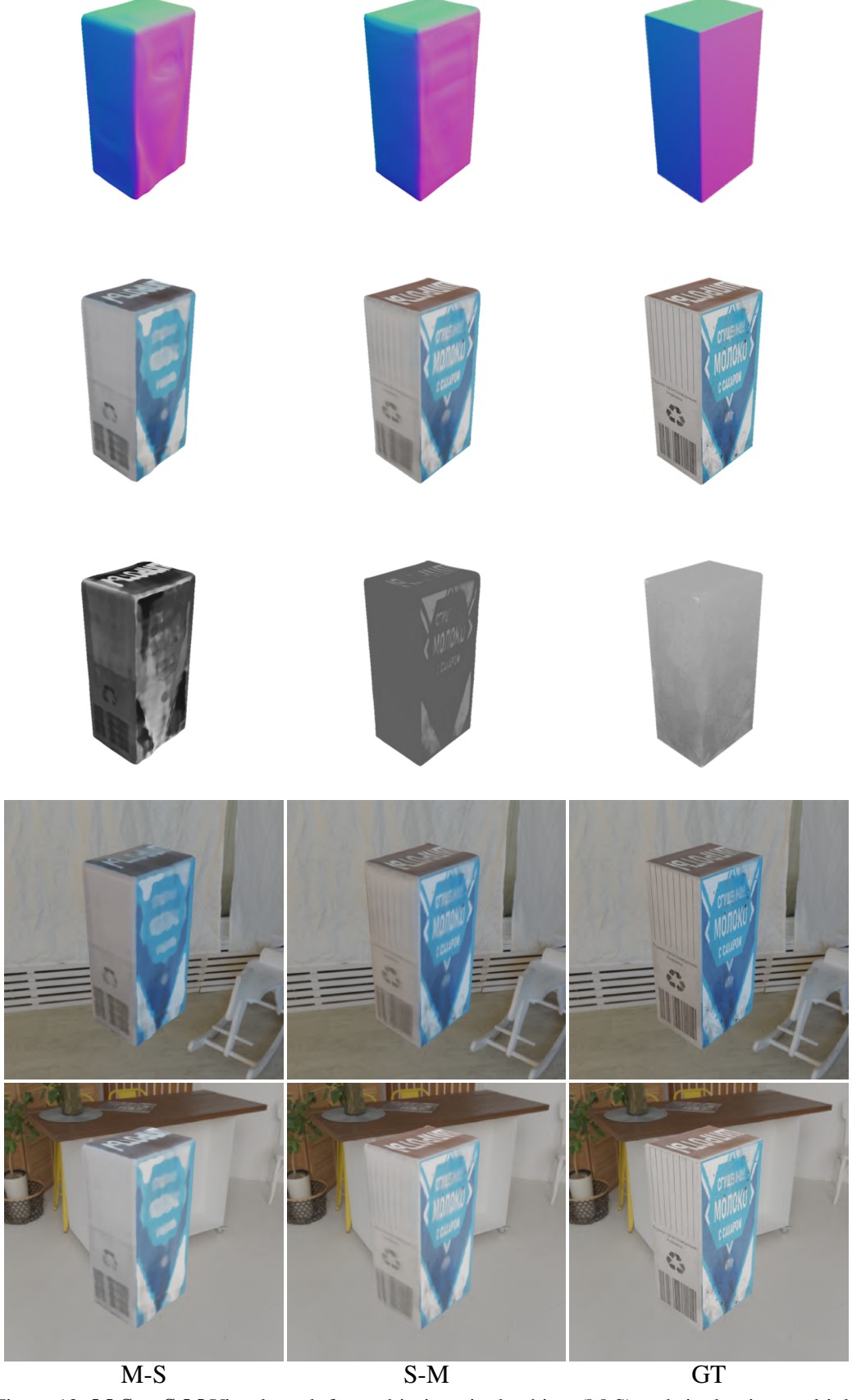

M-S                    S-M                    GT

Figure 13: **M-S vs S-M** Visual result for multi-view single object (M-S) and single-view multiple objects (S-M).

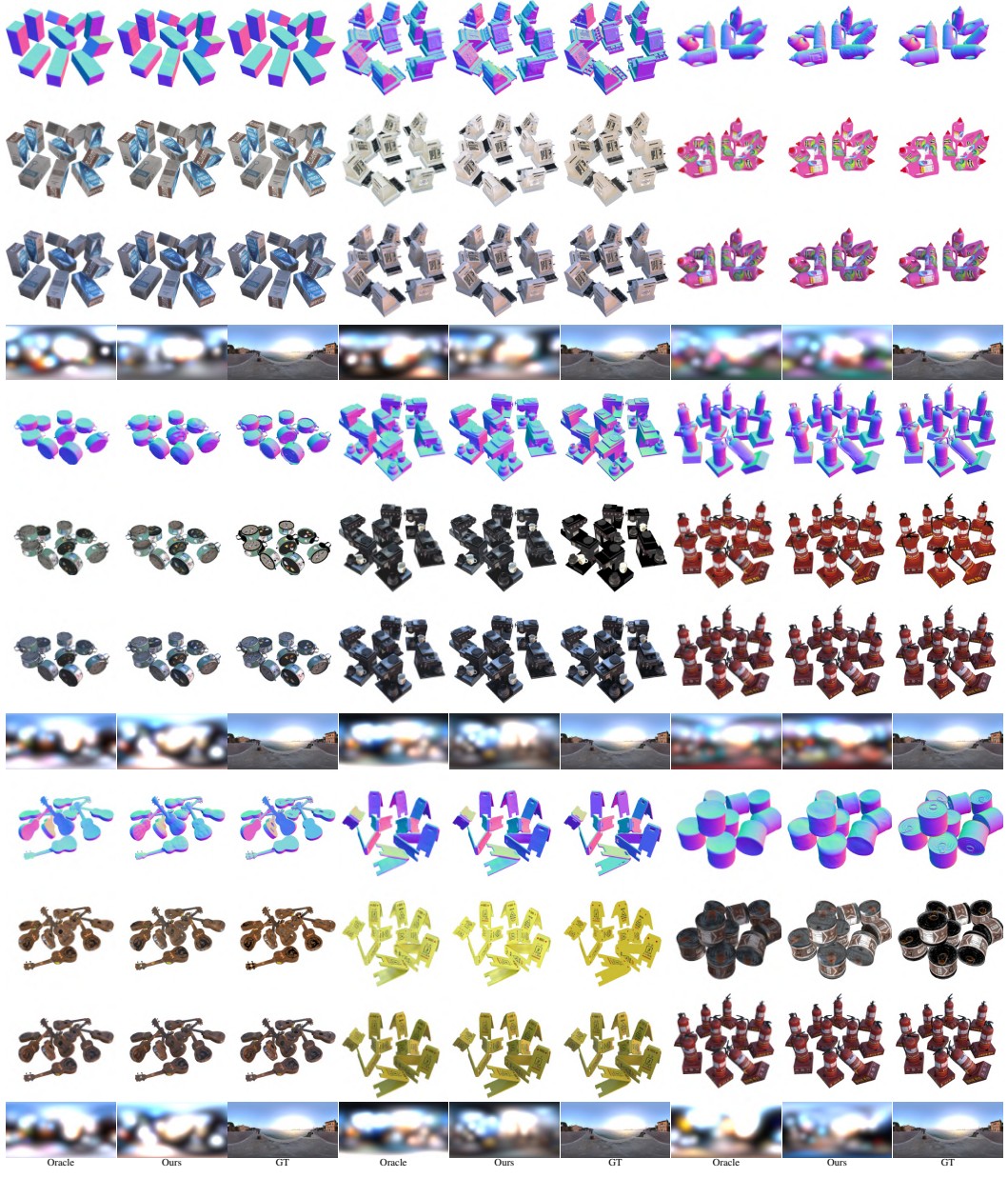

Oracle      Ours      GT      Oracle      Ours      GT      Oracle      Ours      GT

Figure 14: **Use ground-truth pose instead of SfM-derived pose.** Our model get similar results as orcale model.

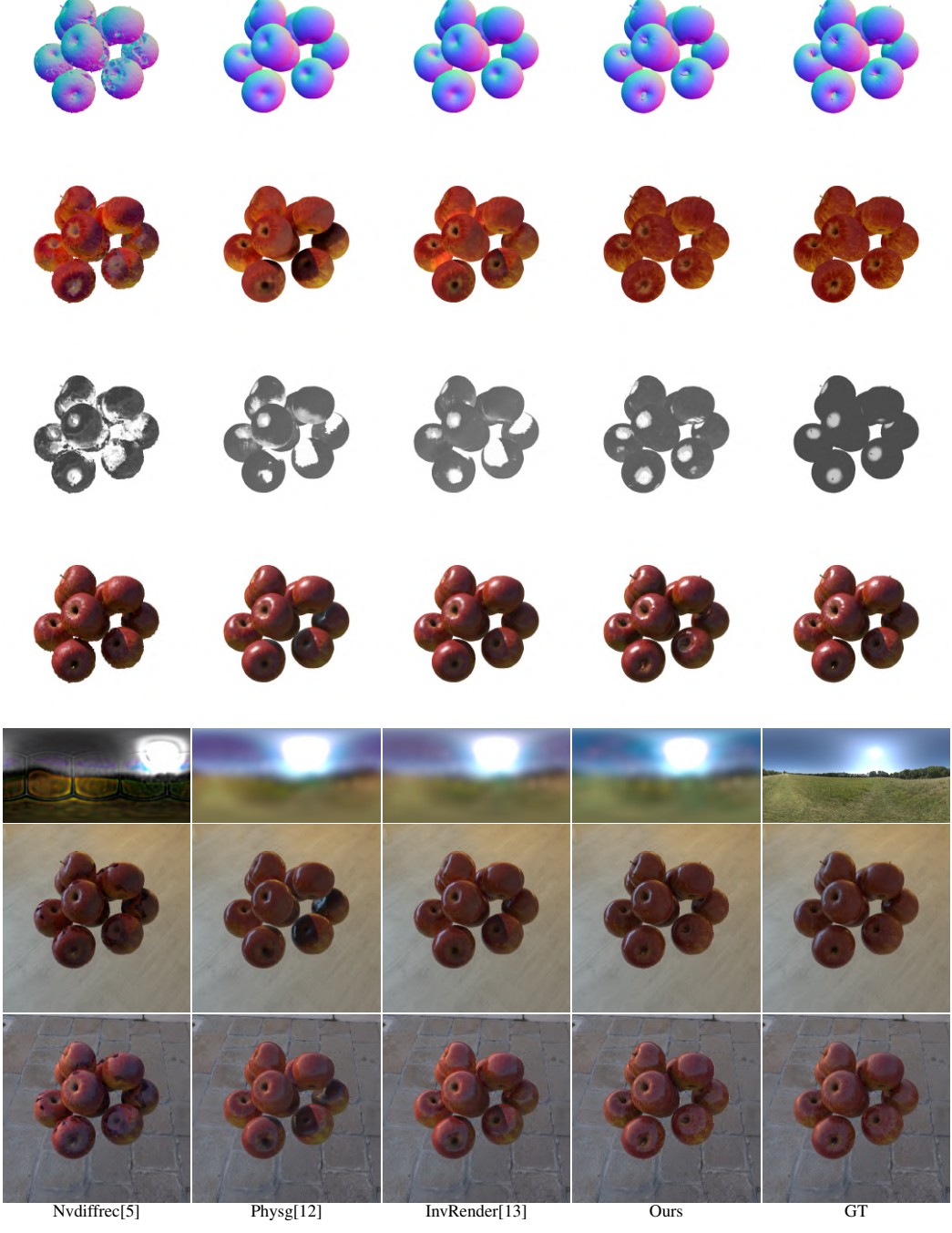

Figure 15: **Multi-view synthetic. Apple.**

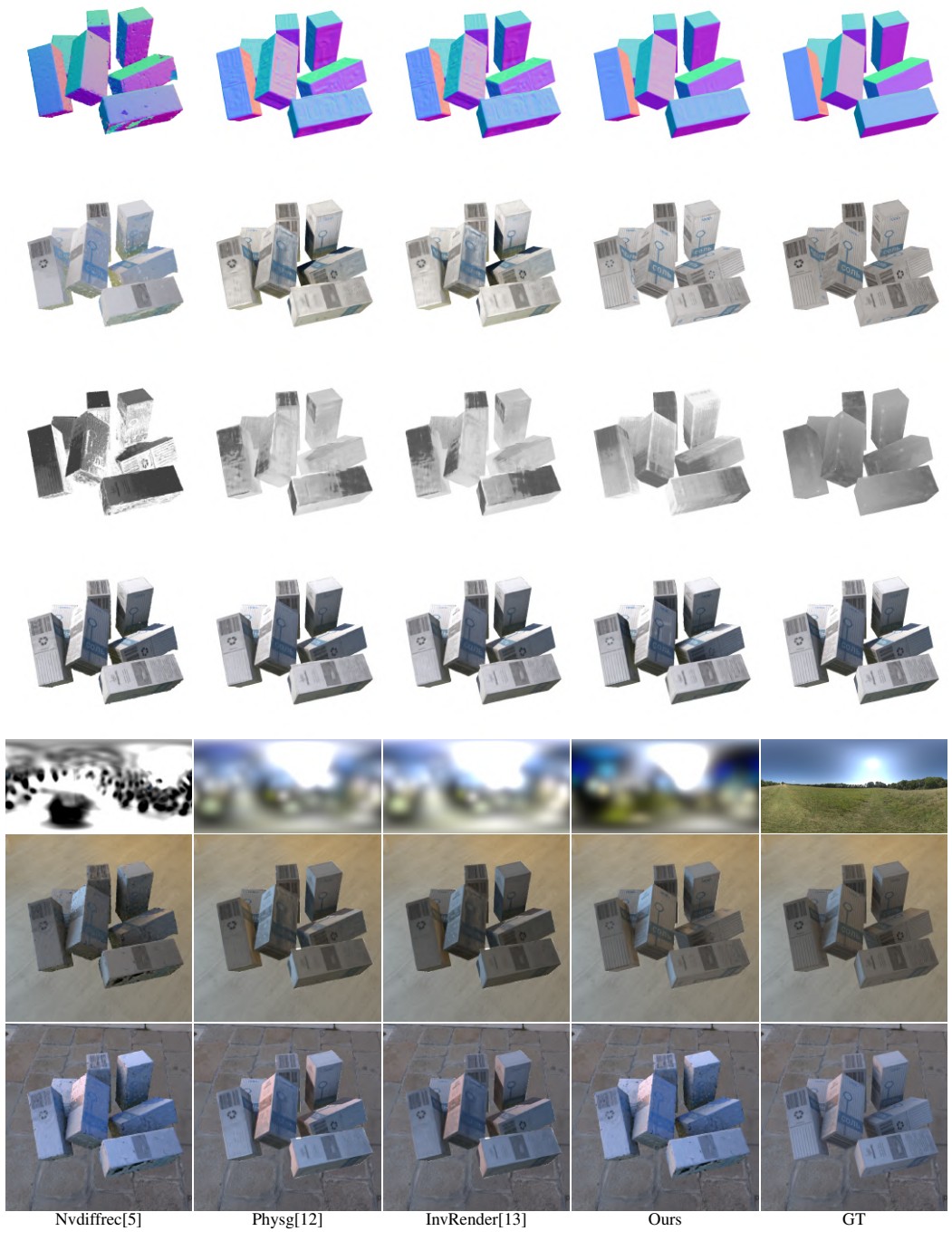

| Nvdiffrec[5] | Physg[12] | InvRender[13] | Ours | GT |

Figure 16: **Multi-view synthetic. Box.**

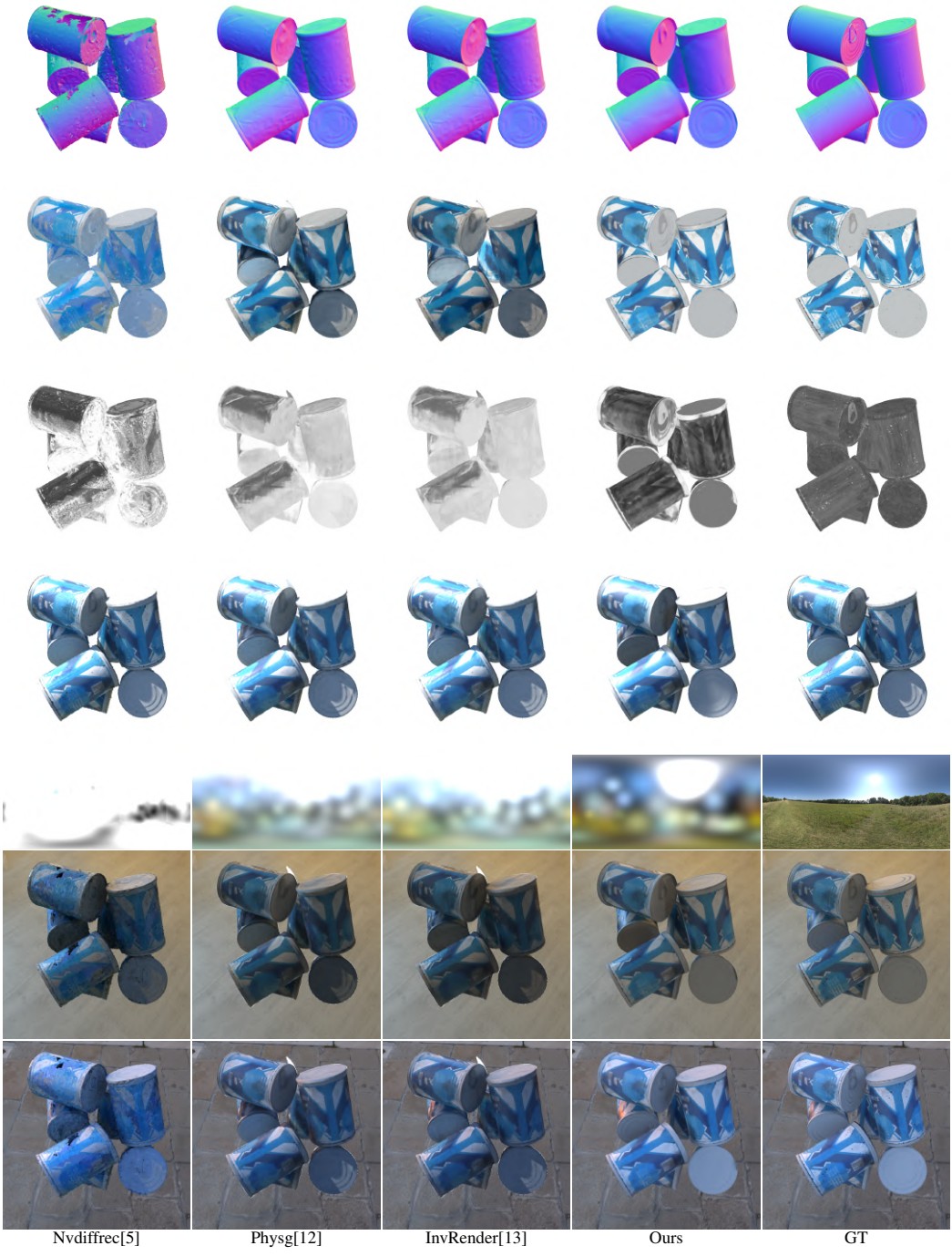

Nvdiffrec[5]     Physg[12]     InvRender[13]     Ours     GT

Figure 17: **Multi-view synthetic. can.**

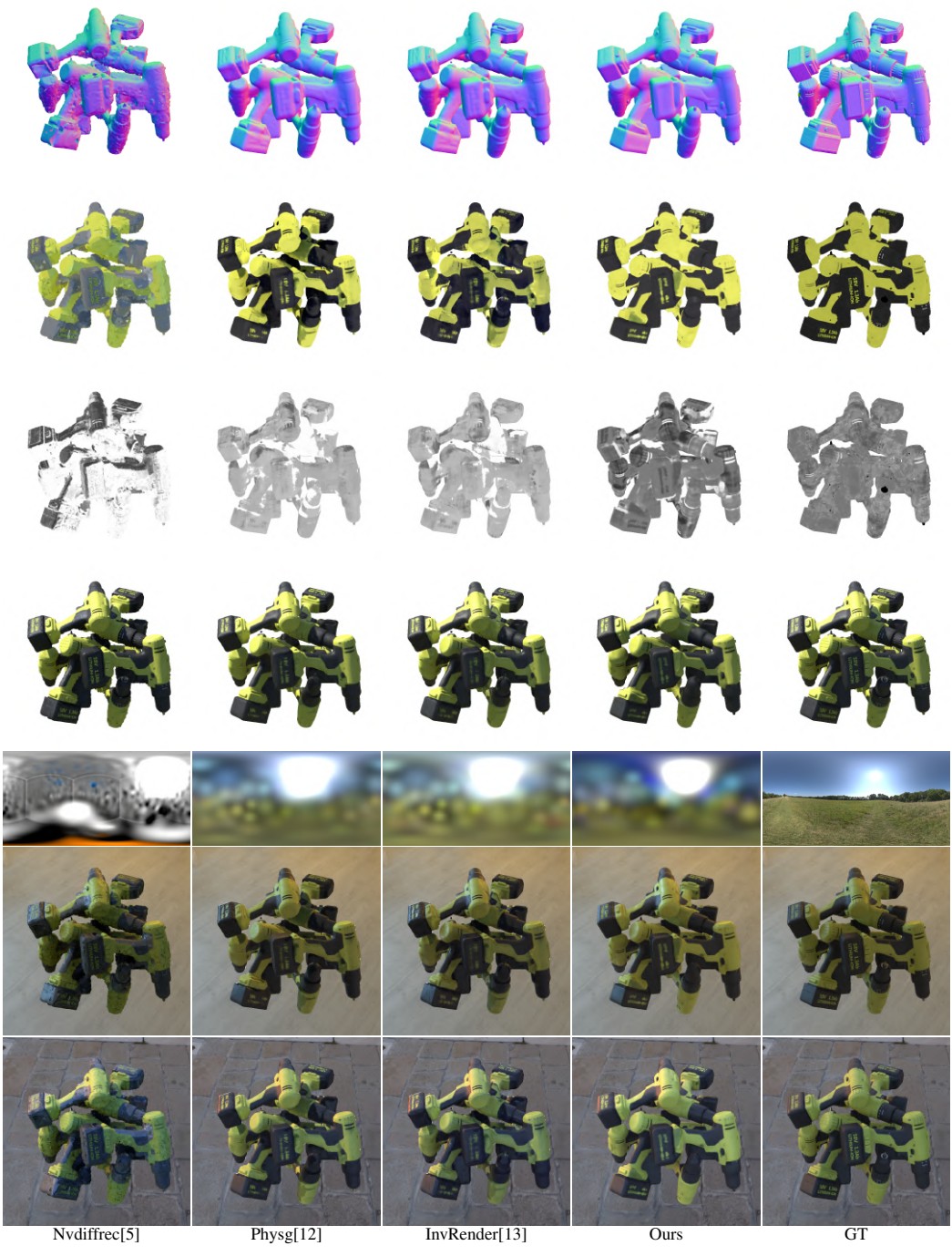

Nvdiffrec[5]    Physg[12]    InvRender[13]    Ours    GT

Figure 18: **Multi-view synthetic. drill.**

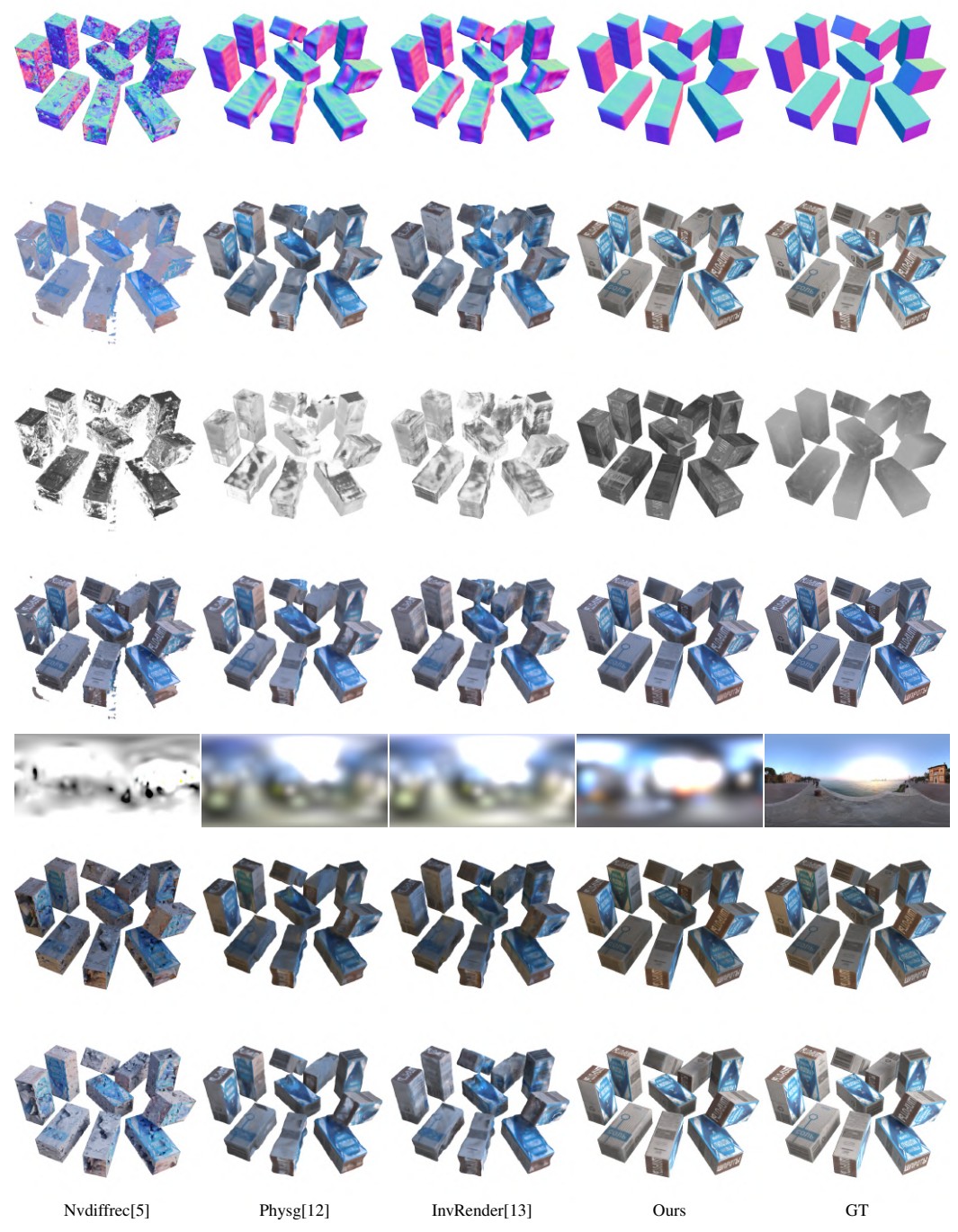

| Nvdiffrec[5] | Physg[12] | InvRender[13] | Ours | GT |

Figure 19: **Single-view synthetic. box.**

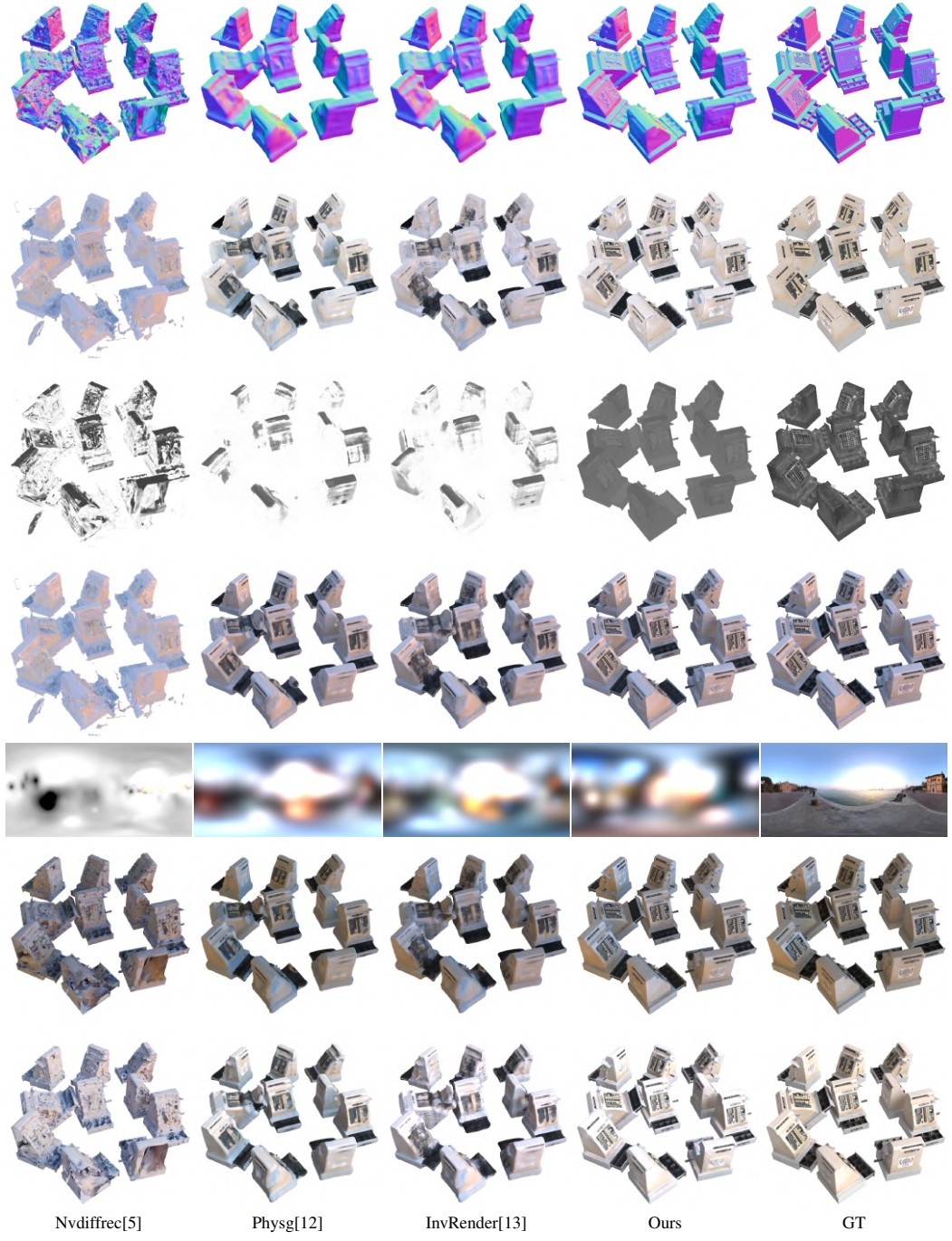

Nvdiffrec[5]   Physg[12]   InvRender[13]   Ours   GT

Figure 20: **Single-view synthetic. cash.**

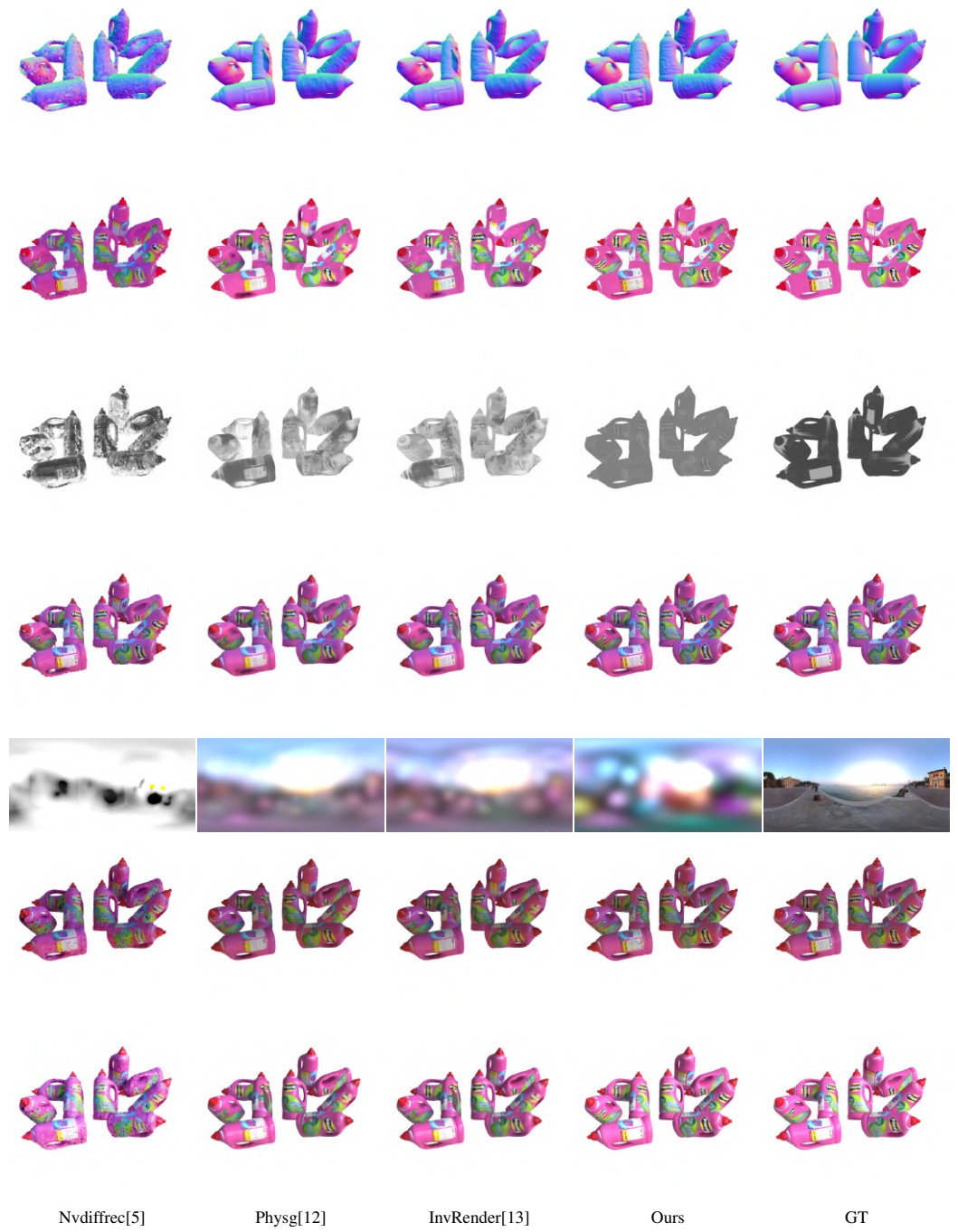

|  |  |  |  |  |
|---|---|---|---|---|
| Nvdiffrec[5] | Physg[12] | InvRender[13] | Ours | GT |

Figure 21: **Single-view synthetic. cleaner.**

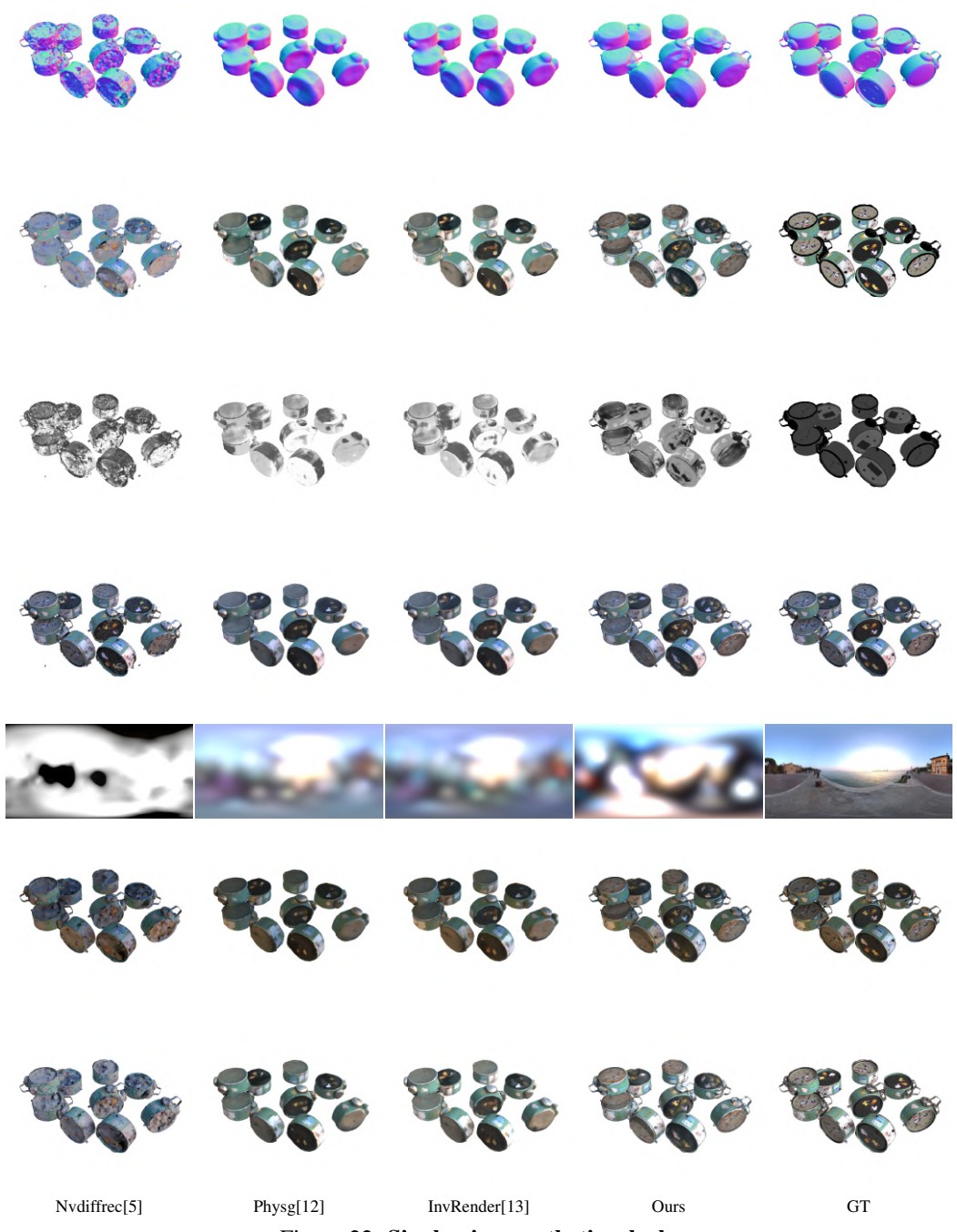

Nvdiffrec[5]   Physg[12]   InvRender[13]   Ours   GT

Figure 22: **Single-view synthetic. clock.**

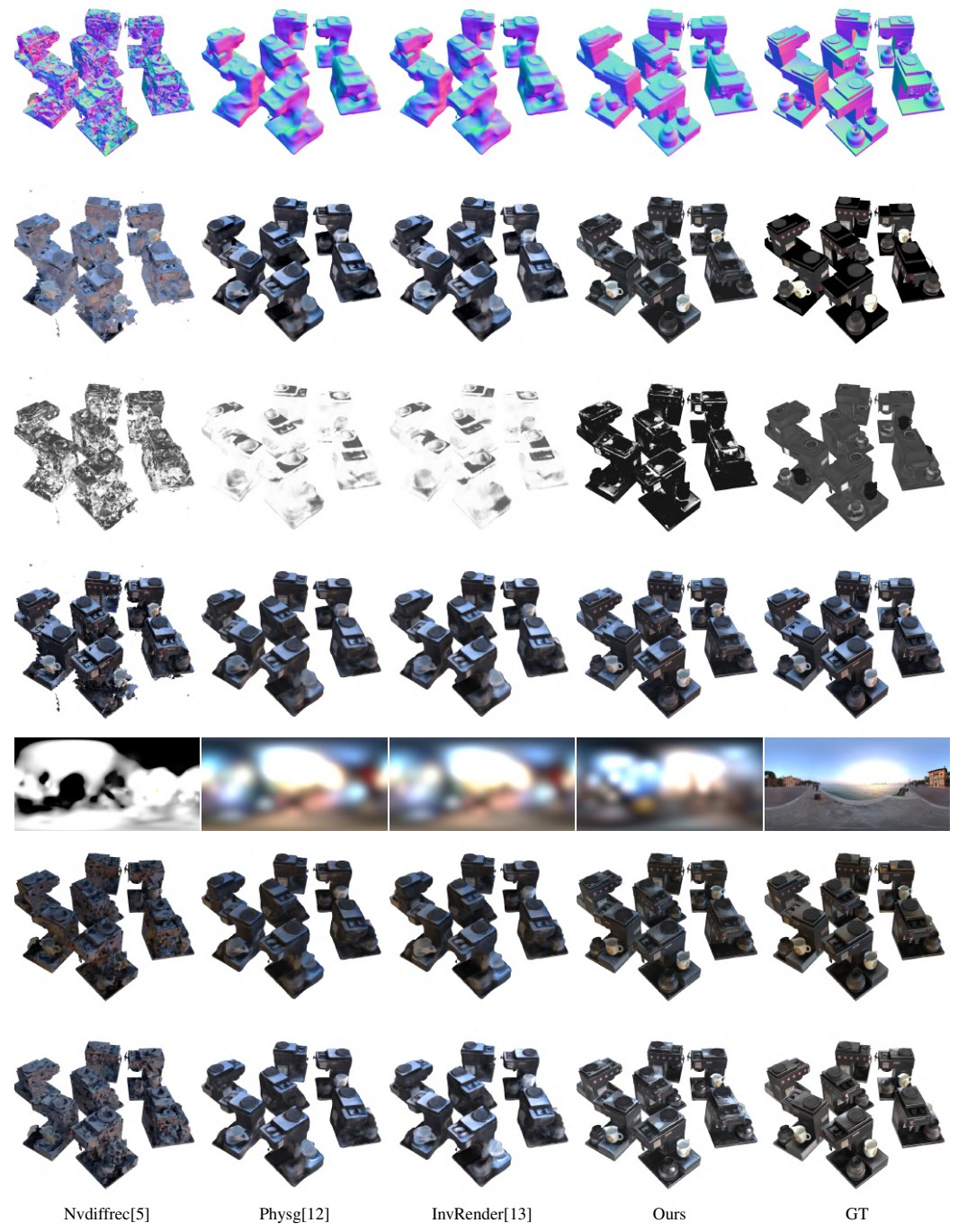

| Nvdiffrec[5] | Physg[12] | InvRender[13] | Ours | GT |

Figure 23: **Single-view synthetic. coffee.**

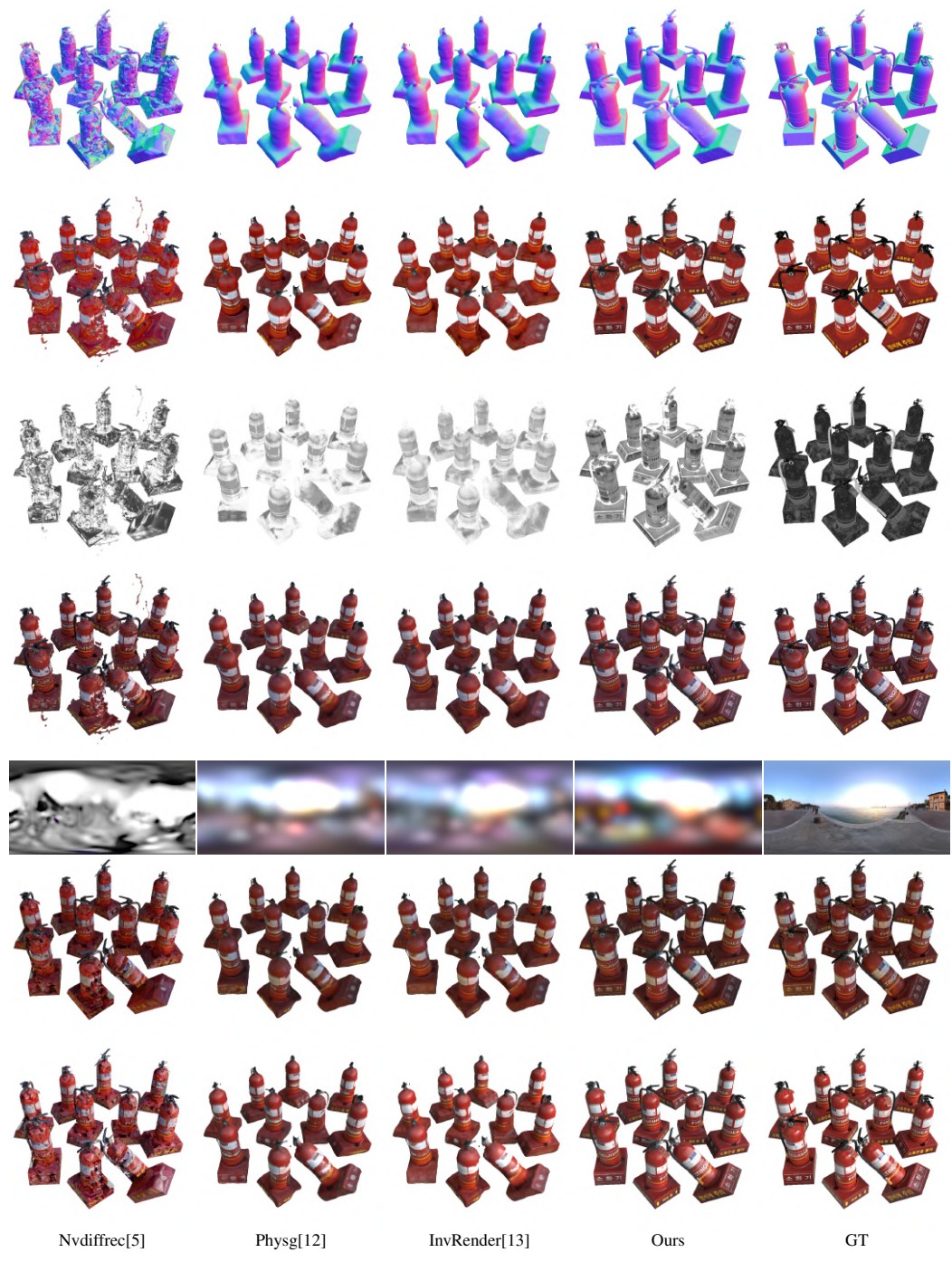

Nvdiffrec[5]    Physg[12]    InvRender[13]    Ours    GT

Figure 24: **Single-view synthetic. fire.**

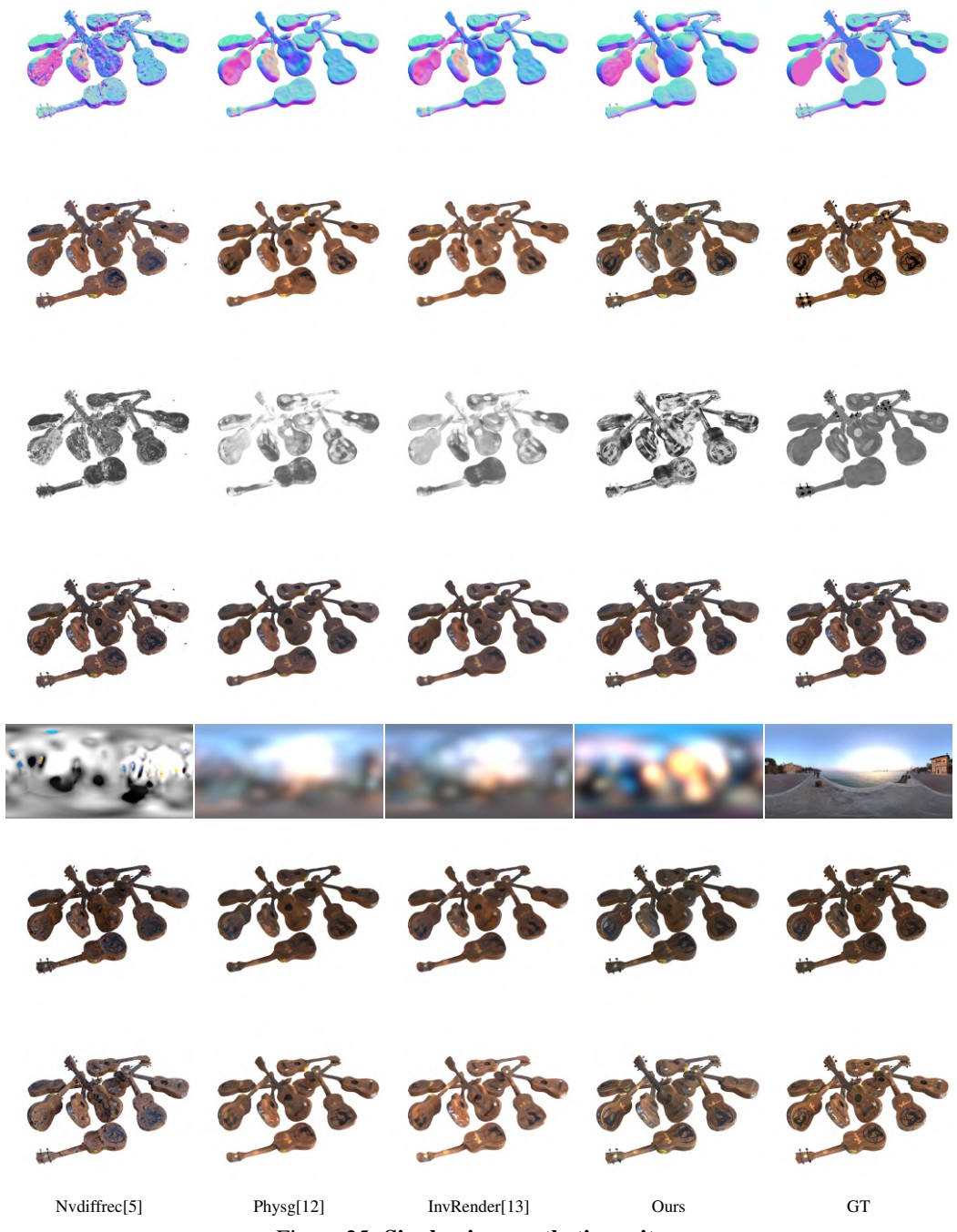

Nvdiffrec[5]      Physg[12]      InvRender[13]      Ours      GT

Figure 25: **Single-view synthetic. guitar.**

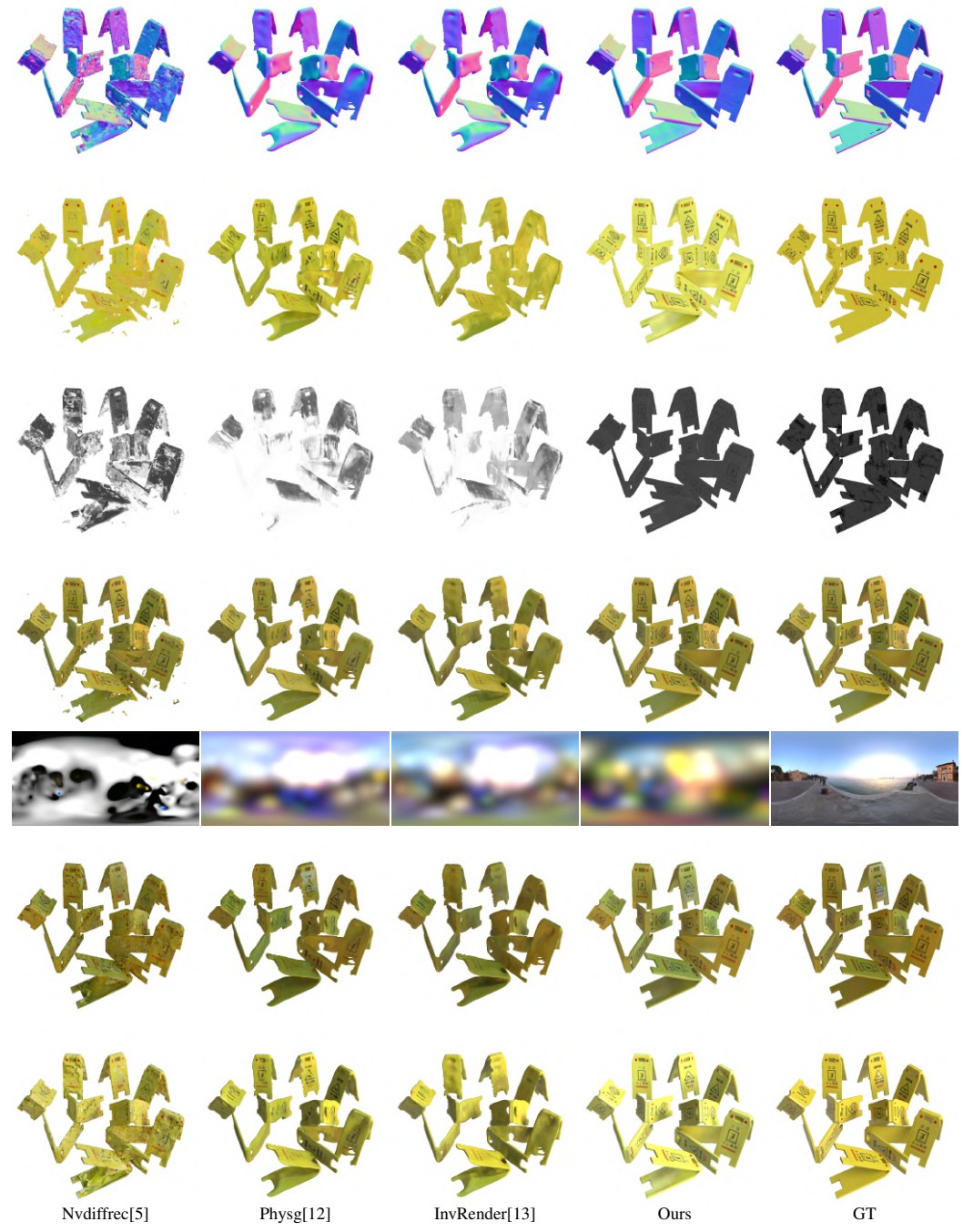

Nvdiffrec[5]       Physg[12]       InvRender[13]       Ours       GT

Figure 26: **Single-view synthetic. sign.**

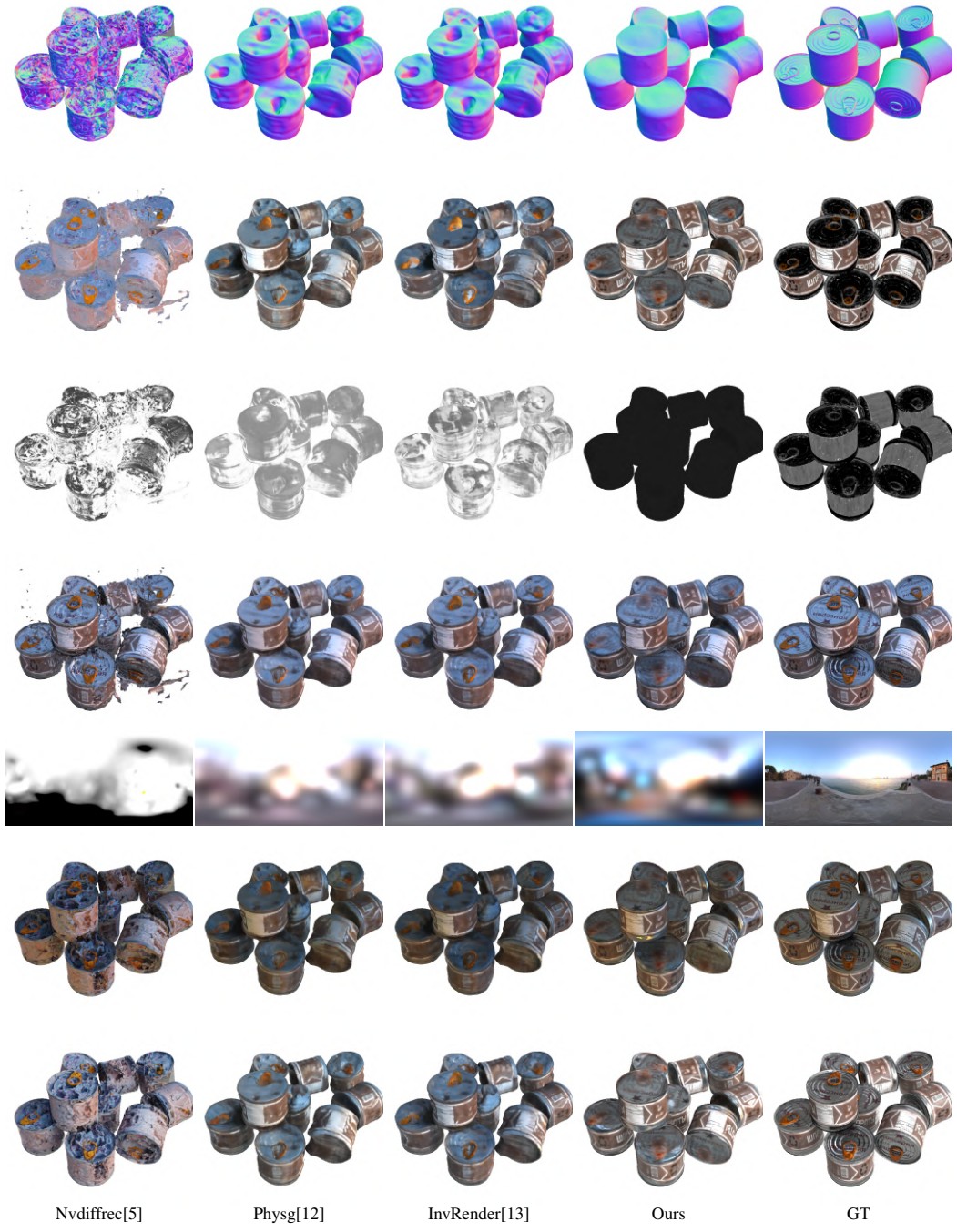

Nvdiffrec[5]      Physg[12]      InvRender[13]      Ours      GT

Figure 27: **Single-view synthetic. tin.**

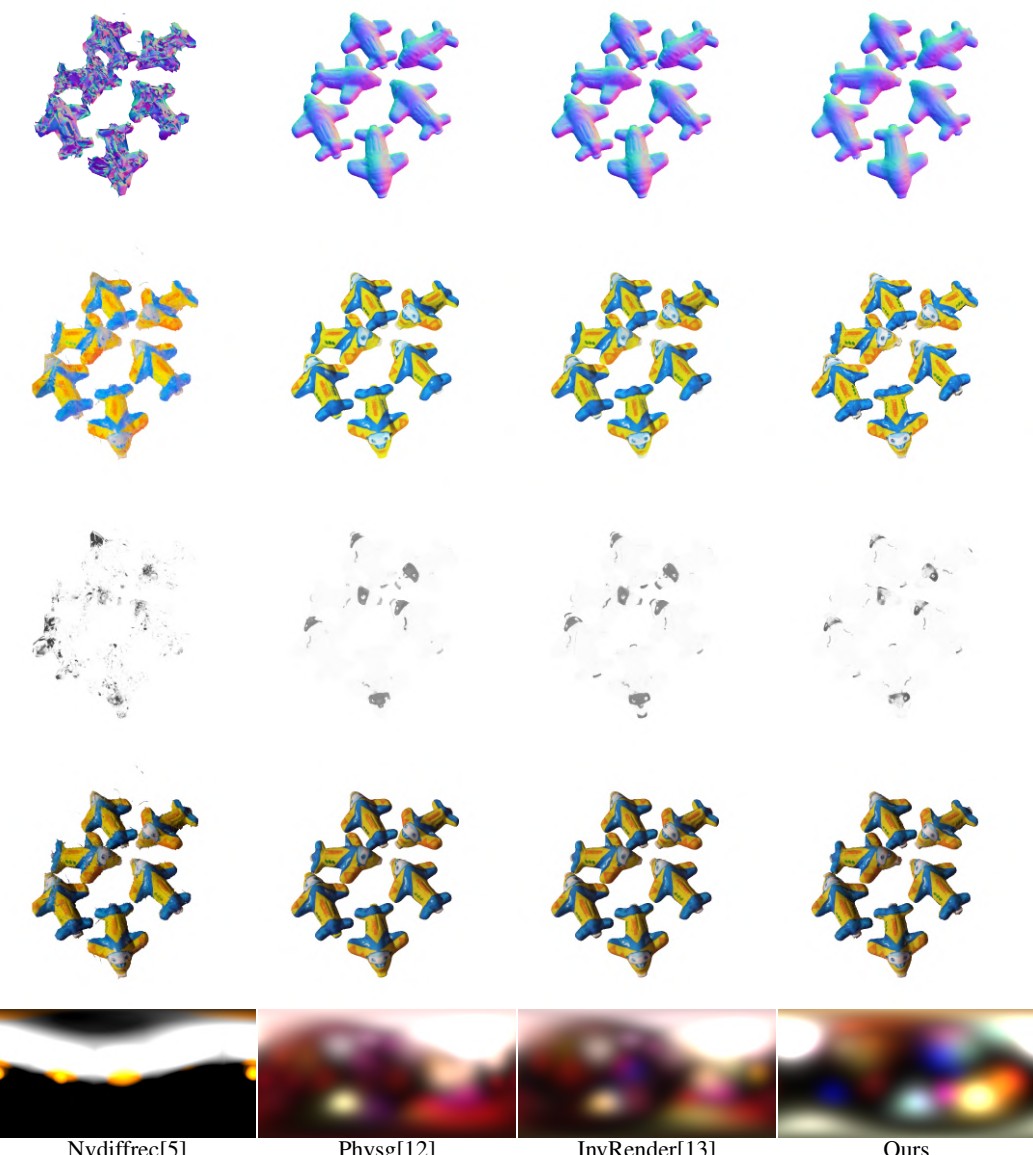

Nvdiffrec[5]                    Physg[12]                    InvRender[13]                    Ours

Figure 28: **Single-view realworld. airplane.**

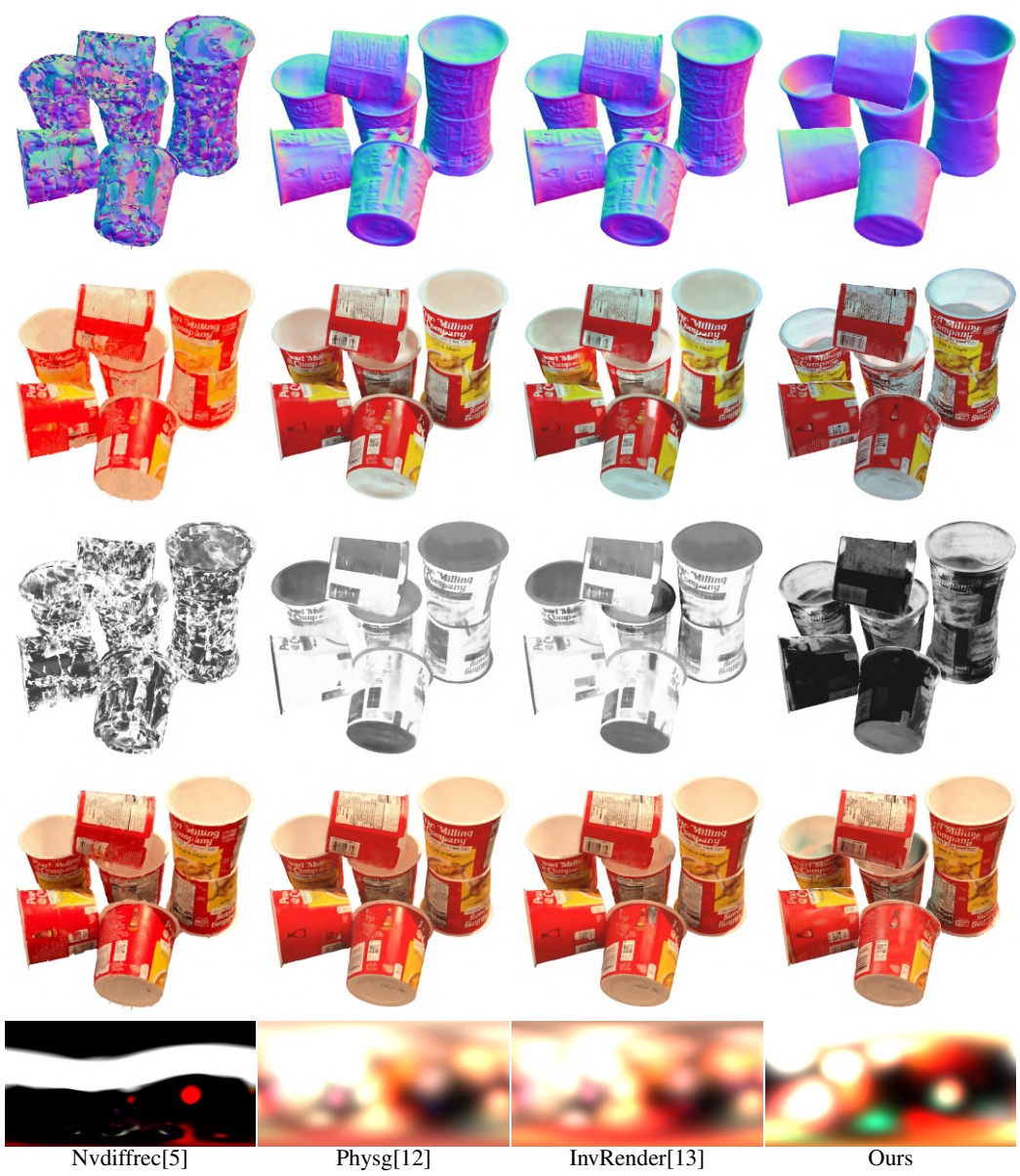

Nvdiffrec[5]    Physg[12]    InvRender[13]    Ours

Figure 29: **Single-view realworld. cake.**

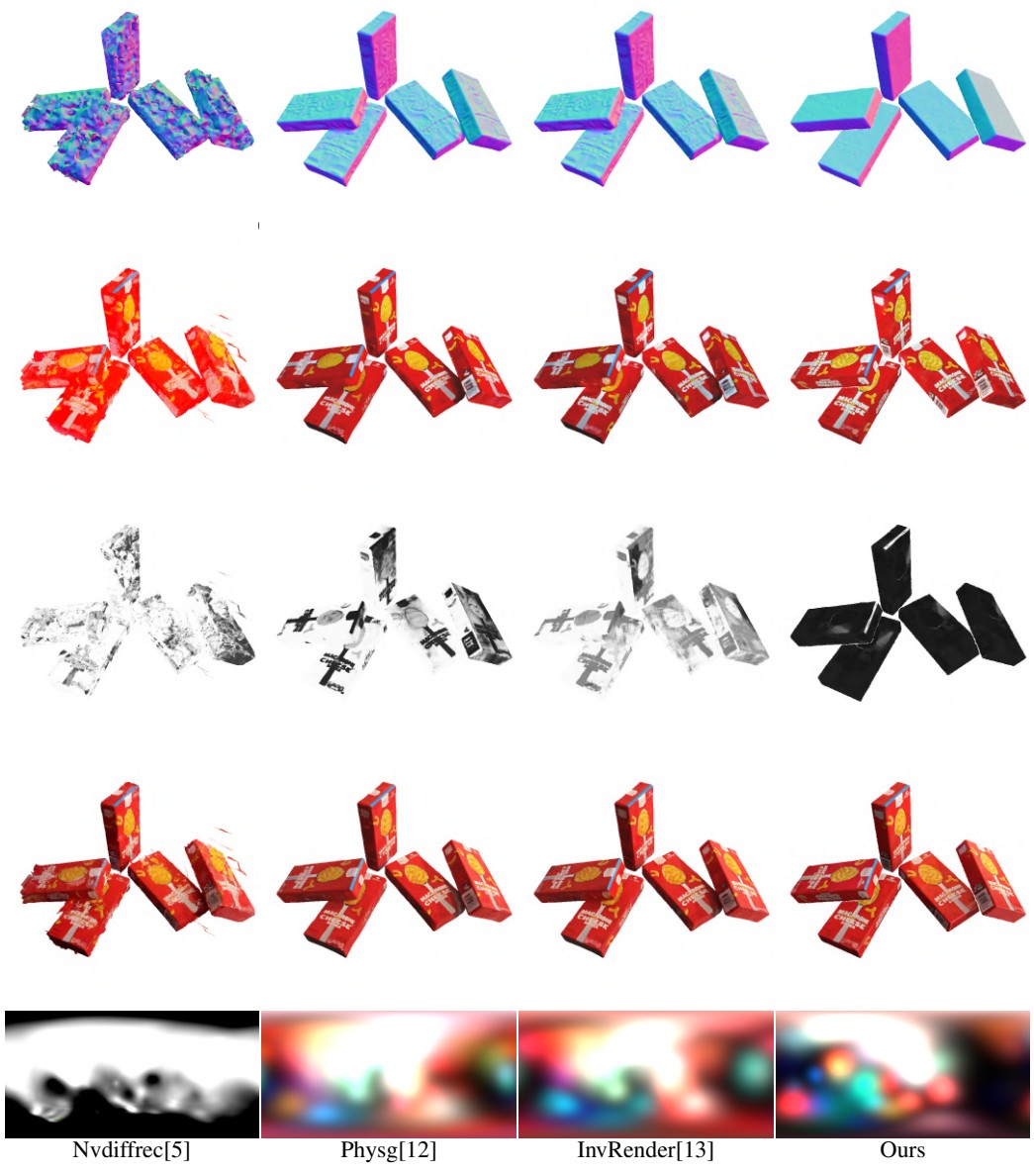

Nvdiffrec[5]    Physg[12]    InvRender[13]    Ours

Figure 30: **Single-view realworld. cheese.**

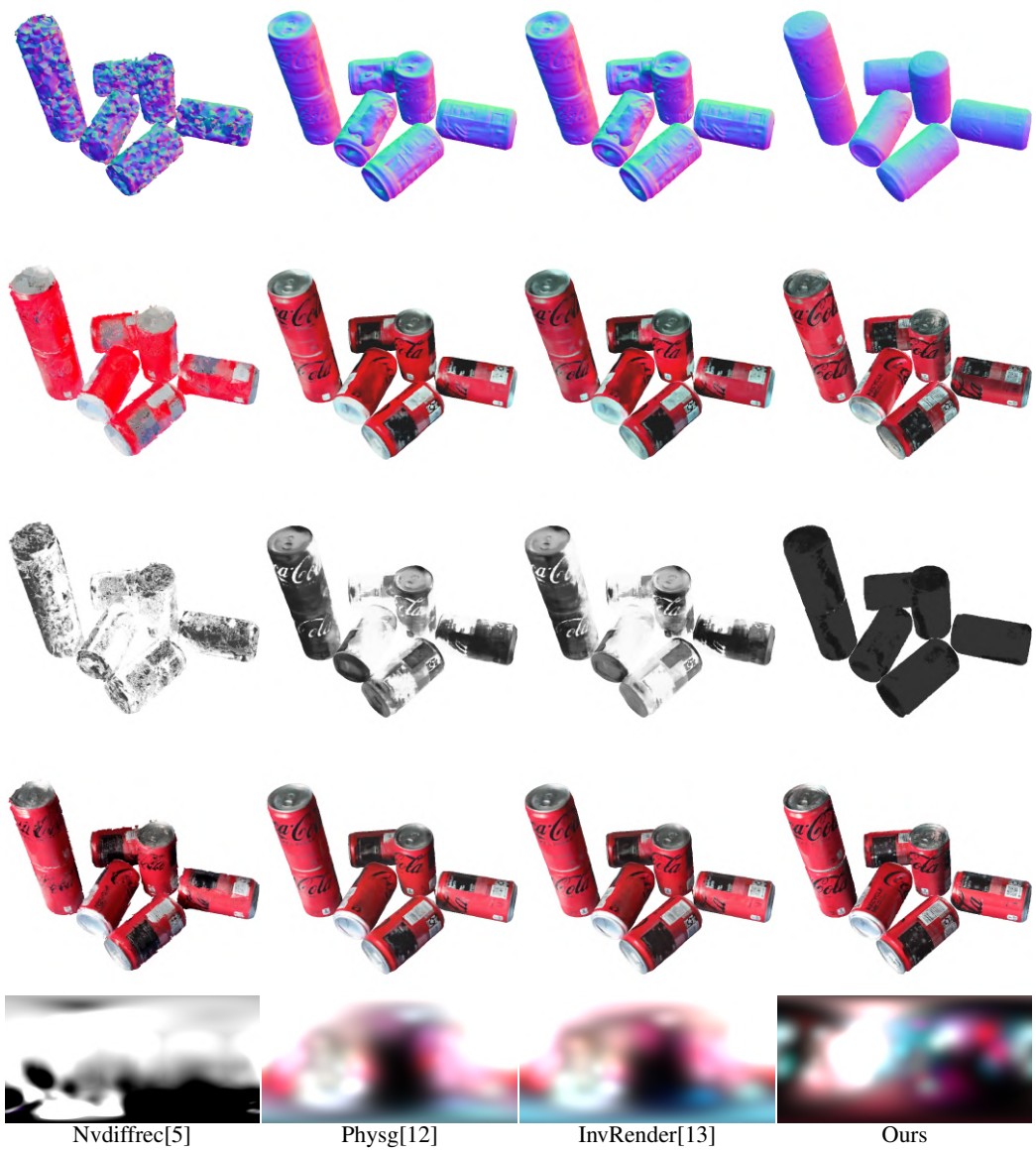

Nvdiffrec[5]           Physg[12]           InvRender[13]           Ours

Figure 31: **Single-view realworld. cola.**

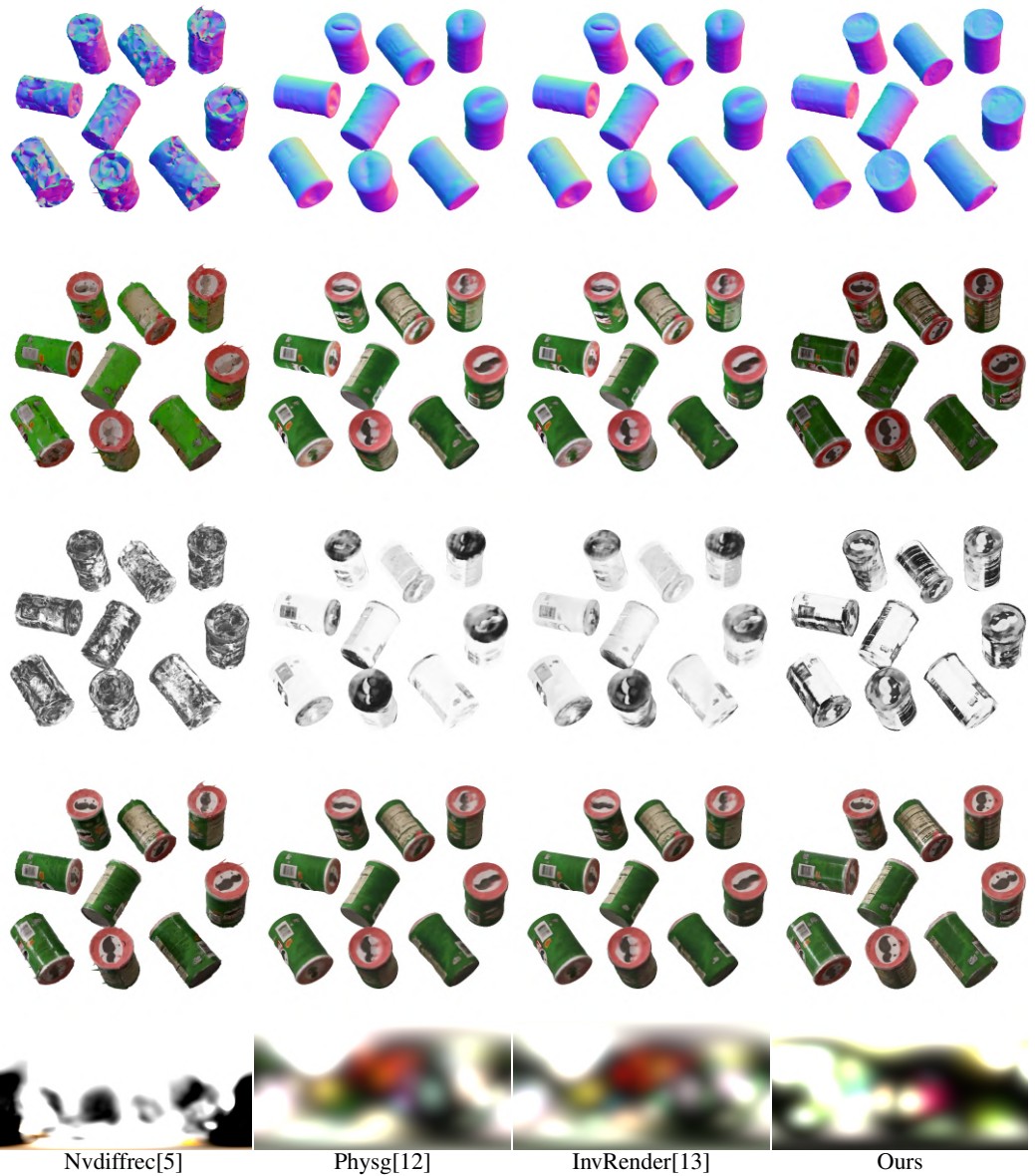

Nvdiffrec[5]          Physg[12]          InvRender[13]          Ours

Figure 32: **Single-view realworld. potato.**

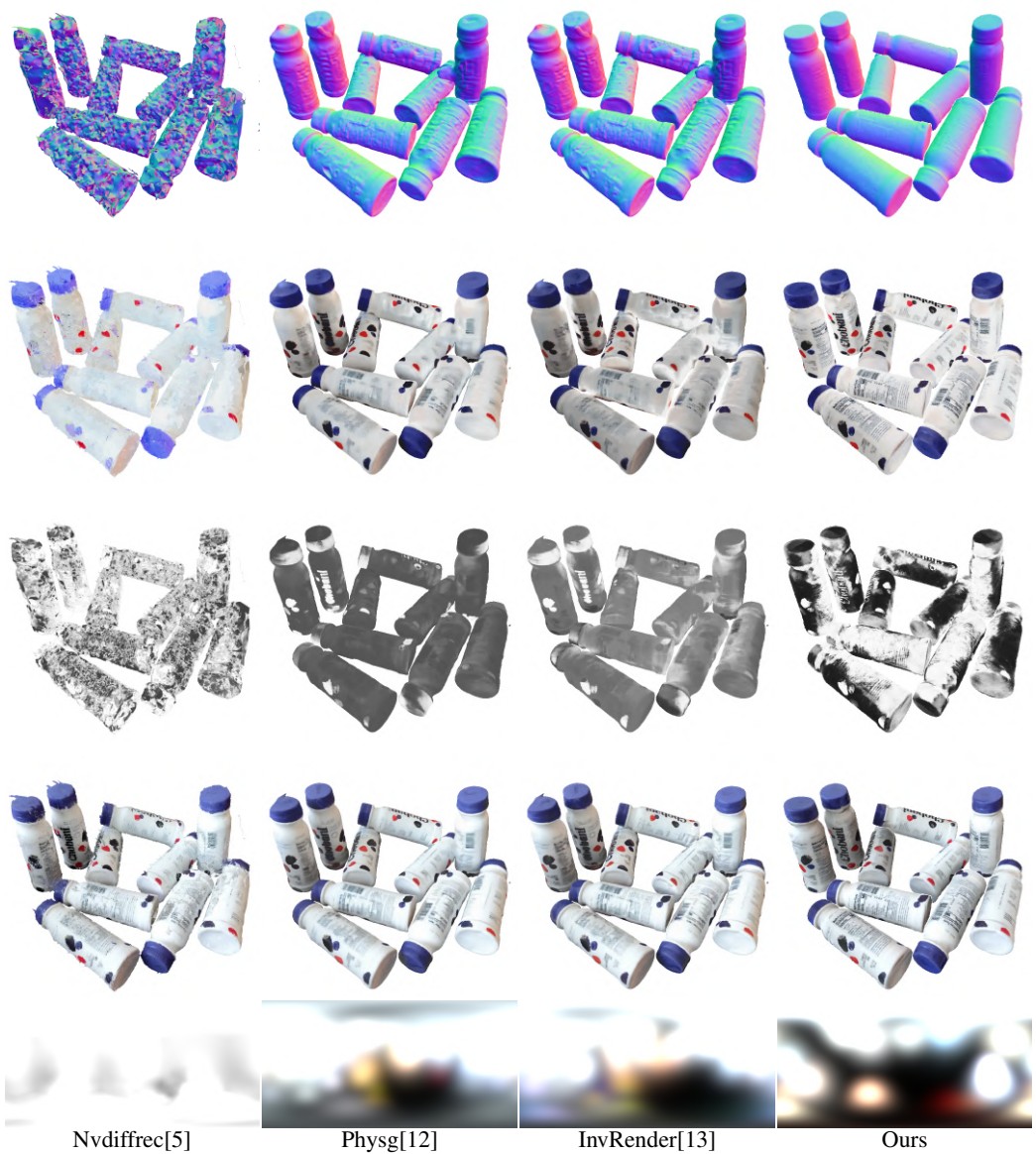

Nvdiffrec[5]  Physg[12]  InvRender[13]  Ours

Figure 33: **Single-view realworld. yogurt.**