# OpenReview forum: "Structure from Duplicates: Neural Inverse Graphics from a Pile of Objects"
_NeurIPS.cc/2023/Conference — NeurIPS 2023 poster_

### Official Review · Reviewer_vbZQ · 2023-07-03

**Soundness:** 3 good
**Presentation:** 3 good
**Contribution:** 3 good
**Rating:** 7
**Confidence:** 4

**Summary:**

The proposed method in Structures from Duplicates leverages multiple appearance of an identical objects in a single image to reconstruct the geometry and material properties of this object. Each instance is assigned a virtual camera, such that the shared object representation is aligned in the same space. Therefore multi-view reconstruction can be applied reconstructing each instance in a separate view.
First each object is identified using a segmentation method and cropped from the image. Initial camera poses are estimated using Structure from Motion (SfM) method COLMAP on the cropped images of each segmented object. To avoid the issues that come with extreme differences in the view points for SfM methods, images of instances are rotated and optimal pairs identified.
From estimated view point and image pairs the surface geometry following NeuS, the object's decomposed BRDF and environmental lighting following InvRender are optimized following known inverse rendering approaches.
Results on synthetic data show comparable quality with multi-view reconstruction approaches, while using duplicates of the same object in a single image and present better performance in the multi-view case, leveraging virtual views from duplicates. Additional results on real data are presented that validate the result on synthetic data.


**Strengths:**

The authors present an interesting solution to a common setting in the real world. The given introduction and related works motivates this setting of multiple instances well, which is an interesting setting by itself, that has not been explored by prior work in a similar way. The method explanation is thorough on all major details. Building on well-known and established decomposed neural rendering architectures and ideas (NeuS, simplified BRDF) is acceptable because this representation is not the main focus of the paper and the method does not lack to my understanding.

Presented results justify most claims made in the beginning and ablation studies validate design choices. Especially showing that explicitly modelling lighting, geometry and material properties has an impact on the reconstruction quality over multi-view reconstruction on a single object is an important insight of the presented work.

One could argue, that this paper lacks novelty and just applies known concepts, such as instances from repeating object classes in the same images and building a prior from that. In my opinion neglecting a pre-trained prior is a strong advantage of the paper.

**Weaknesses:**

A major weakness of the presented paper is the variety in the results. The synthetic and real dataset are both on the smaller side and pretty homogeneous, as elaborated in the following:

- While the motivation is convincing and Fig.2 shows a good variety of examples for the presented setting, the presented results in the main paper lack some variety and complexity that can be seen in the presented real world examples like the screws and chairs. The main paper presents only simple, closed object geometries, such as bottles, cans, apples, or the toy plane in the synthetic data with a lack of thin structures and complex textures. On the other side the alarm clock and coffee maker in the supplementary of this work show that the reconstruction can lead to overly smooth geometry on complex structures. This is expected on the limited number of views, but not properly addressed in the main paper.

- Not purely a weakness, but a suggestion to strengthen the claims I would ask the authors to add results on more complex real geometries and scenes. To my best understanding this can be a complex task to capture and reconstruct, but it would have a positive impact on the limited scope provided by cans, bottles and packaging.

Ablation studies on the number of objects from 2 to 10 objects on synthetic data show results, in which a higher number of objects significantly improves the results on almost all metrics of the reconstructed object. I would argue that even a higher number could improve results further and with respect to the presented motivation such experiments are not unlikely in the real world. Therefore an upper bound for improvements could further strengthen the paper.

**Questions:**

Partially addressed in the weakness sections:
- Were ablation studies performed on all objects (e.g. 2-10 instances per object category), please clarify?

**Limitations:**

I think the authors adequately addressed all major limitations, such as a required instance segmentation and the reliance on COLMAP poses. Additionally they addressed potential solutions. I would suggest to add a short paragraph on the scope of the paper with respect to the experiments, that I suggested in the weakness section.

---

> ### Author Rebuttal · Authors · 2023-08-10
>
> **Clarification on abaltion studies**: The analysis on the number of instances were performed on a randomly selected scene. To validate our analysis holds for other scenes, we conduct the same experiments on all objects. Due to resource constraints, we only managed to fnish training on two more *randomly selected* objects. The rest is still running in the background. As shown in the table below (Table 3 and Table 4), the performance of our approach scales with respect to the number of instances. We will include the numbers for all objects in the final version.
>
>
> **Overly-smoothed geometry**: We conjecture the limitation stems from the representation power of NeuS [1] which forms basis of our geometry backbone. We, howver, note that our pipeline is backbone-agnostic. We can always replace existing components with the latest and great models. We originally chose NeuS because of its simplicity. We are currently exploring more advanced representations (*e.g.*, [2]) and will include our investigations in the final version.
>
>
> **Expanding the limitation section**: Thanks for the great suggestion. We will include an additional paragraph discussing the scope of the paper in the final version.
>
>
> | Cleaner | Rendering |        |         | Albedo  |        |         | Roughness | Relighting |        |          | Env Light | Geometry |             |          |         |
> |---------|-----------|--------|---------|---------|--------|---------|-----------|------------|--------|----------|-----------|----------|-------------|----------|---------|
> | Num     | PSNR ↑    | SSIM ↑ | LPIPS ↓ | PSNR ↑  | SSIM ↑ | LPIPS ↓ | MSE ↓     | PSNR ↑     | SSIM ↑ | LIPIPS ↓ | MSE ↓     | CD ↓     | Precision ↑ | Recall ↑ | F1 ↑    |
> | 2       | 14.390    | 0.457  | 0.439   | 12.358  | 0.412  | 0.504   | 0.077     | 14.090     | 0.423  | 0.457    | 0.045     | 0.050    | 0.808       | 0.576    | 0.673   |
> | 4       | 15.957    | 0.538  | 0.337   | 13.244  | 0.487  | 0.401   | 0.081     | 15.318     | 0.510  | 0.357    | 0.052     | 0.025    | 0.780       | 0.828    | 0.803   |
> | 6       | 17.260    | 0.630  | 0.290   | 14.052  | 0.551  | 0.354   | 0.100     | 16.542     | 0.608  | 0.297    | 0.048     | 0.029    | 0.744       | 0.762    | 0.753   |
> | 8       | 19.381    | 0.689  | 0.242   | 15.370  | 0.607  | 0.304   | 0.159     | 18.083     | 0.661  | 0.256    | 0.064     | 0.018    | 0.872       | 0.967    | 0.917   |
>
> Tabel 3: Experiment result for different number of duplicated objects (Cleaner).
>
>
> | Fire | Rendering |        |         | Albedo  |        |         | Roughness | Relighting |        |          | Env Light | Geometry |             |          |         |
> |------|-----------|--------|---------|---------|--------|---------|-----------|------------|--------|----------|-----------|----------|-------------|----------|---------|
> | Num  | PSNR ↑    | SSIM ↑ | LPIPS ↓ | PSNR ↑  | SSIM ↑ | LPIPS ↓ | MSE ↓     | PSNR ↑     | SSIM ↑ | LIPIPS ↓ | MSE ↓     | CD ↓     | Precision ↑ | Recall ↑ | F1 ↑    |
> | 2    | 16.902    | 0.430  | 0.345   | 13.788  | 0.380  | 0.410   | 0.306     | 16.082     | 0.402  | 0.340    | 0.092     | 0.038    | 0.692       | 0.600    | 0.643   |
> | 4    | 18.157    | 0.531  | 0.271   | 14.799  | 0.461  | 0.325   | 0.311     | 17.224     | 0.500  | 0.272    | 0.080     | 0.029    | 0.713       | 0.781    | 0.745   |
> | 6    | 19.086    | 0.560  | 0.252   | 15.674  | 0.502  | 0.287   | 0.233     | 18.088     | 0.531  | 0.242    | 0.057     | 0.025    | 0.774       | 0.825    | 0.799   |
> | 8    | 21.029    | 0.622  | 0.248   | 16.632  | 0.531  | 0.301   | 0.305     | 19.602     | 0.582  | 0.238    | 0.072     | 0.028    | 0.847       | 0.747    | 0.794   |
>
> Tabel 4: Experiment result for different number of duplicated objects (Fire extinguisher).

---

> > ### Comment · Reviewer_vbZQ · 2023-08-17
> >
> > **Additional results on ablation studies** I still wonder if more than 10 objects (results in the main paper) would improve the reconstruction results significantly and if there is a number of objects that can count as an upper bound for reconstruction improvements. Do you have any insights on that?
> >
> > **Representation agnostic method** Clarifying, that the presented approach is agnostic of the 3D representation and can be improved with more advanced models makes sense. Looking forward to the Neuralangelo [2] results!

---

> > > ### Author Response · Authors · 2023-08-21
> > >
> > > To understand how the performance of our model scales with respect to the number of instances, we increase the number of objects to 15 and 20. Due to time constraint, we only test on the new scene of boxes. The preliminary results are shown below.
> > >
> > > We will include more comprehensive investigation (e.g., more than 20+ objects, more categories) in the final version. Our current hypothesis is that there do exist an "sweet spot" number due to the accumulated 6 DoF pose error brought by heavier occlusion, the limited capacity of the visibility field, and limited image resolution.
> > >
> > > |     | Albedo  | Roughness | Env Light |      Relight | Env Light | Normal error  |
> > > |-----|---------|-----------|-----------|--------------|-----------|---------------|
> > > | Num | PSNR ↑  | MSE ↓     | MSE ↓     | PSNR ↑       | MSE ↓     | °↓            |
> > > | 5   | 20.053  | 0.086     | 0.086     | 19.229       | 0.086     | 3.329         |
> > > | 10  | 19.517  | 0.095     | 0.095     | 19.021       | 0.095     | 2.578         |
> > > | 15  | 20.522  | 0.093     | 0.093     | 18.769       | 0.093     | 2.433         |
> > > | 20  | 22.530  | 0.077     | 0.077     | 19.560       | 0.077     | 2.279         |

---

> > > > ### Comment · Reviewer_vbZQ · 2023-08-21
> > > >
> > > > Thank you for your time investigating my questions and the provided results, those are really insightful. I encourage the authors to add all additional results and discussions to the final manuscript. And after reading all detailed discussions I will keep my rating and recommend to accept this paper.

---

### Official Review · Reviewer_4hmW · 2023-07-04

**Soundness:** 2 fair
**Presentation:** 3 good
**Contribution:** 1 poor
**Rating:** 3
**Confidence:** 4

**Summary:**

This paper presents a pipeline for recovering object shape and surface materials from one single of multiple identical duplicate instances. The pipeline first extracts instance masks, registers camera poses using COLMAP (with in-plane rotation augmentation) and recovers shape, materials and environment lighting through inverse rendering.

**Strengths:**

### S1 - A sound inverse rendering pipeline
- The proposed pipeline is carefully designed, starting from instance segmentation from Mask2former, to a camera pose estimation step using COLMAP with in-plane rotation augmentation, and finally to inverse rendering of shape, BRDF material and SG env map similar to [65].
- Given that the instances are mostly identical, this pipeline in general works well.

### S2 - Good writing
- The paper is well written, with clear descriptions of the problem setup and most technical details.


**Weaknesses:**

### W1 - Quite a trivial task assuming *identical* instances
- Fig. 1: "the world is full of identical objects" - it might be difficult to argue that different instances that appear in one scene are *identical*. This is in contrast to the claim in [66], which says "no two roses in the natural world are identical".
- This paper focuses more on manufactured and synthetic objects, which appear more similar thanks to the consistent manufacturing process, but in reality, different objects often vary in both shape and appearance, due to wearing or natural variations, for instance, multiple apples in a basket.
- This is the fundamental assumption for COLMAP to work on these instances. It can easily fail even if there is a slight variation in geometry and appearance, which raises significant concern on the method's robustness to instance variations.
- Once the multiple instances are registered, the rest of the pipeline seems to be a standard inverse rendering task, and is highly inspired from existing work, which I do not find sufficiently interesting.

### W2 - A special case of the rose paper [66]
- Line 111: "independently and concurrently, Zhang et al. [66]" - NeurIPS has a two-month rule for claiming concurrent work, including arXiv papers. [66] appeared on arXiv in Dec 2022, which is almost half a year from the NeurIPS submission deadline. It might be difficult to claim concurrent work.
- Essential, duplicates are an extreme form of the definition of similarity in [66]. The instance variation is ultimately what makes [66] interesting, as it breaks the assumption for a standard multi-view stereo approach, hence the generative modeling.
- If the proposed method can handle instance variations well, such as different rose instances, without assuming a prior pose distribution, it will make a much more compelling argument.

### Other comments
- I suppose the material network is inherited from [65], as the "latent code" $\mathbf{\rho}$ and its regularizer are the same as [65]. But this is not mentioned in the paper.


**Questions:**

It might be hard to change my opinion on this paper, as most of the technical aspects are clear to me. Rather, this is a slight trivial task, especially given [66] has demonstrated results on a more generic case. Yet, I'm still open to objection.

**Limitations:**

The paper attributes the challenge of the modeling instance variation to the shape/material model, but the fundamental challenge is rather the pose estimation with COLMAP to begin with. COLMAP generally fails terribly with deformable objects - also the reason that recent deformable NeRFs typically mask out deformable objects running COLMAP to register cameras.

---

> ### Author Rebuttal · Authors · 2023-08-10
>
> **The task is NOT trivial**: **We respectfully disagree with the statement that "the task is trivial."** How to make inverse graphics/3D reconstruction more robust and work under more extreme scenarios is a challenging and longstanding problem in computer vision. In this work, we take a step forward by exploring the potential of performing structure from motion and recovering object intrinsics and environmental extrinsics *from a single image without pre-trained priors*. Specifically, we focus on the scenarios where there are multiple (near-)identical objects within the scene. By carefully formulating a duality between multiple copies of an object in a single image and multiple views of a single object, we are able to resolve the ambiguities in 3D and effectively recover the properties of interest. **While the idea may seem straightforward AFTER learned, how to design the low-level details and how to make it work in practice requires careful insights and thoughts (as acknowledged by other reviewers).** We urge the reviewer to acknowledge this.
>
> To our knowledge, we are the first attempt to conduct structure from motion from a single image. We agree with the reviewers that the setup may not be as common as others in practice. However, it's important to emphasize that the problem, in its essence, is profoundly ill-posed and cannot be resolved without relying on certain priors. While some methods, such as [3], may ease the "identical" constraint, they also introduce other strong assumptions (e.g., knowledge of the camera distribution and vast amounts of objects for the generative approach (which we discuss in greater detail below). In contrast, our method chooses a distinct approach, addressing the issue from a purely multi-view geometric perspective. Moreover, our preliminary findings, detailed below, suggest that our methodology can also accommodate minor variations.
>
> **[3] is NOT a superset of our approach**:
> We agree the setup of the two papers are indeed similar. However, instead of one paper being the superset of the other, the two approaches are **orthogonal**.
> - **Assumptions**: First and foremost, while [3] is able to model the variations among the instances, they impose other strong assumptions such as knowing the camera distribution in advance. The strong camera assumption allows them to sidestep the pose estimation step (*i.e.*, SfM) and focus on modeling the variation. In contrast, we assume no knowledge about the poses and attempt to solve for the full inverse rendering pipeline from the beginning. We thus resort to the (near-)identical instances to recover the exact 6 DoF poses.
> - **Approaches**: Secondly, [3] tackle the task through ***generative modeling***. Since they need to train a generative model *per scene*, their approach is very data-hungry. In contrast, our approach mainly exploits ***multi-view geometry*** to recover the underlying intrinsic and extrinsic properties. By explicitly baking the constraints into the modeling procedure, our approach becomes much more data-efficient. To validate our conjecture, we train [3] on three randomly selected scenes from our dataset, each of which has 10 identical instances. As shown in the pdf, the generative model failed to recover either of them. For comparison, we also test our approach on the `crane` image that [3] provided (the only publicly available data), where each instance is slightly different. By augmenting our geomtry backbone with a instance-specific deformation field, we are able to reconstruct resonable poses and recover sensible shape and material.
> - **Extrinsics**: Thirdly, [3] assume a simple phong shading model and assume a dominant directional light, whereas we parameterize our materials with PBR materials and the lighting with enironmental map, allowing us to model complex real world scenarios more effectively. Finally, it is unclear how to extend [3] to multi-view setup. In contrast, our method is naturally compatible with multi-view observations.
>
>
> **Concurrent to [3]**: We are aware of the NeurIPS policy and we do acknowledge that [3] is on arXiv more than 2 months ago. The two works were developed independently, and as evident from our paper, our design choices were not influenced by [3]. The authors did see the paper and were very excited that other researchers share similar interests and have been working on simlar topics. In our submission, rather than overlooking [3], we carefully discussed the differences between the two studies in the "related work" section. We tried our best to put [3] in the light it deserves, despite by the time of submission the paper was not accepted yet and the code was not released. Moreover, now that the code for [3] is available, we are eager to test our method using their dataset and vice versa, aiming for a comprehensive analysis and discussion in the final paper (preliminary results for both can be found in the PDF). Finally, it's crucial to highlight that, while the contexts of our studies are similar, our approaches to addressing the problem are significantly different, as elaborated above.
>
>
> ### Reference
> [1] Wang, Y., Skorokhodov, I., & Wonka, P. (2023). PET-NeuS: Positional Encoding Tri-Planes for Neural Surfaces. In Proceedings of the IEEE/CVF Conference on Computer Vision and Pattern Recognition (pp. 12598-12607).
> [2] Li, Z., Müller, T., Evans, A., Taylor, R. H., Unberath, M., Liu, M. Y., & Lin, C. H. (2023). Neuralangelo: High-Fidelity Neural Surface Reconstruction. In Proceedings of the IEEE/CVF Conference on Computer Vision and Pattern Recognition (pp. 8456-8465).
> [3] Zhang, Y., Wu, S., Snavely, N., & Wu, J. (2023). Seeing a Rose in Five Thousand Ways. In Proceedings of the IEEE/CVF Conference on Computer Vision and Pattern Recognition (pp. 962-971).

---

> > ### Comment · Reviewer_4hmW · 2023-08-17
> > **Thanks for the detailed responses**
> >
> > I would like to thank the authors for preparing the detailed responses. I have posted my response in the general thread above as I was called out there.
> >
> > One additional comment: it would be very helpful if the authors could summarize, once the pose is registered using COLMAP, what are the key differences / technical contributions between the proposed inverse rendering pipeline and [65]?

---

> > > ### Author Response · Authors · 2023-08-19
> > > **Difference to [65]**
> > >
> > > **Contributions from the inverse graphics perspective:**
> > > We have expanded the scope of the inverse graphics family by introducing a novel "single-view duplicate objects" setting. While [65] follows the traditional multi-view single-instance (M-S) framework, our focus lies in a single-view multi-instance (S-M) scenario. It's important to note that these two problem formulations are very distinct.
> > >
> > > Specifically, our proposed setting brings forth several advantages compared to the existing setup. In the conventional configuration, the relative poses between the objects and the lighting sources remain fixed. This poses a significant challenge when attempting to distinguish an object's albedo from its illumination. For instance, it is difficult to tell whether it is a yellow directional light on the left illuminating a white ball or it is a ball that is half-yellow and half-white. In contrast, in our setup, the relative poses between the object and the light sources differ for each "virtual" view (*e.g.,* the env light is rotating in SO(3) space). This allows us to more effectively recover geometry, materials, shadow, and illumination while utilizing the same number of instance observations compared to [65]. Experimental results supporting this claim can be found in Table 1.
> > >
> > > [65] Modeling indirect illumination for inverse rendering
> > >
> > > **Key differences in execution:**
> > > - *Backbone*: In our approach, we utilize NeuS as our neural surface model; and for the visibility field, we opt for Siren instead of ReLU. (We will highlight the distinction in the final version.)
> > > - *Metallic*: We additionally reason for metallic materials (refer to Line 209).
> > > - *MLP distillation*: We distill the geometry MLP into a smaller one for fast classification.
> > > - *Self-occlusion and Inter-occlusion*: Since we have multiple instances in our setup, it is essential to model both inter-object self-casted shadows and inter-object occlusions. Our model goes beyond simple object-centric representation.
> > >
> > > **Remark on “Colmap”:** We stress that our pose estimation process IS NOT just a simple application of Colmap. Merely employing Colmap will not be sufficient, as we have pointed out in the paper (Line 152). Part of our contribution lies in how we jointly reason the 6 DoF poses with a carefully designed matching scheme and how to integrate it into the BA framework. We only use Colmap for its BA optimization.

---

### Official Review · Reviewer_qVSc · 2023-07-05

**Soundness:** 3 good
**Presentation:** 3 good
**Contribution:** 3 good
**Rating:** 6
**Confidence:** 5

**Summary:**

This paper tackles the inverse graphics task of predicting the geometry, material and illumination from a single image containing multiple identical objects. The key insight is to leverage the multiple instances depicted in the image to frame this single-view multi-object reconstruction problem into a better constrained multi-view single-object reconstruction problem. Specifically, the proposed approach (dubbed SfD) first identify and crop the instances in the image, then estimate their 6DoF poses using SfM to create a set of calibrated multi-view images. Finally, a geometric reconstruction module based on prior MVS works is optimized using the resulting multi-views. The framework is validated through experiments conducted on a custom dataset dubbed Dup and introduced in this work.

**Strengths:**

S1. Leveraging duplicated instances to frame a single-view multi-object reconstruction problem into a multi-view single-object one is an interesting idea. Besides, this idea is rather novel; it has only been studied by the concurrent CVPR'23 work of [66]

S2. Experiment section looks strong; the compared methods are relevant and the reported performances are better than prior works

S3. The presentation is very clear and well written. It is easy to walk through the method formulation, which does not contain any major technical flaw. Figures are neat, well designed, and help the understanding.

**Weaknesses:**

**W1. New problem with low applicability**

Although conceptually interesting, this work is tackling a new problem which by essence is very limited in terms of applicability to real-case scenarios, thus questioning its usefulness. Indeed, I see very few cases where the approach developed here could be successful, and quite a lot of failure cases: for example, I think the proposed approach would fail on most of the examples advertised in Figure 2 (see W2 for details).

**W2. Lack of analysis**

The current presentation lacks some crucial analyses to better understand the proposed approach. In particular, I would expect experiments outlining:

- the contribution of each key component (quantitative ablation study): each of the 6 loss terms, rotation aware data augmentation, etc.
- its robustness to real data: in particular not exact duplicates like leaves of a tree
- its robustness to noisy input: an image with duplicates + some other objects (e.g. chairs and tables in Fig. 2) or an image with only one-side instance views (e.g. cars in Fig. 2)
- its robustness to high number of duplicates (e.g. 100/1000 objects like the pile of screws in Fig. 2)
- the failure cases in the general setting (e.g., from the video in the supmat, it seems the geometry of the coke can is actually not so great, why?)

**W3. Highly engineered method**

In addition to the low applicability of the formulated problem, the proposed approach is heavily engineered (6 loss terms, two-level approach including an optimization level with multiple stages, etc). Such amount of engineering typically makes it hard to apply the method to other cases or even to modify and build on top of it to improve the method.


**Questions:**

- Table 1 is overflowing and appears way too early in the presentation (harms the overall readability)
- L142: for completeness, I would suggest formulating the changes in the intrinsic matrix when cropping is performed (in the supplementary material)
- Eq (4) and (5): there is no scaling factors to weight the different loss terms?
- in general, the paper looks crowed because spaces have been modified; I think the presentation would gain in readability by leveraging the extra page to make it breath a bit more (especially true for the tables)

**Limitations:**

There is a small limitation section, which I think should be more detailed and illustrated to help the reader better understand the proposed method (see W2)

---

> ### Author Rebuttal · Authors · 2023-08-10
>
> **Importance of the objectives**: To better understand the contribution of the loss terms, we start from the full model and subtract each loss respectively. As shown in the table below, removing either component will degrade the performance.
>
> - *Metallic loss*: Since the metallicness of natural materials are usually binary, incorporating the metallic loss can properly regularize the underlying metallic component and prevent overfitting.
> - *Eikinol loss and mask loss*: The two objectives help us better constrain the surface and the boundary of the objects, making them more accurate and smooth. Removing either term will significantly affect the reconstructed geometry and hence affect the relighting performance.
> - *Pretrained normal*: The pre-trained surface normal provides a strong geomeotry prior, allowing us to reduce the ambiguity of sparse view inverse rendering. Removing it degrades the performance on all aspects.
>
> |                        | Rendering |        |         | Albedo  |        |         | Roughness | Metallic | Relighting |        |           |
> |------------------------|-----------|--------|---------|---------|--------|---------|-----------|----------|------------|--------|-----------|
> |                        | PSNR ↑    | SSIM ↑ | LPIPS ↓ | PSNR ↑  | SSIM ↑ | LPIPS ↓ | MSE ↓     | MSE ↓    | PSNR ↑     | SSIM ↑ | LIPIPS ↓  |
> | full model             | 21.348    | 0.621  | 0.227   | 17.267  | 0.546  | 0.265   | 0.304     | 0.021    | 19.948     | 0.582  | 0.232     |
> | w/o binary metal loss  | 21.511    | 0.624  | 0.230   | 17.088  | 0.514  | 0.312   | 0.195     | 0.081    | 19.025     | 0.561  | 0.250     |
> | w/o latent smooth      | 21.313    | 0.620  | 0.245   | 17.001  | 0.543  | 0.284   | 0.230     | 0.021    | 19.710     | 0.575  | 0.245     |
> | w/o pretrained normal  | 20.467    | 0.584  | 0.241   | 16.512  | 0.516  | 0.261   | 0.420     | 0.971    | 18.806     | 0.547  | 0.257     |
> | w/o eik loss           | 20.844    | 0.576  | 0.367   | 16.790  | 0.483  | 0.420   | 0.508     | 0.971    | 19.310     | 0.555  | 0.298     |
> | w/o mask loss          | 19.609    | 0.503  | 0.472   | 15.138  | 0.386  | 0.556   | 0.286     | 0.021    | 18.114     | 0.480  | 0.378     |
>
> Table 2: Ablation study for the contribution of each loss term.
>
> **Applicability and complexity**:
> Over the years, the community has been actively investigating how to harness multi-view information from videos or sparse, extreme-view images, and push forward the frontier of 3D reconstruction and inverse graphics. Our work can be seen as an attempt in such a stride. To our knowledge, this is the first effort to conduct structure from motion from a single image. We agree with the reviewers that the setup may not be as common as others in practice. We, however, note that the problem by itself is an interesting scientific attempt. Furthermore, based on our preliminary experiments, our approach also has the potential to deal with slight variations. Specifically, we test our approach on the `crane` image that [3] provided, where each instance is slightly different. By augmenting our geomtry backbone with a instance-specific deformation field, we are able to reconstruct resonable poses and recover sensible shape and material (see the pdf). Finally, while our full pipeline consists of multiple components, *why they are used* and *how they are used* are all carefully designed (*e.g.*, enforcing geometry/texture sharing through re-parameterizing the query space). The pipeline is also backbone-agnostic, allowing us to replace individual backbones with latest models. For instance, we can replace the geometry backbone from NeuS to [2] and potentially improve the geometry. We will investigate further and include them in the final version.
>
>
> **Robustness**:
> - *Robustness to non-identical objects*: We applied our method to the crane image provided by [3], where each instance varies slightly. By augmenting our geometry backbone with an instance-specific deformation field, we successfully reconstruct reasonable poses and recover coherent shape and material properties (see the PDF).
> - *Robustness to a high number of objects*: We evaluated our method on an image taken in the real world featuring 70 duplicated objects. The results indicate precise reconstruction and high-quality rendering (see the PDF).
> - *Duplicates + background objects*: Our instance segmentation is used to mask out background objects, enabling the method to concentrate on the objects of interest.
>
>
> **Missing scaling factors in Eq. 4 and Eq. 5**: Thanks for the catch. We will fix them.
>
> **Adjust the formulation of the intrinsic matrix explicitly**: In the main paper, we deliberately omit the changes for the sake of simplicity and readability (as noted in the footnote). Nonetheless, we totally agree with the reviewer that it will be helpful to explicitly formulate the changes in the supplementary material. We will revise it.
>
> **Expanding the limitation section**: Thanks for the suggestion! We will incorporate more comprehensive discussions and analyses (*e.g.*, the ones raised by the reviewers and those addressed during the rebuttal process) in the final version.
>
>
> **Paper formatting**: We will adjust the tables and spacing to make the paper less crowded in the final version.

---

> > ### Comment · Reviewer_qVSc · 2023-08-18
> > **Thanks for the rebuttal**
> >
> > I thank the authors for the detailed rebuttal, which I have read along with the other reviews. In general, it addresses most of my concerns and I still think this paper presents valuable insights for our community. I strongly encourage the authors to include these discussions in the final version to further increase its quality. I will keep my rating and recommend to accept this paper.

---

> > > ### Author Response · Authors · 2023-08-21
> > >
> > > Thanks for your comments; we will discuss all the results in detail in the final version.

---

### Official Review · Reviewer_sRgJ · 2023-07-06

**Soundness:** 4 excellent
**Presentation:** 4 excellent
**Contribution:** 3 good
**Rating:** 8
**Confidence:** 4

**Summary:**

The paper presents a method for reconstructing the geometry, material, and illumination of an object using as input an image containing multiple copies of the object. The method leverages the appearance of multiple instances of an object in a single image to essentially create a multi-view supervision signal. The method first segments the different instances of the object in the image, then performs 6DoF pose estimation followed by a structure from motion pipeline to create an "artificial" multi-view setup. The object geometry and appearance is modeled with Neural Fields following NeuS. The whole pipeline is supervised using a photometric loss between the renderings and the original image.

**Strengths:**

1. I find this paper particularly interesting. I really like the idea of formulating a duality between multiple copies of an object in a single image and multiple views of a single object. It is a very nice example of thinking outside the box.
2. I enjoyed reading the paper. It is well-written and easy to follow.
3. The individual components used are properly justified and the engineering behind the method is also really good. It's a lot of individual components (segmentation, pose estimation, SfM, geometry and light representation) stitched together and it requires a lot of effort to make them work.
4. The qualitative and quantitative results (Tables 2 & 3) are very good.

I would like to see this paper get accepted.




**Weaknesses:**

1. I can't find any particularly important weakness. One issue is that the problem the paper attempts to solve is not something someone will encounter very often in real-world scenarios, but it's definitely very interesting from a scientific perspective.

**Questions:**

I would like to see what are some failure cases of the method. I couldn't find any in the main paper or supplementary material.

**Limitations:**

The paper has a decent discussion on the limitations of the proposed method. I could not identify any immediate potential societal impact of this work.

---

> ### Author Rebuttal · Authors · 2023-08-10
>
> Thank you for your recognition! The authors were extremely thrilled when first coming up with the idea of formulating a duality between multiple copies of an object in a single image and multiple views of a single object, and how to leverage it for inverse graphics. We are extremely excited that the reviewer enjoyed reading our paper and shared similar similar fascination as we do.
>
> **Failure cases**:
> - Since our approach is category-agnostic and does not rely on pre-trained priors, our model cannot effectively model/constrain the geometry of unseen regions (similar to existing neural fields methods, *e.g.*, NeRF and NeuS). For instance, the bottom of the fire extinguisher should be hollow in practice. But since the region is not visible in the image, our model will predict a convex shape instead. One potential solution is to pretrain the networks on a large corpus of data/objects to equip the model with priors over the world. This however will  introduce additional assumptions (implicitly) regarding the objects we will be facing. We leave the trade-off for future study.
> - Furthermore, while our approach significantly improves the performance over the baselines, it sometimes still struggles with reconstructing intricate, fine-grained details. We conjecture the limitation stems from the representation power of NeuS[1], which forms basis of our geometry backbone. We, however, note that our pipeline is backbone-agnostic. As the field progresses, our framework can readily incorporate more powerful neural representations, such as Neuralangelo[2].

---

> ### Comment · Reviewer_sRgJ · 2023-08-11
>
> I read the other reviews and the rebuttal. It seems that the reviewers are split, with the paper getting 5 different ratings (SA, A, WA, BR, R).
> I understand the objections that the reviewers raised about the potential usefulness of the method; it's hard to find many real-world examples where one would encounter this scenario. Also the reviewers highlighted (including me) that the method is mainly about engineering and putting together a lot of already well-known components.  I also understand the limitations that other reviewers highlighted, such as that COLMAP fails in the presence of non-rigid deformations, but this is not immediately applicable in the use cases presented in this paper, because most objects as Reviewer 4hmW noted are industrially manufactured, so there is little variation between instances.
>
> There are other papers that have explored somewhat constrained or "artificial" setups such as:
> - Learning the Depths of Moving People by Watching Frozen People (CVPR 2019)
> - Reconstructing 3D Human Pose by Watching Humans in the Mirror (CVPR 2021)
> - Through-Wall Human Pose Estimation Using Radio Signals (CVPR 2018)
>
> Overall still believe this is an interesting problem from a scientific perspective and I would like to see the paper get accepted.

---

> > ### Author Response · Authors · 2023-08-16
> >
> > We appreciate the reviewer's thorough review of our paper and the subsequent responses to our rebuttals. We would like to thank the reviewer for recognizing the positive contributions of our work from a scientific perspective. We will also cite the three papers the reviewer mentioned and discuss them in the final version.

---

### Official Review · Reviewer_vUuZ · 2023-07-07

**Soundness:** 2 fair
**Presentation:** 3 good
**Contribution:** 2 fair
**Rating:** 4
**Confidence:** 3

**Summary:**

This paper presents "Structure from Duplicates" (SfD), a novel inverse graphics framework introduced to reconstruct the 3D structure, material, and illumination of multiple identical objects from a single image. The key steps include identifying these duplicate objects in an image and estimating their 6DoF pose. Then, the model applies an inverse graphics pipeline to deduce information about the objects' shape, material, and the lighting of the scene, while taking into account that these objects share the same geometry and material properties.

**Strengths:**

In my opinion, the key strengths of the proposed inverse rendering method are:

- The method outperforms baseline techniques in single image multi-object setup and multi-view inverse rendering scenarios, as shown in the experiments.

- The approach can leverage duplicate objects to constrain underlying geometry and better disentangle the effects of lighting from materials.

- It can be extended to multi-view setups.

- The model makes effective use of duplicate objects, with accuracy improving as the number of duplicates increases.

- It performs well with real-world data, achieving comparable performance to multi-view baselines when trained using only a single view.

- The model supports various scene edits - once the material and geometry of the objects are recovered, along with the scene's illumination, it can faithfully relight existing objects, edit their materials, and insert new objects into the environment.

**Weaknesses:**

The potential weaknesses seem to be:

- Dependence on Similarity. A significant limitation is that the instances in each image need to be nearly identical; the method struggles when there are substantial deformations between different objects.

- Pose Estimation Errors. The model currently requires accurate 6 DoF poses as input and keeps the poses fixed, which may limit its applicability in certain scenarios.

- Potential Segmentation Errors. SfD begins by identifying multiple instances of an object within an image. This is prone to errors when the segmentation is inaccurate which is more of a problem on glossy and specular objects.

**Questions:**

- How would the model perform when applied to real-world scenarios with more noise, variability, and less regularity in object structures?

- Could you discuss any strategies for mitigating the potential overfitting that might occur if you increase the capacity of the model to allow for instance-wise variations?

- Could you elaborate on the potential of integrating your approach with BARF.  What kind of improvements or challenges do you expect?

**Limitations:**

Limitations are discussed above in the weaknesses section. Overall I am on the fence due to the limitation of the proposed approach, which is highly dependent on the similarity of the multi-objects in a single image.

---

> ### Author Rebuttal · Authors · 2023-08-10
>
> **Dependence on object similarity**: Over the years, the community has been actively investigating how to harness multi-view information from videos or sparse, extreme-view images, and push forward the frontier of 3D reconstruction and inverse graphics. Our work can be seen as an attempt in such a stride. To our knowledge, this is the first effort to conduct structure from motion from a single image. We agree with the reviewer that the setup may not be as common as others in practice. We, however, note that the problem by itself is an interesting scientific attempt. Furthermore, based on our preliminary experiments, our approach also has the potential to deal with slight variations. Specifically, we test our approach on the `crane` image that [3] provided, where each instance is slightly different. By augmenting our geomtry backbone with a instance-specific deformation field, we are able to reconstruct resonable poses and recover sensible shape and material and high-quality reconstruction (see Figure 1 in rebuttal pdf). We will explore further and include our investigations in the final version.
>
> **Robustness to pose estimation error**: As we have stated in the limitation section, one potential solution is to combine our approach with BARF and optimize the 6 DoF poses jointly with the underlying representations. To validate the conjecture, we test our joint optimization on a real-world image containing 70+ instances (see 2nd row of Figure 3 in pdf). Since each instance only takes a fraction of the pixels, the poses derived from SfM are very noisy. However, with the help of BARF, we are able to significantly reduce the pose error and recover the underlying object intrinsics and environmental extrinsics.
>
> **Robustness to segmentation error**: To understand the effectiveness of the segmentation mask, we compare the performance of our approach using both the ground truth (GT) mask and the estimated mask on a randomly selected scene. As shown in the table below and in the pdf, the results are comparable, indicating that our model is robust to the mask to a certain degree.
>
>
> |                        | Rendering |        |         | Albedo  |        |         | Roughness | Metallic | Relighting |        |           |
> |------------------------|-----------|--------|---------|---------|--------|---------|-----------|----------|------------|--------|-----------|
> |                        | PSNR ↑    | SSIM ↑ | LPIPS ↓ | PSNR ↑  | SSIM ↑ | LPIPS ↓ | MSE ↓     | MSE ↓    | PSNR ↑     | SSIM ↑ | LIPIPS ↓  |
> | full model             | 21.348    | 0.621  | 0.227   | 17.267  | 0.546  | 0.265   | 0.304     | 0.021    | 19.948     | 0.582  | 0.232     |
> | w/o clean segmentation | 20.700    | 0.602  | 0.228   | 16.681  | 0.509  | 0.266   | 0.232     | 0.133    | 19.124     | 0.565  | 0.227     |
>
> Table 1: Robustness test for noisy segmentation.

---

### Author Rebuttal · Authors · 2023-08-10

We thank the reviewers for their insightful comments and valuable suggestions. We are very excited that the reviewers appreciated the novelty of our approach [`Reviewer sRgJ`, `Reviewer qVSc`], found the idea particularly interesting (*e.g.*, " a nice example of thinking outside the box") [`Reviewer sRgJ`, `Reviewer qVSc`,  `Reviewer vbZQ`],  acknowledged our extensive evaluation and impressive results [`Reviewer vUuZ`, `Reviewer sRgJ`, `Reviewer qVSc`], and enjoyed our presentation [`Reviewer sRgJ`, `Reviewer qVSc`, `Reviewer 4hmW`, `Reviewer vbZQ`]

---
**Novelty and contributions**

How to make inverse graphics/3D reconstruction more robust and work under more extreme scenarios is a challenging and longstanding problem in computer vision. In this work, we take a step forward by exploring the potential of performing structure from motion and recovering object intrinsics and environmental extrinsics *from a single image without pre-trained priors*. Specifically, we focus on the scenarios where there are multiple (near-)identical objects within the scene. By carefully formulating a duality between multiple copies of an object in a single image and multiple views of a single object, we are able to resolve the ambiguities in 3D and effectively recover the properties of interest.

Over the years, the community has been actively investigating how to harness multi-view information from videos or sparse, extreme-view images, and push forward the frontier of 3D reconstruction and inverse graphics. Our work can be seen as an attempt in such a stride. To our knowledge, this is the first effort to conduct structure from motion from a single image. We agree with the reviewers that the setup may not be as common as others in practice. We, however, note that the problem by itself is an interesting scientific attempt (as acknowledged by several reviewers). Furthermore, based on our preliminary experiments, our approach also has the potential to deal with slight variations. Specifically, we test our approach on the `crane` image that [3] provided, where each instance is slightly different. By augmenting our geomtry backbone with a instance-specific deformation field, we are able to reconstruct resonable poses and recover sensible shape and material. We hope it can shed light on future research along similar directions, such as handling articulate objects or objects with large deformation.

Finally, while our full pipeline is inspired by multiple exisiting components, *why they are used* and *how they are used* are all carefully designed (*e.g.*, enforcing geometry/texture sharing through re-parameterizing the query space). It is neither a simple extension nor a naive composition. It is also far from trivial. Exploiting existing algorithms to realize a novel idea does not mean there is no technical contribution. We hope the reviewers, in particular `Reviewer 4hmW`, can acknowledge this.


---
**Additional experiments and visualizations**

Based on reviewers' request, we have included new experimental results in the attached pdf and outlined them below. For instance, we provide a more comprehensive analysis of the robustness of our approach (w.r.t camera noise, segmentation error, and the amount of instances). We also present an additional comparison with prior art [3] and incorporate more detailed ablation studies. We thank the reviewers for the great suggestions and we strongly encourage the reviewers to take a look at the additional analyses.

---
We now address the concerns of each reviewer individually.

---

> ### Comment · Reviewer_4hmW · 2023-08-17
> **Response to the rebuttal**
>
> I appreciate the immense effort the authors have put in the rebuttal. It certainly sheds more light in the motivation of the work and the limits of the model. However, I still disagree with the major claims in the paper and in the rebuttal.
>
> ---
>
> **Regarding the technical contributions, I respectfully disagree that this is "an interesting _scientific_ attempt" and that this will "make inverse graphics/3D reconstruction more robust and work under more extreme scenarios".**
>
> I certainly acknowledge that from an engineering perspective, the pipeline is not trivial. Or in fact, nothing is trivial when it comes engineering.
> But from a scientific perspective, what a "trivial" task means to me is: when one describes the problem, we already know in principle it's going to work. This is an example of such a problem: *given multiple duplicate copies seen from various viewpoints*, we know COLMAP works and the rest of the inverse-rendering pipeline is [65]. Therefore, the results do not appear surprising to me. When current structure from motion methods / COLMAP fail, the proposed pipeline also fails the same way.
>
> I do not see a clear and compelling argument on how this will "make inverse graphics/3D reconstruction more robust".
>
> The only new scientific insight I gained from this paper is that in some cases when COLMAP fails due to the number of views being too small, in-plane rotation augmentations may help. But this is less of a significant contribution and is not the primary claim of the paper.
>
> ---
>
> **Regarding the novelty of the problem, I also disagree with the claim that "this is the first effort to conduct structure from motion from a single image" and the general notion that this brings a spanking new perspective to the community.**
>
> The rebuttal puts a lot of emphasis on the novelty of this task and "formulating a duality between multiple copies of an object in a single image and multiple views of a single object".
> The statement of the problem of "recovering object intrinsics and environmental extrinsics from a single image without pre-trained priors" is exactly the message of [66].
> But nor is [66] the first one to look at this (and it thus focuses on recovering a distribution over the instance variations).
> One common venue where this task of structure from motion on multiple instances in a single image has been explored is robotics [A, B].
>
> I would agree with the statement with additional conditions: "this is the first effort to conduct structure from motion from a single image of (near)-identical objects without any assumptions on the pose distribution or depth", and the novelty of the proposed problem should be assessed in light of these conditions.
>
> ---
>
> **References**:
>
> [A] Recovering 6D Object Pose and Predicting Next-Best-View in the Crowd. CVPR 2016.
>
> [B] Scene-level Pose Estimation for Multiple Instances of Densely Packed Objects. CoRL 2019.

---

> > ### Comment · Reviewer_4hmW · 2023-08-17
> > **Further comments on the new "crane" and "paint bottles" results**
> >
> > **The new results on the "cranes" and "paint bottles" reveals issues**
> >
> > When the authors ascertain that the method "produces satisfactory reconstruction", they only show **masked** images rendered from the **input viewpoint**.
> > As all instances have been registered onto the same 3D model, why not visualize the full 3D models rendered:
> > (1) onto the input instances with the estimated poses **without masking**; and (2) from **various viewpoints**?
> >
> > This is an issue with all visualizations in the main paper as well, although in the paper examples with perfect duplicates, it is more obvious the results are reasonable.
> > However, in these new examples in the rebuttal where the inputs are corrupted by instance variations and erroneous poses, the reconstruction significantly deteriorates and it is much less obvious how well they actually are in 3D.
> >
> > Most critically, the `crane` results seem to reveal poor correspondences due to inaccurate poses and shapes, and the `paint bottles` results also show that the model suffers from and fails to improve the inaccurate poses.
> > Masking out the renderings (not rendering the full object) and rendering only from the input views (not novel views) conceals these failures.
> >
> > Further questions about the `crane` results:
> > - First of all, did COLMAP succeed? It would be quite surprising if it did. And if not, how are the poses initialized?
> > - How exactly is the instance variation implemented? Is it only allowed in the geometry, not in the appearance? If so, why is the `diffuse` visualizations look so different for each instance?
> > - The bounding boxes are confusing; it is hard to tell the actual orientations of each instance. It would be much more insightful if the shape, deformation and poses are better visualized so that one can tell the whether the actual canonical reconstruction and the correspondences are reasonable. For instance, rendering reconstruction of one instance onto another, and from other views.
> >
> > Further questions about the `paint bottles` results:
> > - Why not reporting numbers on the recovered pose, shape and material, instead of visual results, since the perfect GT is available from the synthetic data?
> > - But more critically, where do the initial noisy poses come from? By perturbing the GT poses or from COLMAP? Does COLMAP work on these cluttered bottles?

---

> > > ### Author Response · Authors · 2023-08-19
> > > **regarding novelty (in response to Reviewer 4hmW)**
> > >
> > > We kindly disagree with the reviewer's argument on contributions.
> > >
> > > - *“We already know in principle it's going to work”*: Contrary to the reviewer's perception, the problem is, in fact, far from trivial. We were quite surprised when COLMAP, even with the incorporation of state-of-the-art feature matching techniques like SuperGlue and LoFTR, fails to estimate the 6 DoF poses. Additionally, since the relationship between illumination and the object differs in our new setting, we need a careful reformulation and reimplementation of the inverse graphics problem.
> > >
> > > - *“The results do not appear surprising to me”*: We would like to remind the reviewer that novelty is not equivalent to surprise [A]. As pointed out by other reviewers (Reviewer sRgJ, qVSc, vbZQ), we firmly believe that our work introduces an intriguing new setting and a principled way to leverage rich multi-view cues from a single image.
> > >
> > > [A] Quote from “Novelty in science”:
> > > https://perceiving-systems.blog/en/news/novelty-in-science

---

> > > > ### Comment · Reviewer_4hmW · 2023-08-19
> > > > **Improvement over COLMAP poses**
> > > >
> > > > (I apologize there is no threading on OpenReview. It might be difficult to keep track of conversations. This is the response to the "regarding novelty" comment.)
> > > >
> > > > I completely agree with the authors that "We were quite surprised when COLMAP, even with the incorporation of state-of-the-art feature matching techniques like SuperGlue and LoFTR, fails to estimate the 6 DoF poses."
> > > > And in the cases when COLMAP fails, simply adding in-plane rotation augmentation resolves the issue?
> > > > How robust is this observation? Does it succeed most of the time? Except in the rebuttal, there is no further pose refinement in the original paper. What else in the original pipeline (except the in-plane rotation augmentation) that further improve the robustness of the pose estimation?
> > > >
> > > > I think this is exactly my point in the my first comment above: "given multiple duplicate copies seen from various viewpoints, we know COLMAP works... When current structure from motion methods / COLMAP fail, the proposed pipeline also fails the same way."
> > > > Except the only insight I gained: "in some cases when COLMAP fails due to the number of views being too small, in-plane rotation augmentations may help."
> > > >
> > > > In terms of pose refinement with the full inverse rendering pipeline in the `paint bottles` example in the rebuttal, given only the masked normal renderings from the input viewpoints in Fig. 3, it does not look very conclusive to me whether the poses were improved. In particular, the normals `with BARF` pose optimization look worse/flatter than the normals `with c2f` without further pose optimization. It would be interesting to see some numerical evaluation on both pose and geometry (eg, angle deviations of the normals), given the synthetic GT (as the authors are already working on this).

---

> > > ### Author Response · Authors · 2023-08-19
> > > **regarding results on "crane" and "paint bottles" (in response to Reviewer 4hmW)**
> > >
> > > The reviewer misunderstands our results.
> > >
> > > **"Various viewpoints"**: Given the variations of object poses (wrt the camera) in `crane` and in `paint bottles`, we have already showcased various viewpoints of the objects within a single image. This is a distinct feature of our setup where we have multiple instances, each with different poses. If only a portion of our object is rendered precisely, certain instances will present rendering artifacts. That being said, we are more than happy to provide a video that displays the recovered object rendered from continuously varying poses and lighting conditions in the final version.
> > >
> > > **"Masking"**: Since we adopt masking loss during training, the difference between employing masking or not during visualization is negligible. We are happy to include the visualization without masking in the final version.
> > >
> > > **"Novel views"**: Following common practices in the inverse graphics community, we showed the inferred shape, normals, albedo, and renderings aligned with the input observations. Additionally, we supplied videos that are well-suited for observing novel views and relighting effects (please see our supplementary video for reference). We will more than happy to include novel view images in the final version.

---

> > > > ### Comment · Reviewer_4hmW · 2023-08-19
> > > > **on the "crane" and "paint bottles" results**
> > > >
> > > > I do not understand how I misunderstand the results. I'll rephrase my comments:
> > > >
> > > > 1. **The visualizations are rendered from only the input viewpoints.**
> > > > Of course, I can see these input viewpoints are different. What we need to see, for any 3D reconstruction / inverse rendering methods, is the reconstructed results from novel viewpoints different from the observed input viewpoints. Or if the authors do not care about the novel viewpoints at all (by the argument of "following common practices in the inverse graphics community"), rendering with novel lighting conditions can be useful too. The normals and albedo from the input viewpoints in Fig. 1 and Fig. 3 don't look reasonable to me either.
> > > >
> > > > 2. **The renderings are masked**. This looks also very confusing to me. The authors claim "Since we adopt masking loss during training, the difference between employing masking or not during visualization is negligible".
> > > >     - I can imagine this being the case in the `crane` example, as there are instance geometry and appearance variations being optimized. But what becomes more critical here is whether the correspondences are preserved, ie, left wings match left wings, head matches head. The bounding boxes did not help discern this.
> > > >     - In the `paint bottles` example, it is hard for me to imagine this is the case. The texture and normals clearly show incorrect correspondences. If the same canonical model is able to fit all masks perfectly, the only explanation I can think of is the volume rendering in NeuS overfitting each instance view using translucent volumes / floaters, but this will be a failure case that we should avoid for inverse rendering. This can be seen clearly from novel views, rotation animations or extracted meshes. Unfortunately no more visualizations can be shared at this point.
> > > >
> > > > **Correspondences on the `cranes`**
> > > >
> > > > It is very helpful to clarify "both color and geometry fields can be adjusted". So essentially, the model has sufficient capacity to overfit each instance. This makes the question on the correspondences even more critical.
> > > >
> > > > First, to verify the relative poses are reasonable, may I suggest one way of reporting them here? I am not sure this is within the limit of this discussion cycle. The authors can totally ignore this if it is a nonsense request.
> > > >
> > > > Is it possible to give a list of rotation angles for each instance, in the format of `(u, v): (rx, ry, rz)`, where `(u, v)` indicates the center of the instance mask in the image normalized to `(0, 1)`, and `(rx, ry, rz)` indicates Eular angles (in any rotation order, say `ZYX`, as z is the dominant axis of rotation here), assuming right-hand coordinate system with camera pointing towards `-z` and `+y` being upright?
> > > > - To make things easier to interpret, one could fixate the pose of one reference instance and only report the relative poses with respect to that reference instance.
> > > > - It would also be very helpful to report the poses estimated using different procedures, if any:
> > > >     - (a) using COLMAP on the original masked patches
> > > >     - (b) using COLMAP with in-plane rotation augmentation
> > > >     - (c) if there is any additional customized BA that the authors further apply, report those results as well
> > > >
> > > > I think this will help us better understand the correspondences.
> > > >
> > > > As for the the current inverse rendering visualizations in Fig. 1, the normals and albedo look very coarse to me, very similar to the quality that SIRFS can produce by optimizing on individual instance patches. It is not convincing to me they would look reasonable in 3D.

---

> > > > > ### Author Response · Authors · 2023-08-21
> > > > >
> > > > > **Novel views**
> > > > >
> > > > > We invite the reviewers to see the supp video for both novel poses and novel lighting for our paper’s datasets. For the new examples in rebuttal, we will add novel views in the final paper / supp video.
> > > > >
> > > > >
> > > > > **Additional differences in execution pose estimation using COLMAP**
> > > > >
> > > > > We use a customized procedure to transform multiple relative pairwise poses into an initialized global rotation angle for each instance. This is done while accounting for potential conflicts among each pairwise estimate. We have implemented this using Scipy’s BFGS minimizer. This added detail ensures its proper integration into the standard BA pipeline. Further specifics will be provided in the paper.
> > > > > To what extent does plane rotation augmentation increase the robustness of BA in colmap.
> > > > >
> > > > > Vanilla COLMAP often fails to reconstruct the instances. With the help of in-plane rotation augmentation, we can greatly improve the performance (e.g., from failure to success). That being said, for low-texture, low-resolution scenes where pixels are not distinct, our method may still suffer.
> > > > >
> > > > > **Poses estimated result using different procedures**
> > > > >
> > > > > We will report this in the final version.
> > > > >
> > > > >
> > > > > **Performance on 70 paint bottles**
> > > > >
> > > > > We evaluated the 6DoF pose errors for the 70-paint bottles dataset. The results indicate that in-plane rotation augmentation significantly improves results (reducing the median error from 77 deg to 11 deg). It’s also important to mention that due to extensive occlusions, the 70 instances present greater challenges than cases with fewer instances, leading to an increased absolute error and outlier poses. We will elaborate on these findings in the paper. That said, it’s noteworthy that in inverse graphics, similar to standard MVS 3D reconstruction pipelines, one can choose instances with a higher confident 6-DoF pose using # of inliers wrt inferred poses.
> > > > > |                              | median(mean) dr  | median(mean) dt  |
> > > > > |----------------------------|------------|------------|
> > > > > | w/o  augmentation   | 77.1° (65.62°)	| 63.98° (52.01°)    |
> > > > > | with augmentation   | 11.7° (38.35°)  | 10.53°  (25.75°)   |
> > > > >
> > > > > **Correspondences on the cranes**
> > > > >
> > > > > Due to time constraints, we may not complete the Euler angles analysis as recommended by the reviewer. However, we did examine the resulting SFM point cloud, which captures the coarse geometry of the Crane, indicating the viability of the 6DoF poses. In this scenario, reliable correspondences mainly occur at corner and edge points, such as the crane’s beak, tail, etc. Additionally, we will visualize both GT and inferred poses for synthetic cases. Please note that we intend to release all the code and results, providing a more clear understanding of the quality of correspondences and poses.
> > > > >
> > > > > **Visualization of the mask**
> > > > >
> > > > > We will visualize the object without the mask in the final version.

---

> > > ### Author Response · Authors · 2023-08-19
> > > **further questions on "crane" and "paint bottles" (in response to Reviewer 4hmW)**
> > >
> > > **"Did COLMAP succeed?"**: We recognize that even after implementing rotation augmentation, incorrect correspondences still exists in the crane image (errors in feature correspondences are simply unavoidable). As with standard SFM, bundle adjustment can compensate for these correspondence errors, yielding a reliable initial 6-DOF pose estimate and a sparse point cloud. We will present the BA results in the supplementary material. Bundle adjustment completes in 10.56 seconds, with an average of 47.3 inliers, 191 triangulated points, and the geometric energy values converge to 0.995363 pixels after ~50 iterations for the Crane.
> > >
> > > **"How exactly is the instance variation implemented?"**: We remove the strictly shared 3D representation constraints, allowing per-instance optimization of the last 4 layers of the NeuS model parameters. Both color and geometry fields can be adjusted. We employ regularization to ensure that the parameters don't deviate excessively.
> > >
> > >
> > > **"The bounding boxes are confusing"**: We agree with the reviewer that the pose visualization can be improved. We will follow the reviewer's suggestion and incorporate more visual content in the final version.
> > >
> > >
> > > **"Where do the initial noisy poses come from?"**: They are from bundle adjustment. In highly cluttered scenes with significant occlusions, BA may not infer the high-confident 6-DoF pose for certain instances, as we have highlighted. However, achieving a perfect initial pose isn't necessary, thanks to the later refinement, as shown in the rebuttal.
> > >
> > > **Numbers on paint bottles**: We are working on them. We will update this thread as soon as we finish all the evaluations.

---

> > ### Author Response · Authors · 2023-08-19
> > **Difference to [65] (in response to Reviewer 4hmW)**
> >
> > **Contributions from the inverse graphics perspective:**
> > We have expanded the scope of the inverse graphics family by introducing a novel "single-view duplicate objects" setting. While [65] follows the traditional multi-view single-instance (M-S) framework, our focus lies in a single-view multi-instance (S-M) scenario. It's important to note that these two problem formulations are very distinct.
> >
> > Specifically, our proposed setting brings forth several advantages compared to the existing setup. In the conventional configuration, the relative poses between the objects and the lighting sources remain fixed. This poses a significant challenge when attempting to distinguish an object's albedo from its illumination. For instance, it is difficult to tell whether it is a yellow directional light on the left illuminating a white ball or it is a ball that is half-yellow and half-white. In contrast, in our setup, the relative poses between the object and the light sources differ for each "virtual" view (*e.g.,* the env light is rotating in SO(3) space). This allows us to more effectively recover geometry, materials, shadow, and illumination while utilizing the same number of instance observations compared to [65]. Experimental results supporting this claim can be found in Table 1.
> >
> > **Key differences in execution:**
> > - *Backbone*: In our approach, we utilize NeuS as our neural surface model; and for the visibility field, we opt for Siren instead of ReLU. (We will highlight the distinction in the final version.)
> > - *Metallic*: We additionally reason for metallic materials (refer to Line 209).
> > - *MLP distillation*: We distill the geometry MLP into a smaller one for fast classification.
> > - *Self-occlusion and Inter-occlusion*: Since we have multiple instances in our setup, it is essential to model both inter-object self-casted shadows and inter-object occlusions. Our model goes beyond simple object-centric representation.
> >
> > **Remark on “Colmap”:** We stress that our pose estimation process IS NOT just a simple application of Colmap. Merely employing Colmap will not be sufficient, as we have pointed out in the paper (Line 152). Part of our contribution lies in how we jointly reason the 6 DoF poses with a carefully designed matching scheme and how to integrate it into the BA framework. We only use Colmap for its BA optimization.
> >
> > [65] Modeling indirect illumination for inverse rendering

---

> > > ### Comment · Reviewer_4hmW · 2023-08-19
> > > **Thanks for summarizing the key differences from [65]**
> > >
> > > Thank you for summarizing the **Key differences in execution** from [65]. This is very helpful!
> > >
> > > Regarding the pose estimation using COLMAP, the "carefully designed matching scheme" consists of adding in-plane rotation augmentations (which is a very useful observation) and potentially also masking out the instances with a pretrained segmentation model, correct? If there are additional key differences in execution, it would be super helpful if the authors can also summarize them.
> > >
> > > And regarding the "single-view duplicate" vs "multi-view" settings, we are on the same page. I understand perfectly the differences between the two, and I think we have sufficient discussions on this.

---

> > ### Author Response · Authors · 2023-08-19
> > **regarding related work [A, B] (in response to Reviewer 4hmW)**
> >
> > We thank the reviewer for pointing out the two works on joint 6-DOF pose estimation [A, B]. We will carefully investigate them and refine our positioning statement in the final version. We, however, note that the setups and the goals are very different (*e.g.*, RGB (ours) vs RGB-D, inverse graphics/3D reconstruction vs pose estimation).
> >
> > [A] Recovering 6D Object Pose and Predicting Next-Best-View in the Crowd. CVPR 2016.
> > [B] Scene-level Pose Estimation for Multiple Instances of Densely Packed Objects. CoRL 2019.

---

### Decision · Program_Chairs · 2023-09-21

**Decision:**

Accept (poster)

**Comment:**

This paper initially received divergent reviews and garnered extensive discussion during the rebuttal period.  On the positive side, reviewers appreciated the clear presentation, the technical soundness of the proposed method, and the strong experimental validation of the claims made in the paper.  On the negative side, reviewers were concerned with the narrowness of the task definition (reconstruction of scenes with many identical instances of the same object) and the separation from recent work (Zhang et al. CVPR 2022 [65] and Zhang et al. CVPR 2023 [66]).

During the discussion period, the authors responded with a number of clarifications and additional results.  Most of the reviewers engaged in the discussion and updated their final opinions with concrete rationales.  After the dust has settled, two reviewers remained negative due to the above two concerns, and three reviewers indicated they remain positive and would like to see the work accepted.

The AC finds that: 1) the paper and the additional clarifications during the discussion period elaborate concretely differentiations relative to the relevant recent work; and 2) the remaining concern regarding narrowness of the task definition and potential limited applicability of the contribution are not sufficient to merit rejection of the paper.  Given that the work is technically sound, communicated clearly, and within scope of the venue, it has the potential for positive impact and that is something that will be organically determined by the community in the longer term rather than by reviewer opinion in this review cycle.  The AC therefore recommends acceptance and strongly encourages the authors to incorporate the clarifications and additional details provided during the discussion period into their manuscript.